# DeFoG: Discrete Flow Matching for Graph Generation

Yiming Qin [* 1]   Manuel Madeira [* 1]   Dorina Thanou [1]   Pascal Frossard [1]

## Abstract

Graph generative models are essential across diverse scientific domains by capturing complex distributions over relational data. Among them, graph diffusion models achieve superior performance but face inefficient sampling and limited flexibility due to the tight coupling between training and sampling stages. We introduce DeFoG, a novel graph generative framework that disentangles sampling from training, enabling a broader design space for more effective and efficient model optimization. DeFoG employs a discrete flow-matching formulation that respects the inherent symmetries of graphs. We theoretically ground this disentangled formulation by explicitly relating the training loss to the sampling algorithm and showing that DeFoG faithfully replicates the ground truth graph distribution. Building on these foundations, we thoroughly investigate DeFoG's design space and propose novel sampling methods that significantly enhance performance and reduce the required number of refinement steps. Extensive experiments demonstrate state-of-the-art performance across synthetic, molecular, and digital pathology datasets, covering both unconditional and conditional generation settings. It also outperforms most diffusion-based models with just 5–10% of their sampling steps.

## 1. Introduction

Graph generation has become a fundamental task across diverse fields, from molecular chemistry to social network analysis, due to graphs' capacity to represent complex relationships and generate realistic structured data. Diffusion-based graph generative models (Niu et al., 2020; Jo et al., 2022), particularly those tailored for discrete data (Vignac et al., 2022), have emerged as compelling ap-

proaches, demonstrating pioneering performance in applications such as molecular generation (Irwin et al., 2024), reaction pathway design (Igashov et al., 2024), neural architecture search (Asthana et al., 2024), and combinatorial optimization (Sun & Yang, 2024). Recently, continuous-time discrete diffusion frameworks have further advanced the domain of discrete graph diffusion (Xu et al., 2024; Siraudin et al., 2024). These frameworks leverage the robustness and flexibility of continuous-time modeling, while preserving the natural alignment with the discrete structure of graphs.

Despite their state-of-the-art performance, the training and sampling stages of diffusion-based models is tightly entangled, restricting sampling options to training-phase choices. Thus, optimizing components such as noise schedules or rate matrices requires re-training for each configuration, resulting in prohibitive computational costs. Consequently, these models often adopt a single configuration across graph datasets. This one-size-fits-all approach fails to accommodate the diverse structural characteristics of different datasets, leaving room for further improvement.

In this work, we present DeFoG, a novel graph generative framework that disentangles the training and sampling stages (Figure 1a), addressing the inefficiencies in graph diffusion models and achieving state-of-the-art (SOTA) performance. DeFoG leverages a discrete flow matching (DFM) inspired formulation (Campbell et al., 2024) that we tailor to graph settings. It features a linear interpolation noising process and a continuous-time Markov chain (CTMC)-based denoising process, while ensuring node permutation equivariance and addressing the model expressivity limitations inherent to this data modality (Morris et al., 2019). We demonstrate that training-sampling decoupling not only enhances flexibility but is also provably sound. By theoretically establishing that training loss optimization leads to improved sampling dynamics, DeFoG enables faithful replication of the ground truth graph distribution. To navigate the expanded design space enabled by such disentanglement, we take a critical step by "defogging" this space. Specifically, we explore and propose various sampling methods, including time-adaptive methods and modifications to CTMC rate matrices, to better govern denoising trajectories and align with the unique characteristics of graph datasets.

---

*Equal contribution   [1]EPFL, Lausanne, Switzerland. Correspondence to: Yiming Qin <yiming.qin@epfl.ch>, Manuel Madeira <manuel.madeira@epfl.ch>.

*Proceedings of the 42$^{nd}$ International Conference on Machine Learning*, Vancouver, Canada. PMLR 267, 2025. Copyright 2025 by the author(s).

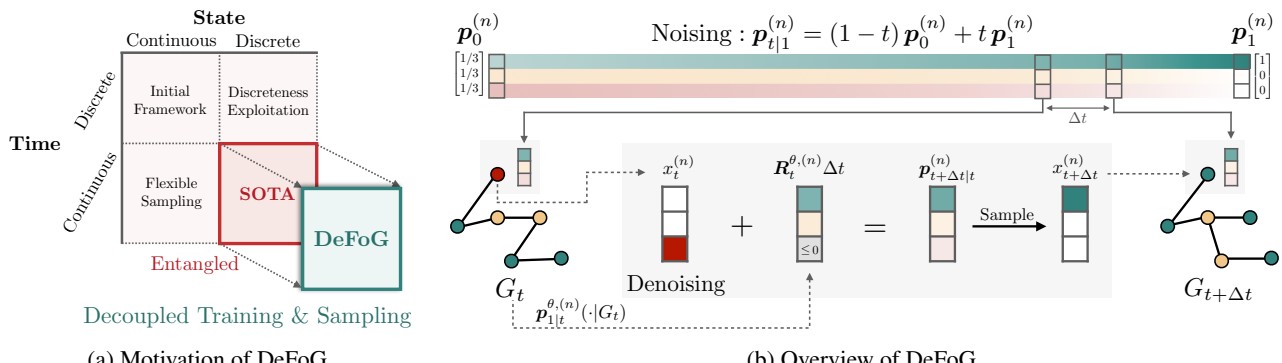

(a) Motivation of DeFoG.                (b) Overview of DeFoG.

Figure 1: (a) DeFoG enhances graph generation by introducing training-sampling decoupling, an orthogonal improvement within graph iterative refinement models, while preserving the sampling flexibility and inherent discreteness exploitation of prior SOTA models. (b) One node, $x^{(n)}$, is selected to illustrate both *noising* and *denoising* processes. For *noising*, DeFoG follows a straight path from the one-hot encoding $p_1$ of the clean node to the initial distribution $p_0$. For *denoising*, a network parameterized by $\theta$ predicts the marginal distributions of the clean graph, there the node's distribution $\boldsymbol{p}_{1|t}^{\theta,(n)}(\cdot|G_t)$ is used to compute its rate matrix $\boldsymbol{R}_t^{\theta,(n)}$ and, subsequently, its probability at the next time point $t + \Delta t$.

Our experiments show that DeFoG achieves SOTA performance across diverse datasets, with near-saturated validity of 99.5%, 96.5%, and 90% on planar, tree, and stochastic block model (SBM) datasets, respectively. On complex molecular data, it achieves 92.8% validity on MOSES (Polykovskiy et al., 2020), surpassing the previous SOTA of 90.5%. Moreover, DeFoG achieves 95.0% and 86.5% validity on planar and SBM datasets, respectively, with only 5–10% of the sampling steps used by diffusion models. This performance surpasses all but one diffusion model on the planar dataset and ranks best on SBM, highlighting substantial efficiency gains. To further highlight the versatility of DeFoG, we also test it in conditional generation tasks for digital pathology, where it largely outperforms existing unconstrained models. Ablation studies further confirm the need of each proposed sampling method and highlight the importance of dataset-specific sampling procedures to effectively address diverse data characteristics.

Our main contributions are as follows:

- We introduce DeFoG, a novel graph generative model that effectively exploits the training-sampling decoupling enabled by its flow-based formulation, significantly enhancing sampling flexibility and efficiency.

- We provide a theoretical foundation for both our training and sampling algorithms, validating the soundness of the disentanglement framework;

- We comprehensively explore DeFoG's design space with novel training and sampling approaches, highlight critical configurations for graph data and attain state-of-the-art performance across diverse datasets.

Overall, DeFoG enables more effective graph generation with reduced computational costs under theoretical guarantees, paving the way for broader adoption of graph generative models in real-world applications.

## 2. Background

In generative modeling, the primary goal is to generate new data samples from the underlying distribution that produced the original data, $\boldsymbol{p}_{\mathrm{data}}$. An effective approach is to learn a mapping between a simpler distribution $\boldsymbol{p}_\epsilon$ that can be easily sampled, and $\boldsymbol{p}_{\mathrm{data}}$.

Iterative refinement models achieve this mapping through a stochastic process over the time interval $t \in [0, 1]$ for variables in discrete state spaces. For the sake of simplicity, we describe here an univariate formulation. At any time $t$, we consider a discrete variable with $Z$ possible values, denoted by $z_t \in \mathcal{Z} = \{1, \ldots, Z\}$. The *marginal distribution* of $z_t$ is represented by the vector $\boldsymbol{p}_t \in \Delta^Z$, with $\Delta^Z = \left\{ \mathbf{u} \in \mathbb{R}^Z \mid \sum_{i=1}^Z u_i = 1, \ u_i \geq 0, \ \forall i \right\}$. The initial distribution is set to a predefined noise distribution, $\boldsymbol{p}_0 = \boldsymbol{p}_\epsilon$, while $\boldsymbol{p}_1 = \boldsymbol{p}_{\mathrm{data}}$ represents the target data distribution. We refer to the mapping $t : 1 \to 0$ as the *noising* process and $t : 0 \to 1$ as the *denoising* process.

DFM (Campbell et al., 2024) builds upon a streamlined noising process. In particular, the noising trajectory $p_{t|1}(z_t|z_1) \in [0, 1]$ is defined through a simple linear interpolation starting from a chosen datapoint $z_1$:

$$p_{t|1}(z_t|z_1) = t\,\delta(z_t, z_1) + (1 - t)\,p_0(z_t), \qquad (1)$$

where $\delta(z_t, z_1)$ is the Kronecker delta (1 when $z_t = z_1$). A usual choice for the initial distribution is the uniform distribution over the state space, $\boldsymbol{p}_0 = [1/Z, \ldots, 1/Z]$.

In the *denoising* stage, DFM leverages a CTMC formulation. In general, a CTMC is characterized by an initial distribution, $\boldsymbol{p}_0$, and a *rate matrix*, $\boldsymbol{R}_t \in \mathbb{R}^{Z \times Z}$ that governs

its evolution across time $t \in [0, 1]$. Specifically, the rate matrix defines the instantaneous transition rates between states, such that:

$$p_{t+\mathrm{d}t|t}(z_{t+\mathrm{d}t}|z_t) = \delta(z_t, z_{t+\mathrm{d}t}) + R_t(z_t, z_{t+\mathrm{d}t})\mathrm{d}t, \quad (2)$$

where $R_t(z_t, z_{t+\mathrm{d}t})$ denotes an entry in the rate matrix. Intuitively, $R_t(z_t, z_{t+\mathrm{d}t})\mathrm{d}t$ yields the probability that a transition from state $z_t$ to state $z_{t+\mathrm{d}t}$ will occur in the next infinitesimal time step $\mathrm{d}t$. By definition, we have $R_t(z_t, z_{t+\mathrm{d}t}) \geq 0$ for $z_t \neq z_{t+\mathrm{d}t}$. Consequently, we further have $R_t(z_t, z_t) = -\sum_{z_{t+\mathrm{d}t} \neq z_t} R_t(z_t, z_{t+\mathrm{d}t})$ to ensure normalization $\sum_{z_{t+\mathrm{d}t}} p_{t+\mathrm{d}t|t}(z_{t+\mathrm{d}t}|z_t) = 1$. Under this definition, the marginal distribution and the rate matrix of a CTCM are related by a conservation law, the Kolmogorov equation, given by $\partial_t \boldsymbol{p}_t = \boldsymbol{R}_t^\top \boldsymbol{p}_t$. If expanded, this expression unveils the time derivative of the marginal distribution as the net balance between the inflow and outflow of probability mass at that state.

Similarly to the noising process of Eq. (1), the denoising is also performed under conditioning on $z_1$. Specifically, we consider a $z_1$-conditional rate matrix, $\boldsymbol{R}_t(\cdot, \cdot|z_1) \in \mathbb{R}^{Z \times Z}$, that will govern the denoising in DFM. Under mild assumptions, Campbell et al. (2024) present a closed-form for a valid conditional rate matrix, i.e., a matrix that verifies the corresponding Kolmogorov equation, for $z_t \neq z_{t+\mathrm{d}t}$, defined as:

$$R_t^*(z_t, z_{t+\mathrm{d}t}|z_1) = \frac{\mathrm{ReLU}[\partial_t p_{t|1}(z_{t+\mathrm{d}t}|z_1) - \partial_t p_{t|1}(z_t|z_1)]}{Z_t^{>0} \, p_{t|1}(z_t|z_1)}$$
(3)

and $Z_t^{>0} = |\{z_t : p_{t|1}(z_t|z_1) > 0\}|$. Again, normalization is performed for the case $z_{t+\mathrm{d}t} = z_t$. Intuitively, $R_t^*$ applies a positive rate to states needing more mass than the current state $z_t$ (details in Appendix B.5). Finally, it can be shown that $R_t(z_t, z_{t+\mathrm{d}t}) = \mathbb{E}_{p_{1|t}(z_1|z_t)}[R_t(z_{t+\mathrm{d}t}, z_t|z_1)]$, which is employed in Eq. (2) for denoising.

While the DFM paradigm enables training-sampling disentanglement, it lacks a complete formulation and empirical validation on graph data. Moreover, how to effectively leverage this disentanglement to enhance sampling performance remains underexplored, particularly for graph-specific tasks. We introduce DeFoG to address these gaps.

## 3. DeFoG Framework

In this section, we present DeFoG (Discrete Flow Matching on Graphs), a novel iterative refinement framework for graph generation that leverages the decoupling of training and sampling stages. We begin by describing its noising and denoising processes, highlighting how they enable this disentanglement, as illustrated in Figure 1b. We theoretically demonstrate that this flexible framework is also robust by proving that optimizing the training loss improves sampling dynamics, ensuring that DeFoG can faithfully

replicate graph distributions. Then, we discuss the expanded design space enabled by DeFoG's disentanglement, which drives key improvements in graph generation performance. Finally, we establish the node permutation equivariance/invariance guarantees of DeFoG.

### 3.1. Learning Discrete Flows over Graphs

We instantiate undirected graphs with $N$ nodes as $G = (x^{1:n:N}, e^{1:i<j:N})$, where $x^{1:n:N} = (x^{(n)})_{1 \leq n \leq N}$ and $e^{1:i<j:N} = (e^{(ij)})_{1 \leq i < j \leq N}$ denote the node and edge sets, respectively, with $x^{(n)} \in \mathcal{X} = \{1, \ldots, X\}$ and $e^{(ij)} \in \mathcal{E} = \{1, \ldots, E\}$. Following standard practice in the field (Vignac et al., 2022; Xu et al., 2024; Siraudin et al., 2024), we consider an edge between every pair of nodes, where one of the edge categories explicitly represents the absence of an edge (i.e., a "non-existing" edge).)

**Noising** We now define the noising process of DeFoG. According to Eq. (1), with shared initial distributions across nodes and edges, denoted as $p_0^{\mathcal{X}}$ and $p_0^{\mathcal{E}}$, respectively, we formulate the noising trajectory by independently adding noise to each node and each edge:

$$p_{t|1}(G_t|G_1) = \prod_n p_{t|1}\left(x_t^{(n)}|x_1^{(n)}\right) \prod_{i<j} p_{t|1}\left(e_t^{(ij)}|e_1^{(ij)}\right).$$

Different $p_0^{\mathcal{X}}$ and $p_0^{\mathcal{E}}$ are further discussed in Appendix C.1.

**Sampling** As formulated in Sec. 2, the denoising process requires simulating a CTMC, driven by its rate matrix $\boldsymbol{R}_t$. We start by sampling a purely noisy graph $G_0$ from the predefined initial distribution $p_0(G_0) = \prod_n p_0^{\mathcal{X}}(x_0^{(n)}) \prod_{i<j} p_0^{\mathcal{E}}(e_0^{ij})$. Then, we progress in the denoising process by employing independent Euler steps for each node and edge, with a finite time step $\Delta t$, i.e., we iteratively sample progressively denoised graphs from $\tilde{p}_{t+\Delta t|t}(G_{t+\Delta t}|G_t)$, given by:

$$\prod_n \tilde{p}_{t+\Delta t|t}^{(n)}(x_{t+\Delta t}^{(n)}|G_t) \prod_{i<j} \tilde{p}_{t+\Delta t|t}^{(ij)}(e_{t+\Delta t}^{(ij)}|G_t). \quad (4)$$

Each term $\tilde{p}_{t+\Delta t|t}^{(n)}(x_{t+\Delta t}^{(n)}|G_t)$ corresponds to the Euler step given in Eq. (2), where the transition dynamics are governed by the rate matrix computed as:

$$R_t^{(n)}\left(x_t^{(n)}, x_{t+\mathrm{d}t}^{(n)}\right) = \mathbb{E}_{p_{1|t}^{(n)}(x_1^{(n)}|G_t)}\left[R_t^{(n)}\left(x_t^{(n)}, x_{t+\Delta t}^{(n)}|x_1^{(n)}\right)\right],$$
(5)

and similarly for $\tilde{p}_{t+\Delta t|t}^{(ij)}(e_{t+\Delta t}^{(ij)}|G_t)$.

**Training** The rate matrix used in the denoising steps above requires the knowledge of the marginal distributions $\boldsymbol{p}_{1|t}^{(n)}(\cdot|G_t) \in \Delta^X$ and $\boldsymbol{p}_{1|t}^{(ij)}(\cdot|G_t) \in \Delta^E$ for all nodes and all edges, respectively. Both are gathered in $\boldsymbol{p}_{1|t}(\cdot|G_t) = \left(\boldsymbol{p}_{1|t}^{1:n:N}(\cdot|G_t), \boldsymbol{p}_{1|t}^{1:i<j:N}(\cdot|G_t)\right)$. Each of these components consists of the clear marginal distribution prediction given a noisy graph $G_t$. However, the computation

---

**Algorithm 1** DeFoG Training

1: **Input:** Graph dataset $\mathcal{D} = \{G^1, \ldots, G^M\}$
2: **while** $f_\theta$ not converged **do**
3:      Sample $G \sim \mathcal{D}$
4:      Sample $t \sim \mathcal{T}$
5:      Sample $G_t \sim p_{t|1}(G_t|G)$      ▷ Noising
6:      $h \leftarrow \text{RRWP}(G_t)$      ▷ Extra features
7:      $\boldsymbol{p}_{1|t}^\theta(\cdot|G_t) \leftarrow f_\theta(G_t, h, t)$    ▷ Denoising prediction
8:      $\text{loss} \leftarrow \text{CE}_\lambda(G, \boldsymbol{p}_{1|t}^\theta(\cdot|G_t))$
9:      optimizer. step(loss)

---

**Algorithm 2** DeFoG Sampling

1: **Input:** # graphs to sample $S$
2: **for** $i = 1$ **to** $S$ **do**
3:      Sample $N$ from train set      ▷ # Nodes
4:      Sample $G_0 \sim p_0(G_0)$
5:      **for** $t = 0$ **to** $1 - \Delta t$ **with step** $\Delta t$ **do**
6:          $h \leftarrow \text{RRWP}(G_t)$      ▷ Extra features
7:          $\boldsymbol{p}_{1|t}^\theta(\cdot|G_t) \leftarrow f_\theta(G_t, h, t)$ ▷ Denoising prediction
8:          $G_{t+\Delta t} \sim \tilde{p}_{t+\Delta t|t}(G_{t+\Delta t}|G_t)$    ▷ Eq. (4)
9:      Store $G_1$

---

of these terms is generally intractable. Instead, we train a neural network, parameterized by $\theta$, to approximate them, $\boldsymbol{p}_{1|t}^\theta(\cdot|G_t)$. To cover different noise levels, we sample $t \sim \mathcal{T}$, where $\mathcal{T}$ is an arbitrary distribution over $[0, 1]$. In DFM, $\mathcal{T}$ is typically set to the uniform distribution, though alternative choices can enhance performance, as later explored in Sec. 3.2. The training loss is naturally formulated as:

$$\mathcal{L}_{\text{DeFoG}} = \mathbb{E}_{t\sim\mathcal{T}, p_1(G_1), p_{t|1}(G_t|G_1)} \text{CE}_\lambda(G_1, \boldsymbol{p}_{1|t}^\theta(\cdot|G_t)),$$

where $\text{CE}_\lambda(G_1, \boldsymbol{p}_{1|t}^\theta(\cdot|G_t))$ is defined as:

$$-\sum_n \log\left(p_{1|t}^{\theta,(n)}(x_1^{(n)}|G_t)\right) - \lambda \sum_{i<j} \log\left(p_{1|t}^{\theta,(ij)}(e_1^{(ij)}|G_t)\right).$$

Here, $\lambda \in \mathbb{R}^+$ is introduced to weight nodes and edges differently to more flexibly capture varying topologies.

**Decoupled Training and Sampling** DeFoG exhibits a clear disentanglement of training and sampling. The training phase focuses on predicting the marginal probabilities of the clean graph $\boldsymbol{p}_{1|t}^\theta(\cdot|G_t)$, while sampling relies on the rate matrix formulation. Importantly, the training process is agnostic to the choice of the $z_1$-conditional rate matrix. Thus, different $z_1$-conditional rate matrices can be employed at sampling time, such as $\boldsymbol{R}^\star(\cdot, \cdot|z_1)$ in Eq. (3). This decoupling of sampling from training provides additional flexibility in DeFoG's design, which can be leveraged to further enhance performance. Notably, we further demonstrate that, upon this decoupling, optimizing DeFoG's training loss improves its sampling dynamics, ensuring the soundness of our framework.

**Corollary 1** (Bounded estimation error of rate matrix for graphs). *Given $t \in [0, 1]$ and graphs $G_t, G_{t+dt}$, and $G_1 \sim p_1(G_1)$, there exist constants $\bar{C}_0, \bar{C}_1, \bar{C}_3 > 0$ such that the rate matrix estimation error can be upper bounded by:*

$$|R_t(G_t, G_{t+dt}) - R_t^\theta(G_t, G_{t+dt})|^2 \leq \bar{C}_0 +$$

$$+ \bar{C}_1 \, \mathbb{E}_{p_1(G_1)} \left[ p_{t|1}(G_t|G_1) \sum_n - \log p_{1|t}^{\theta,(n)}(x_1^{(n)}|G_t) \right]$$

$$+ \bar{C}_2 \, \mathbb{E}_{p_1(G_1)} \left[ p_{t|1}(G_t|G_1) \sum_{i<j} - \log p_{1|t}^{\theta,(ij)}(e_1^{(ij)}|G_t) \right].$$

By taking the expectation over $t \sim \mathcal{T}$ and summing over $G_t$, minimizing the derived upper bound of rate matrix estimation error in Theorem 1 with respect to $\theta$ corresponds directly to minimizing the loss function of DeFoG with $\lambda = 1$. Therefore, we guarantee that our training loss minimization is aligned with accurate rate matrix estimation.

Upon this result, we are now in conditions of justifying DeFoG's approximated denoising algorithm.

**Corollary 2** (Bounded deviation of the generated graph distribution). *Let $p_1$ be the marginal distribution at $t = 1$ of a groundtruth CTMC, $\{G_t\}_{0 \leq t \leq 1}$, and $\tilde{p}_1$ be the marginal distribution at $t = 1$ of its independent-dimensional Euler sampling approximation, with a maximum step size $\Delta t$. Then, under Theorem 6, the following total variation bound holds:*

$$\|p(G_1) - p_{data}\|_{TV} \leq \bar{U}\left(XN + E\frac{N(N-1)}{2}\right)$$

$$+ \bar{B}\left(XN + E\frac{N(N-1)}{2}\right)^2 \Delta t + O(\Delta t),$$

*where $\bar{U}$ and $\bar{B}$ are constant upper bounds for the bound from Theorem 1 and for the denoising process relative to its noising counterpart, respectively, for any $t \in [0, 1]$.*

The first term of the bound captures the estimation error introduced by using the neural network approximation $R_t^\theta(G_t, G_{t+dt})$. From Theorem 1, this term is bounded. The remaining terms arise from the discretization of the CTMC and can be controlled by reducing the step size $\Delta t$. Consequently, the deviation introduced by this approximation can be made arbitrarily small, ensuring that the generated distribution remains close to the groundtruth and validating our graph sampling scheme.

The resulting training and sampling processes are detailed in Algs. 1 and 2. The proofs of Theorems 1 and 2 are provided in Appendix D.1.

### 3.2. Design Space of DeFoG

As described in the previous section, DeFoG benefits from greater flexibility due to its training-sampling decoupling.

For example, it allows the number and size of sampling steps to be adjusted dynamically, unlike the fixed steps in discrete-time diffusion (Vignac et al., 2022), and enables adjustment of the rate matrix without retraining, addressing limitations of continuous-time diffusion graph models (Siraudin et al., 2024; Xu et al., 2024). This decoupling supports extensive, training-free performance optimization during the sampling stage, which is crucial for improving performance and reducing the number of sampling steps. Below, we propose the key components of DeFoG that are enabled by this disentanglement.

**Sample Distortion** In DFM's sampling process, the discretization is performed using equally sized time steps (Alg. 2, line 5). However, this uniformity may fail to preserve key properties during critical intervals where finer control is needed. For instance, smaller steps are essential near the final stages of sampling to prevent sudden edge alterations that could compromise global properties such as planarity. To overcome this limitation, we propose using variable step sizes, allocating smaller, more frequent steps during these critical intervals to better capture essential graph characteristics. Specifically, we apply to each timestep $t$ a bijective, increasing *distortion function $f$* defined for $t \in [0, 1]$, yielding $t' = f(t)$. For example, the choice $f(t) = 2t - t^2$ (referred to as *polydec*) creates monotonically decreasing step sizes, emphasizing the final stages of sampling, where error correction can be most critical. The specific distortion functions employed are described in Appendix B.1. Importantly, we can efficiently (i.e., without re-training) adjust the sample distortion adopted for each dataset to better accommodate its graphs characteristics, leading to significant performance improvements.

**Train Distortion** Once the optimal sampling distortion is identified, it can guide training by highlighting the critical time ranges for graph generation in a specific dataset. This enables adjustments to the training distribution $\mathcal{T}$, skewing it toward these ranges to focus the model's capacity on the most relevant regions. The skewed distributions are obtained by passing uniformly sampled times $t$ through the same distortion functions. While similar strategies in other modalities, such as image generation (Esser et al., 2024), often emphasize intermediate time ranges, we find that optimal time ranges in graph generation vary across datasets. Aligning the distortion function in training with sampling typically enhances the algorithm by focusing on critical time ranges. For instance, for larger datasets, such as drug-sized molecular graphs, the *polydec* distortion function accelerates convergence significantly and provides noticeable performance improvements.

**Target Guidance** The application of time distortions is not the sole avenue for optimizing the sampling process; the design of the conditional rate matrices also offers significant potential for improvement. One promising direc-

tion arises from the goal of better guiding the generation process toward the clean data distribution (Song et al., 2020a). This also aligns with the fundamental design of diffusion and flow matching models, which are structured to predict clean data directly and subsequently use that prediction to generate the denoising trajectory (Ho et al., 2020; Lipman et al., 2023; Vignac et al., 2022). Inspired by these principles, we propose an alternative sampling mechanism that seeks to further amplify the influence of the denoising neural network's predictions in the designed rate matrices, by setting $R_t(z_t, z_{t+\mathrm{d}t}|z_1) = R_t^*(z_t, z_{t+\mathrm{d}t}|z_1) + R_t^\omega(z_t, z_{t+\mathrm{d}t}|z_1)$ for $z_t \neq z_{t+\mathrm{d}t}$, such that:

$$R_t^\omega(z_t, z_{t+\mathrm{d}t}|z_1) = \omega \frac{\delta(z_{t+\mathrm{d}t}, z_1)}{\mathcal{Z}_t^{>0} \ p_{t|1}(z_t|z_1)}. \qquad (6)$$

This adjustment increases the weight of transitions in the rate matrix when $z_{t+\mathrm{d}t} = z_1$, where $z_1$ is the predicted clean data. While moderate increases in $\omega$ significantly enhances performance by steering the generation toward high confidence domains, excessively high $\omega$ leads to performance drop. This behavior is explained in Theorem 10, in Appendix B.2, where we show that target guidance introduces an $O(\omega)$ violation of the Kolmogorov equation. Consequently, finding an optimal value for $\omega$ is essential.

**Stochasticity** Orthogonal to target guidance, there also exists unexplored potential in the design space of conditional rate matrices that preserve the Kolmogorov equation as the standard formulation of $R_t^*(z_t, z_{t+\mathrm{d}t}|z_1)$ does not fully capture this space. As demonstrated by Campbell et al. (2024), for any rate matrix $R_t^{\mathrm{DB}}$ satisfying the detailed balance condition $p_{t|1}(z_t|z_1)R_t^{\mathrm{DB}}(z_t, z_{t+\mathrm{d}t}|z_1) = p_{t|1}(z_{t+\mathrm{d}t}|z_1)R_t^{\mathrm{DB}}(z_{t+\mathrm{d}t}, z_t|z_1)$, the modified rate matrix $R_t^\eta = R_t^* + \eta R_t^{\mathrm{DB}}$, with $\eta \in \mathbb{R}^+$, is also valid. Intuitively, increasing $\eta$ facilitates more transitions to other states while reducing the likelihood of remaining in the same state, thereby increasing stochasticity in the trajectory of the denoising process. This approach can be interpreted as a correction mechanism, as it draws transitions back to states that would otherwise be forbidden according to the rate matrix formulation, as described in Appendix B.5. Additionally, different designs of $R_t^{\mathrm{DB}}$ encode different priors for preferred transitions between states, which we investigate in detail in Appendix B.3.

### 3.3. Permutation Invariance Guarantees

Graph generative models should respect the inherent permutation symmetries of graphs. Accordingly, DeFoG's training and sampling algorithms should be independent of node ordering. This requires that both DeFoG's loss function during training and its probability of generating a specific graph during sampling be permutation invariant. We formally demonstrate these results in Theorem 3, whose proof is in Appendix D.2.1.

**Lemma 3** (Node Permutation Equivariance and Invariance Properties of DeFoG). *For any permutation equivariant denoising neural network, the loss function of DeFoG is permutation invariant, and its sampling probability is permutation invariant.*

We further describe the permutation equivariant denoising neural network of DeFoG in Sec. 5.

## 4. Related Work

**Graph Generative Models**  Graph generation has applications across various domains, including molecular generation (Mercado et al., 2021), combinatorial optimization (Sun & Yang, 2024), and inverse protein folding (Yi et al., 2024). Existing methods for this task generally fall into two main categories. First, *autoregressive* models progressively grow the graph by inserting nodes and edges (You et al., 2018; Liao et al., 2019). Although these methods offer high flexibility in sampling and facilitate the integration of domain-specific knowledge (e.g., for molecule generation, Liu et al. (2018) perform valency checks at each iteration), they suffer from a fundamental drawback: the need to learn a node ordering (Kong et al., 2023; Han et al., 2023), or use a predefined node ordering (You et al., 2018) to avoid the overly large learning space. In contrast, *one-shot* models circumvent such limitation by predicting the entire graph in a single step, enabling the straightforward incorporation of node permutation equivariance/invariance properties. Examples of these approaches include graph-adapted versions of VAEs (Kipf & Welling, 2016), GANs (De Cao & Kipf, 2018), or normalizing flows (Liu et al., 2019). Among one-shot methods, diffusion models have gained prominence for their state-of-the-art performance, attributed to their iterative mapping between noise and data distributions.

**Graph Diffusion**  One of the initial research directions in graph diffusion sought to adapt continuous diffusion frameworks (Sohl-Dickstein et al., 2015; Ho et al., 2020; Song et al., 2020b) for graph-structured data (Niu et al., 2020; Jo et al., 2022; 2024). Those however faced challenges in preserving the inherent discreteness of graphs. In response, discrete diffusion models (Austin et al., 2021) were effectively extended to the graph domain (Vignac et al., 2022; Haefeli et al., 2022), utilizing Discrete-Time Markov Chains to model the stochastic diffusion process. However, this method restricts sampling to the discrete time points used during training. To address this limitation, continuous-time discrete diffusion models incorporating CTMCs have emerged (Campbell et al., 2022), and have been recently applied to graph generation (Siraudin et al., 2024; Xu et al., 2024). Despite employing a continuous-time framework, their optimization space is constrained by training-dependent choices like fixed-rate matrices, limiting further performance gains.

**Discrete Flow Matching**  Flow matching (FM) models emerged as a compelling alternative to diffusion models among iterative refinement generative approaches for continuous state spaces (Lipman et al., 2023; Liu et al., 2023). FM frameworks have been empirically shown to enhance performance and efficiency in image generation (Esser et al., 2024; Ma et al., 2024). To address discrete state spaces, a DFM formulation has been introduced (Campbell et al., 2024; Gat et al., 2024). This approach streamlines its diffusion counterpart by employing a linear interpolation noising process and a more flexible CTMC-based denoising process, whose rate matrices, unlike diffusion models, need not be fixed during training. While other flow-based formulations for graphs have been proposed (Eijkelboom et al., 2024), DeFoG stands out as the first DFM-based model for graphs, leveraging a training-sampling decoupled formulation for improved performance.

## 5. Experiments

This section highlights DeFoG's SOTA performance, enabled by its highly decoupled framework and effective sampling methods. We present DeFoG's performance in generating graphs with diverse topological structures and in molecular datasets with rich prior information. We also provide ablations to showcase the effectiveness and necessity of each proposed sampling method. We highlight the **best result** and the second-best method. Results for generation with 5-10% of steps used by previous SOTA diffusion models are also provided to demonstrate DeFoG's sampling efficiency. We show DeFoG's versatility on conditional generation for digital pathology in Appendix G.1. [1]

**Setup**  To isolate the effect of the network architecture, we build DeFoG on the best-performing graph transformer on generative tasks (Vignac et al., 2022). To enhance expressivity, we incorporate Relative Random Walk Probabilities (RRWP) (Ma et al., 2023; Siraudin et al., 2024) as node and edge features. More details on the architecture in Appendix F.1. Our ablations in Appendix G.4 show that, while RRWP features encode structural properties more efficiently and effectively than prior alternatives, DeFoG's disentangled framework is the primary driver of performance gains. Importantly, the overall architecture, together with RRWP features, is guaranteed to be permutation equivariant (see Appendix D.2.1).

### 5.1. Graph Generation Performance

**Synthetic Graph Generation**  We evaluate DeFoG using the widely adopted *Planar*, *SBM* (Martinkus et al., 2022), and *Tree* datasets (Bergmeister et al., 2023), along with the associated evaluation methodology. In Tab. 1, we report the proportion of generated graphs that are valid, unique,

---

[1]Code at `github.com/manuelmlmadeira/DeFoG`.

Table 1: Graph generation performance on the synthetic datasets: Planar, Tree and SBM. We present the results from five sampling runs, each generating 40 graphs, reported as the mean ± standard deviation. Full version in Tab. 7.

| Model | Class | Planar | | Tree | | SBM | |
|---|---|---|---|---|---|---|---|
| | | V.U.N. ↑ | Ratio ↓ | V.U.N. ↑ | Ratio ↓ | V.U.N. ↑ | Ratio ↓ |
| Train set | — | 100 | 1.0 | 100 | 1.0 | 85.9 | 1.0 |
| GraphRNN (You et al., 2018) | Autoregressive | 0.0 | 490.2 | 0.0 | 607.0 | 5.0 | 14.7 |
| GRAN (Liao et al., 2019) | Autoregressive | 0.0 | 2.0 | 0.0 | 607.0 | 25.0 | 9.7 |
| SPECTRE (Martinkus et al., 2022) | GAN | 25.0 | 3.0 | — | — | 52.5 | 2.2 |
| DiGress (Vignac et al., 2022) | Diffusion | 77.5 | 5.1 | 90.0 | **1.6** | 60.0 | 1.7 |
| EDGE (Chen et al., 2023) | Diffusion | 0.0 | 431.4 | 0.0 | 850.7 | 0.0 | 51.4 |
| BwR (EDP-GNN) (Diamant et al., 2023) | Diffusion | 0.0 | 251.9 | 0.0 | 11.4 | 7.5 | 38.6 |
| BiGG (Dai et al., 2020) | Autoregressive | 5.0 | 16.0 | 75.0 | 5.2 | 10.0 | 11.9 |
| GraphGen (Goyal et al., 2020) | Autoregressive | 7.5 | 210.3 | 95.0 | 33.2 | 5.0 | 48.8 |
| HSpectre (Bergmeister et al., 2023) | Diffusion | 95.0 | 2.1 | **100.0** | 4.0 | 75.0 | 10.5 |
| GruM (Jo et al., 2024) | Diffusion | 90.0 | 1.8 | — | — | 85.0 | **1.1** |
| CatFlow (Eijkelboom et al., 2024) | Flow | 80.0 | — | — | — | 85.0 | — |
| DisCo (Xu et al., 2024) | Diffusion | 83.6±2.1 | — | — | — | 66.2±1.4 | — |
| Cometh (Siraudin et al., 2024) | Diffusion | **99.5**±0.9 | — | — | — | 75.0±3.7 | — |
| DeFoG (5% steps) | Flow | 95.0±3.2 | 3.2±1.1 | 73.5±9.0 | 2.5±1.0 | 86.5±5.3 | 2.2±0.3 |
| DeFoG | Flow | **99.5**±1.0 | **1.6**±0.4 | 96.5±2.6 | **1.6**±0.4 | **90.0**±5.1 | 4.9±1.3 |

Table 2: Large molecule generation performance. Only iterative denoising-based methods are reported here. Respective full versions in Tab. 10 (Guacamol) and Tab. 9 (MOSES), Appendix G.3.

| Model | Guacamol | | | | | MOSES | | | | | | |
|---|---|---|---|---|---|---|---|---|---|---|---|---|
| | Val. ↑ | V.U. ↑ | V.U.N. ↑ | KL div ↑ | FCD ↑ | Val. ↑ | Unique. ↑ | Novelty ↑ | Filters ↑ | FCD ↓ | SNN ↑ | Scaf ↑ |
| Training set | 100.0 | 100.0 | 0.0 | 99.9 | 92.8 | 100.0 | 100.0 | 0.0 | 100.0 | 0.01 | 0.64 | 99.1 |
| DiGress (Vignac et al., 2022) | 85.2 | 85.2 | 85.1 | 92.9 | 68.0 | 85.7 | **100.0** | 95.0 | 97.1 | **1.19** | 0.52 | 14.8 |
| DisCo (Xu et al., 2024) | 86.6 | 86.6 | 86.5 | 92.6 | 59.7 | 88.3 | **100.0** | **97.7** | 95.6 | 1.44 | 0.50 | 15.1 |
| Cometh (Siraudin et al., 2024) | 98.9 | 98.9 | 97.6 | 96.7 | 72.7 | 90.5 | 99.9 | 92.6 | **99.1** | 1.27 | 0.54 | 16.0 |
| DeFoG (10% steps) | 91.7 | 91.7 | 91.2 | 92.3 | 57.9 | 83.9 | 99.9 | 96.9 | 96.5 | 1.87 | 0.50 | **23.5** |
| DeFoG | **99.0** | **99.0** | **97.9** | **97.7** | **73.8** | 92.8 | 99.9 | 92.1 | 98.9 | 1.95 | **0.55** | 14.4 |

and novel (V.U.N.), as well as the average ratio of the usual distances between graph statistics of the generated and test sets relative to the train and test sets (Ratio) to assess sample quality. As shown in Tab. 1, for the Planar dataset, DeFoG achieves the best performance across both metrics, with a nearly saturated V.U.N. value of 99.5%. On the Tree dataset, it is only surpassed by HSpectre, which leverages a local expansion procedure particularly well-suited to hierarchical structures like trees. On the SBM dataset, DeFoG achieves the highest V.U.N. score among all methods, even with just 50 steps.

**Molecular Graph Generation** Molecular design is a prominent real-world application of graph generation. We evaluate DeFoG's performance on this task using the QM9 (Wu et al., 2018), ZINC250k (Sterling & Irwin, 2015), MOSES (Polykovskiy et al., 2020), and Guacamol (Brown et al., 2019) datasets. For QM9, we follow the dataset split and evaluation metrics from Vignac et al. (2022), presenting the results in Appendix F.2.2, Tab. 8. For ZINC250k, we provide the experimental setup in Appendix F. For the larger MOSES and Guacamol datasets, we adhere to the training setup and evaluation metrics established by Polykovskiy et al. (2020) and Brown et al. (2019), re-

spectively, with results in Tabs. 9 and 10. As illustrated in Tab. 2, on Guacamol, it ranks best across all metrics, achieving a nearly saturated validity of 99.0%. Notably, DeFoG achieves over 90% validity with just 10% of the sampling steps, surpassing the well-established baseline DiGress with 500 steps. On MOSES, DeFoG also outperforms diffusion models, achieving SOTA validity of 92.8% while maintaining a high uniqueness of 99.9%.

### 5.2. Efficiency Improvement

We now show that DeFoG enhances both training and sampling efficiency significantly across diverse datasets.

**Sampling Efficiency** Figure 2a highlights the cumulative benefits of sampling approaches from Sec. 3.2. Starting with a vanilla DeFoG model (initially slightly below DiGress), each optimization step, denoted by + symbols, progressively improves performance, culminating in significant gains using only 50 steps on the Planar dataset. This demonstrates that the three sampling approaches are each essential components for optimizing generation.

**Training Efficiency** Figure 2b illustrates convergence curves for the tree and MOSES datasets. We observe that incorporating sampling distortion enhances performance

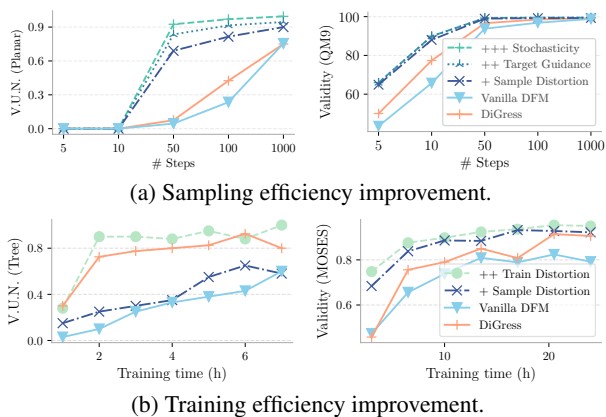

(a) Sampling efficiency improvement.

(b) Training efficiency improvement.

Figure 2: DeFoG's improvements on efficiency.

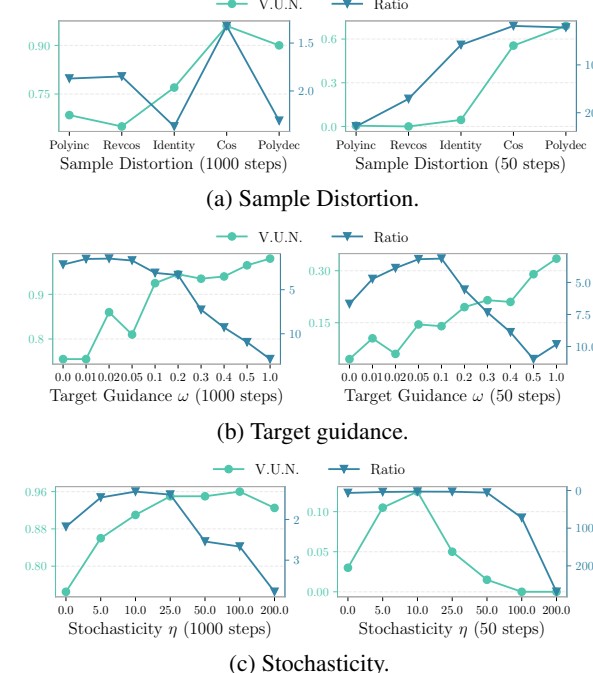

(a) Sample Distortion.

(b) Target guidance.

(c) Stochasticity.

Figure 3: Effect of proposed sampling approaches on the Planar dataset. Higher values on the vertical axis correspond to more favorable values for both V.U.N. and Ratio.

significantly beyond the vanilla implementation, making it particularly useful for generation with undertrained models in resource-constrained settings (see Appendix B.6). Additionally, applying the same optimal distortion found in sampling during training typically yields further gains in convergence (see Appendix C.2). The convergence on some graph datasets may also benefit from an appropriate initial distribution, as shown for SBM in Appendix C.1.

### 5.3. Ablations

Here, we focus on evaluating the impact of different sampling approaches. We start from the vanilla sampling setup and sweep over sample distortion, target guidance, or stochasticity independently. More details in Appendix B.4. For illustration, here we focus on the Planar dataset:

**Sample Distortion** In Figure 3a, we observe that *cos* and *polydec* distortions which emphasize refinement at later steps, perform better on the Planar dataset. This aligns with the intuition that, unlike continuous data undergoing gradual refinement, graphs often experience abrupt transitions due to the random sampling of categorical data. These transitions can violate hard constraints, such as planarity, as categorical values shift abruptly (e.g., from 0 to 1 in one-hot encoding) when $t$ approaches 1. These later steps are thus critical for error detection and correction. On the contrary, for datasets like SBM, where properties are not deterministic and such strict constraints are absent, this refinement does not provide any advantage (see Appendix B.1).

**Target Guidance** As shown in Figure 3b, $\omega$ improves both V.U.N. and Ratio by biasing generation toward predicted clean data. However, excessive $\omega$ skews the generated distribution to the high density regions of training set distribution, leading to higher planarity (reflected by V.U.N.) but increased divergence from test graphs (reflected by Ratio). We also observe that the Ratio only begins to worsen at 0.1 with 50 steps, compared to 0.02 with 1000 steps. This demonstrates that target guidance is particularly effective with fewer steps, where the genera-

tion process becomes more challenging due to larger transitions, as it steers the model toward higher-confidence regions, safeguarding generative performance.

**Stochasticity** As shown in Figure 3c, a moderate level of stochasticity benefits both metrics, while extreme values introduce excessive noise, disrupting the generation process. This indicates a sweet spot exists between effective error correction and over-stochasticity. Furthermore, the V.U.N. of generated graphs decreases with increasing $\eta$ values when more steps are utilized (drop after $\eta = 100$ for 1000 steps *vs.* $\eta = 10$ for 50 steps). As this approach preserves the Kolmogorov equation, it benefits from more sampling steps to mitigate simulation errors.

## 6. Conclusion

We introduce DeFoG, a novel discrete flow matching framework for graphs. This formulation enables training-sampling decoupling, which we ground theoretically to ensure faithful graph distribution modeling. Extensive experiments demonstrate the importance of our proposed strategies in achieving state-of-the-art performance on synthetic and molecular graph generation tasks. DeFoG currently employs a simple but efficient hyperparameter search for sampling, yielding impressive results and underscoring its potential for further improvement with more advanced search algorithms. Future work will further explore purely

sampling-stage methods to enhance performance. Generating high-quality graphs in even fewer steps and scaling to larger graphs also remain key challenges.

## Acknowledgements

We would like to thank Clément Vignac and Andrew Campbell for the useful discussions and suggestions.

## Impact Statement

The primary objective of this paper is to advance graph generation under a more flexible framework, with applications spanning general graph generation, molecular design, and digital pathology. The ability to generate graphs with discrete labels can have broad-reaching implications for fields such as drug discovery and diagnostic technologies. While this development has the potential to bring about both positive and negative societal or ethical impacts, particularly in areas like biomedical and chemical research, we currently do not foresee any immediate societal concerns associated with the proposed methodology.

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

# Appendix Overview

In the Appendix, we provide additional details organized as follows:

1. Appendix A: Contextualizing Related Research.

2. Appendix B: Sample Optimization.

3. Appendix C: Train Optimization.

4. Appendix D: Theoretical Results.

5. Appendix E: Conditional Generation.

6. Appendix F: Experimental Details.

7. Appendix G: Additional Results.

# A. Contextualizing Related Research

In this section, we further contextualize DeFoG within the scope of related work. We begin by introducing the methods used for comparison with DeFoG in Appendix A.1. Subsequently, we outline the key distinctions between DeFoG and existing diffusion-based graph generative models in Appendix A.2.

## A.1. Overview of Compared Methods

In Sec. 5, we evaluate DeFoG against a diverse set of graph generative models, which we introduce below:

- GraphRNN (You et al., 2018) and GRAN (Liao et al., 2019), two pioneering autoregressive models for graph generation;
- SPECTRE (Martinkus et al., 2022), a spectrally conditioned GAN-based model for graph generation;
- DiGress (Vignac et al., 2022), the first discrete diffusion model for graph generation;
- EDGE (Chen et al., 2023), a discrete diffusion model leveraging graph sparsity and degree guidance for scalability.
- BwR (Diamant et al., 2023), which focuses on efficient graph representations via bandwidth restriction schemes that are compatible with various graph generation models. We report its results in combination with EDP-GNN (Niu et al., 2020), which was the first graph diffusion model;
- BiGG (Dai et al., 2020), an autoregressive model that exploits graph sparsity and training parallelization to scale to larger graphs;
- GraphGen (Goyal et al., 2020), a scalable autoregressive approach utilizing graph canonization with minimum DFS codes, notable for being domain-agnostic and inherently supporting attributed graphs;
- HSpectre (Bergmeister et al., 2023), a hierarchical graph generation method that utilizes a score-based formulation for iterative local expansion steps;
- DisCo (Xu et al., 2024) and Cometh (Siraudin et al., 2024), two continuous-time discrete diffusion models for graph generation;
- GruM (Jo et al., 2024), which employs a diffusion mixture to explicitly learn the final graph topology and structure;
- CatFlow (Eijkelboom et al., 2024), which results from the instantiation of variational flow matching to graph generation.

## A.2. DeFoG and Graph Diffusion Models

In this section, we contextualize DeFoG in relation to existing graph diffusion models.

### A.2.1. FROM CONTINUOUS TO DISCRETE STATE-SPACES

Early diffusion-based graph generative models extended continuous diffusion and score-based methods from image generation to graphs by relaxing adjacency matrices into continuous state-spaces (Niu et al., 2020; Jo et al., 2022). However, this approach overlooks the inherent discreteness of graph-structured data, resulting in topologically uninformed noising processes. For instance, these methods often destroy graph sparsity and generate noisy complete graphs (Vignac et al., 2022; Xu et al., 2024; Siraudin et al., 2024), making it more challenging for denoising neural networks to recover meaningful structural properties from the noisy inputs. Some recent formulations operating on continuous state-spaces have tried to overcome these limitations: GruM (Jo et al., 2024) introduces an endpoint-conditioned diffusion mixture strategy to enhance accuracy by explicitly learning final graph structures, while CatFlow (Eijkelboom et al., 2024) proposes variational flow matching to handle categorical data more effectively.

Alternatively, discrete diffusion models have emerged as a more natural solution, directly preserving the discrete nature of graph data (Vignac et al., 2022; Haefeli et al., 2022). These models have demonstrated state-of-the-art performance across a variety of applications, including neural architecture search (Asthana et al., 2024), combinatorial optimization (Sun & Yang, 2024), molecular generation (Irwin et al., 2024), and reaction pathway design (Igashov et al., 2024).

DeFoG aligns with this second family of methods, modeling nodes and edges in discrete state-spaces to leverage the structural properties of graph data effectively.

### A.2.2. FROM DISCRETE TO CONTINUOUS TIME

The initial discrete-time diffusion frameworks for graph generation (Vignac et al., 2022; Haefeli et al., 2022) were built upon Discrete Denoising Diffusion Probabilistic Models (D3PMs) (Austin et al., 2021), which operate with a fixed partitioning of time. This discretization constrains the model to denoise at specific time points and ties the sampling process to the same fixed time steps used during training, leading to a rigid coupling between the training and sampling stages. Such inflexibility in time discretization can limit the quality of generated graphs.

In contrast, continuous-time discrete diffusion frameworks (Campbell et al., 2022; Sun et al., 2023) overcome these limitations by enabling the model to denoise at arbitrary time points within a continuous interval (typically between 0 and 1). This flexibility allows the time discretization strategy for sampling to be selected post-training, enabling the use of advanced sampling techniques (Jolicoeur-Martineau et al., 2021; Zhang & Chen, 2023; Salimans & Ho, 2022; Chung et al., 2022; Song et al., 2020b; Dockhorn et al., 2022) to improve generation performance. These continuous-time frameworks have been successfully extended to graph generative models (Xu et al., 2024; Siraudin et al., 2024), achieving notable improvements.

DeFoG follows a continuous-time formulation, leveraging its flexibility in sampling to achieve enhanced performance while maintaining the strengths of discrete state-space modeling.

### A.2.3. FROM CONTINUOUS-TIME DISCRETE DIFFUSION TO DISCRETE FLOW MATCHING

While both continuous-time discrete diffusion and discrete flow matching (DFM) share the CTMC formulation for the denoising process, they differ fundamentally in the formulation of the noising process. Continuous-time discrete diffusion-based graph generative models (Xu et al., 2024; Siraudin et al., 2024) define the noising process as a CTMC, akin to the denoising process. However, this approach imposes two significant limitations:

1. **Incomplete Coupling of Training and Sampling**: The rate matrices of the noising and denoising processes are explicitly interrelated, and the noising rate matrix must be fixed during training. This restricts the sampling stage, preventing full decoupling of training and sampling.

2. **Limited Design Space**: The noising process must be derived analytically, which is not straightforward and is only feasible for rate matrices suitable for matrix exponentiation. Additionally, the denoising rate matrix is implicitly defined during training, constraining the flexibility of the denoising trajectory at sampling time (e.g., fixing the level of stochasticity).

In contrast, DeFoG allows for direct prescription of the noising process, $p_{t|1}$, without these constraints. The rate matrix for the denoising process is selected exclusively at sampling time, fully decoupling the training and sampling stages. This flexibility enables performance optimization during sampling, such as tuning the stochasticity of the denoising trajectory via $R_t^{\mathrm{DB}}$ or adjusting target guidance magnitude with $R_t^\omega$.

The benefits of this decoupled framework are evident in Figures 2, 7 and 8, which demonstrate that the vanilla DeFoG configuration alone does not outperform existing diffusion-based graph generative models. However, our extensive sampling optimization pipeline capitalizes on DeFoG's flexible design space to achieve state-of-the-art results. These observations align with findings in iterative refinement methods across other data modalities. For instance, Karras et al. (2022) elaborate on the benefits of stochasticity adjustment in denoising trajectories within diffusion models for image generation.

For a comprehensive discussion of the differences between continuous-time discrete diffusion and DFM frameworks, see Appendix H of Campbell et al. (2024).

### A.2.4. MIXED INTEGRATION OF CONTINUOUS AND DISCRETE STATE-SPACES

Integrating continuous and categorical data within graph generative models is an important challenge, as many real-world applications involve heterogeneous data types (e.g., molecular graphs containing atomic coordinates alongside categorical atom and bond types). A recent example addressing this challenge is GBD (Liu et al., 2024), which incorporates beta diffusion to jointly model both continuous and discrete variables. Similarly, DeFoG is amenable to formulations involving mixed data types by leveraging an approach akin to MiDi (Vignac et al., 2023), independently factorizing continuous and discrete variables. However, explicitly exploring this integration is beyond the scope of this work.

# B. Sample Optimization

In this section, we explore the proposed sampling optimization in more detail. We start by analysing the different time distortion functions in Appendix B.1. Next, in Appendix B.2, we prove that the proposed target guidance mechanism actually satisfies the Kolmogorov equation, thus yielding valid rate matrices and, in Appendix B.3, we provide more details about the detailed balance equation and how it widens the design space of rate matrices. In Appendix B.4, we also describe the adopted sampling optimization pipeline. Finally, in Appendix B.5, we provide more details to better clarify the dynamics of the sampling process.

## B.1. Time Distortion Functions

In Sec. 3, we explore the utilization of different *distortion functions*, i.e., functions that are used to transform time. The key motivation for employing such functions arises from prior work on flow matching in image generation, where skewing the time distribution during training has been shown to significantly enhance empirical performance (Esser et al., 2024). In practical terms, this implies that the model is more frequently exposed to specific time intervals. Mathematically, this transformation corresponds to introducing a time-dependent re-weighting factor in the loss function, biasing the model to achieve better performance in particular time ranges.

In our case, we apply time distortions to the probability density function (PDF) by introducing a function $f$ that transforms the original uniformly sampled time $t$, such that $t' = f(t)$ for $t \in [0, 1]$. These time distortion functions must satisfy certain conditions: they must be monotonic, with $f(0) = 0$ and $f(1) = 1$. Although the space of functions that satisfy these criteria is infinite, we focus on five distinct functions that yield fundamentally different profiles for the PDF of $t'$. Our goal is to gain intuition about which time ranges are most critical for graph generation and not to explore that function space exhaustively. Specifically:

- *Polyinc*: $f(t) = t^2$, yielding a PDF that decreases monotonically with $t'$;

- *Cos*: $f(t) = \frac{1 - \cos \pi t}{2}$, creating a PDF with high density near the boundaries $t' = 0$ and $t' = 1$, and low for intermediate $t'$;

- *Identity*: $f(t) = t$, resulting in a uniform PDF for $t' \in [0, 1]$;

- *Revcos*: $f(t) = 2t - \frac{1 - \cos \pi t}{2}$, leading to high PDF density for intermediate $t'$ and low density at the extremes $t' = 0$ and $t' = 1$;

- *Polydec*: $f(t) = 2t - t^2$, where the PDF increases monotonically with $t'$.

The PDF resulting from applying a monotonic function $f$ to a random variable $t$ is given by:

$$\phi_{t'}(t') = \phi_t(t) \left| \frac{\mathrm{d}}{\mathrm{d}t'} f^{-1}(t') \right|,$$

where $\phi_t(t)$ and $\phi_{t'}(t')$ denote the PDFs of $t$ and $t'$, respectively. In our case, $\phi_t(t) = 1$ for $t \in [0, 1]$. The distortion functions and their corresponding PDFs are illustrated in Figure 4.

One of the strategies the proposed in **sampling** optimization procedure is the use of variable step sizes throughout the denoising process. This is achieved by mapping evenly spaced time points (DeFoG's vanilla version) through a transformation that follows the same constraints as the training time distortions discussed earlier. We employ the same set of time distortion functions, again not to exhaustively explore the space of applicable functions, but to gain insight into how varying step sizes affect graph generation. The expected step sizes for each distortion can be directly inferred from Figure 4. For instance, the *polydec* function leads to progressively smaller time steps, suggesting more refined graph edits in the denoising process as $t'$ approaches 1.

Note that even though we apply the same time distortions for both training and sample stages, in each setting they have different roles: in training, the time distortions skew the PDFs from where $t'$ is sampled, while in sampling they vary the denoising step sizes.

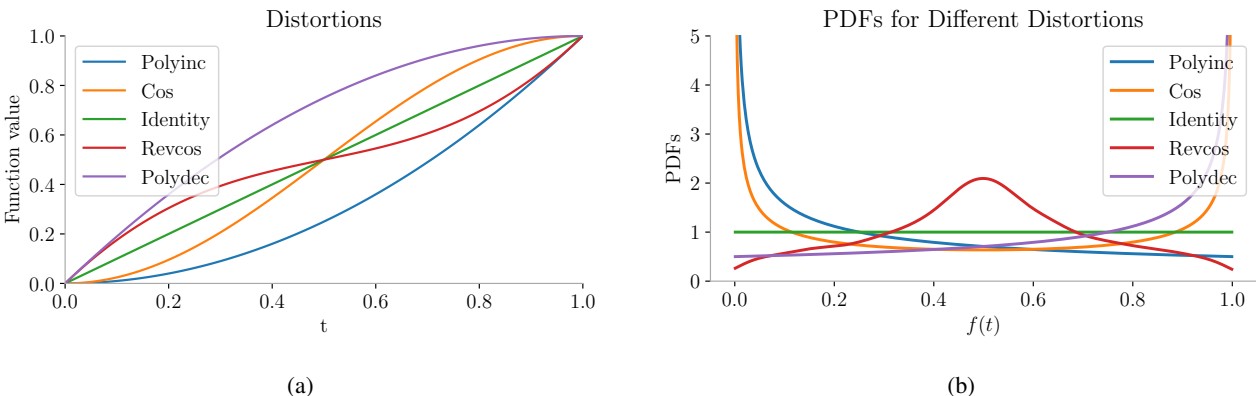

Figure 4: (a) The five distortion functions explored. (b) The resulting PDFs for the five distortion functions. For polydec, identity, and polyinc, they were computed in closed-form. For revcos and cos, they were simulated with $10^4$ repetitions.

### B.2. Target Guidance

In this section, we demonstrate that the proposed *target guidance* design for the rate matrices violates the Kolmogorov equation with an error that is linear in $\omega$. This result indicates that a small guidance factor effectively helps fit the distribution, whereas a larger guidance factor, as shown in Figure 9, while enhancing topological properties such as planarity, increases the distance between generated and training data on synthetic datasets according to the metrics of average ratio. Similarly, for molecular datasets, this also leads to an increase in validity and a decrease in novelty by forcing the generated data to closely resemble the training data.

**Lemma 10** (Rate matrices for target guidance). *Let $R_t^\omega(z_t, z_{t+dt}|z_1)$ be defined as:*

$$R_t^\omega(z_t, z_{t+dt}|z_1) = \omega \frac{\delta(z_1, z_{t+dt})}{Z_t^{>0} \, p_{t|1}(z_t|z_1)}. \tag{7}$$

*Then, the univariate rate matrix $R_t^{\mathrm{TG}}(z_t, z_{t+dt}|z_1) = R_t^*(z_t, z_{t+dt}|z_1) + R_t^\omega(z_t, z_{t+dt}|z_1)$ violates the Kolmogorov equation with an error of $-\frac{\omega}{Z_t^{>0}}$ when $z_t \neq z_1$, and an error of $\omega \frac{Z_t^{>0}-1}{Z_t^{>0}}$ when $z_t = z_1$.*

*Proof.* In the remaining of the proof, we consider the case $z_t \neq z_1$. We consider the same assumptions as Campbell et al. (2024):

- $p_{t|1}(z_t|z_1) = 0 \Rightarrow R_t^*(z_t, z_{t+dt}|z_1) = 0$;

- $p_{t|1}(z_t|z_1) = 0 \Rightarrow \partial_t p_{t|1}(z_t|z_1) = 0$ ("dead states cannot ressurect").

The $z_1$-conditioned Kolmogorov equation is given by:

$$\partial_t p_{t|1}(z_t|z_1) = \sum_{z_{t+dt} \neq z_t} R_t(z_{t+dt}, z_t|z_1) p_{t+dt|1}(z_{t+dt}|z_1) - \sum_{z_{t+dt} \neq z_t} R_t(z_t, z_{t+dt}|z_1) p_{t|1}(z_t|z_1) \tag{8}$$

We denote by RHS and LHS the right-hand side and left-hand side, respectively, of Eq. (8). For the case in which $p_{t|1}(z_t|z_1) > 0$, we have:

$$\text{RHS} = \sum_{z_{t+dt} \neq z_t, p_{t+dt|1}(z_{t+dt}|z_1)>0} (R_t^*(z_{t+dt}, z_t|z_1) + R_t^\omega(z_{t+dt}, z_t|z_1))p_{t+dt|1}(z_{t+dt}|z_1)$$

$$- \sum_{z_{t+dt} \neq z_t, p_{t+dt|1}(z_{t+dt}|z_1)>0} (R_t^*(z_t, z_{t+dt}|z_1) + R_t^\omega(z_t, z_{t+dt}|z_1))p_{t|1}(z_t|z_1)$$

$$= \sum_{z_{t+dt} \neq z_t, p_{t+dt|1}(z_{t+dt}|z_1)>0} R_t^*(z_{t+dt}, z_t|z_1)p_{t+dt|1}(z_{t+dt}|z_1) - R_t^*(z_t, z_{t+dt}|z_1)p_{t|1}(z_t|z_1)$$

$$+ \sum_{z_{t+dt} \neq z_t, p_{t+dt|1}(z_{t+dt}|z_1)>0} R_t^\omega(z_{t+dt}, z_t|z_1)p_{t+dt|1}(z_{t+dt}|z_1) - R_t^\omega(z_t, z_{t+dt}|z_1)p_{t|1}(z_t|z_1),$$

For the first sum, we have:

$$\sum_{z_{t+dt} \neq z_t, p_{t+dt|1}(z_{t+dt}|z_1)>0} R_t^*(z_{t+dt}, z_t|z_1)p_{t+dt|1}(z_{t+dt}|z_1) - R_t^*(z_t, z_{t+dt}|z_1)p_{t|1}(z_t|z_1)$$

$$= \partial_t p_{t|1}(z_t|z_1).$$

since the $z_1$-conditioned $R_t^*$ generates $p_{t|1}$.

For the second sum, we have:

$$\sum_{z_{t+dt} \neq z_t, p_{t+dt|1}(z_{t+dt}|z_1)>0} R_t^\omega(z_{t+dt}, z_t|z_1)p_{t+dt|1}(z_{t+dt}|z_1) - R_t^\omega(z_t, z_{t+dt}|z_1)p_{t|1}(z_t|z_1) =$$

$$= \sum_{z_{t+dt} \neq z_t, p_{t+dt|1}(z_{t+dt}|z_1)>0} \omega \frac{\delta(z_1, z_t)}{Z_t^{>0}} \frac{1}{p_{t+dt|1}(z_{t+dt}|z_1)} p_{t+dt|1}(z_{t+dt}|z_1)$$

$$- \sum_{z_{t+dt} \neq z_t, p_{t+dt|1}(z_{t+dt}|z_1)>0} \omega \frac{\delta(z_1, z_{t+dt})}{Z_t^{>0}} \frac{1}{p_{t|1}(z_t|z_1)} p_{t|1}(z_t|z_1)$$

$$= \frac{\omega}{Z_t^{>0}} \sum_{z_{t+dt} \neq z_t, p_{t+dt|1}(z_{t+dt}|z_1)>0} (\delta(z_1, z_t) - \delta(z_1, z_{t+dt}))$$

If $z_1 \neq z_t$, we have:

$$\sum_{z_{t+dt} \neq z_t, p_{t+dt|1}(z_{t+dt}|z_1)>0} R_t^\omega(z_{t+dt}, z_t|z_1)p_{t+dt|1}(z_{t+dt}|z_1) - R_t^\omega(z_t, z_{t+dt}|z_1)p_{t|1}(z_t|z_1) =$$

$$= \frac{\omega}{Z_t^{>0}} \sum_{z_{t+dt} \neq z_t, p_{t+dt|1}(z_{t+dt}|z_1)>0} (\delta(z_1, z_t) - \delta(z_1, z_{t+dt}))$$

$$= \frac{\omega}{Z_t^{>0}} \sum_{z_{t+dt} \neq z_t, p_{t+dt|1}(z_{t+dt}|z_1)>0} (0 - \delta(z_1, z_{t+dt}))$$

$$= -\frac{\omega}{Z_t^{>0}},$$

Here, we apply the property that $z_t \neq z_1$, which indicates that $\delta(z_1, z_t) = 0$ and that there exists one and only one $z_{t+dt} \in \{z_{t+dt}, z_{t+dt} \neq z_t\}$ such that $z_{t+dt} = z_1$, which verifies that $p_{t+dt|1}(z_{t+dt}|z_1) > 0$ — a condition satisfied by any initial distribution proposed in this work when $t$ strictly positive—the sum simplifies to $-\frac{\omega}{Z_t^{>0}}$.

If $z_1 = z_t$, we have:

$$\sum_{z_{t+dt} \neq z_t, p_{t+dt|1}(z_{t+dt}|z_1) > 0} R_t^\omega(z_{t+dt}, z_t|z_1) p_{t+dt|1}(z_{t+dt}|z_1) - R_t^\omega(z_t, z_{t+dt}|z_1) p_t(z_t|z_1) =$$

$$= \frac{\omega}{Z_t^{>0}} \sum_{z_{t+dt} \neq z_t, p_{t+dt|1}(z_{t+dt}|z_1) > 0} (\delta(z_1, z_t) - \delta(z_1, z_{t+dt}))$$

$$= \frac{\omega}{Z_t^{>0}} \sum_{z_{t+dt} \neq z_t, p_{t+dt|1}(z_{t+dt}|z_1) > 0} (1 - 0)$$

$$= \omega \frac{Z_t^{>0} - 1}{Z_t^{>0}},$$

$\square$

**Intuition**  The aim of *target guidance* is to reinforce the transition rate to the state predicted by the probabilistic model, $z_1$. The $\omega$ term is an hyperparameter used to control the target guidance magnitude.

### B.3. Detailed Balance, Prior Incorporation, and Stochasticity

Campbell et al. (2024) show that although their $z_1$-conditional formulation of $R_t^*$ generates $p_{t|1}$, it does not span the full space of valid rate matrices — those that satisfy the conditional Kolmogorov equation (Eq. (8)). They derive sufficient conditions for identifying other valid rate matrices. Notably, they demonstrate that matrices of the form

$$R_t^\eta := R_t^* + \eta R_t^{\text{DB}},$$

with $\eta \in \mathbb{R}^{\geq 0}$ and $R_t^{\text{DB}}$ any matrix that verifies the *detailed balance condition*:

$$p_{t|1}(z_t|z_1) R_t^{\text{DB}}(z_t, z_{t+dt}|z_1) = p_{t|1}(z_{t+dt}|z_1) R_t^{\text{DB}}(z_{t+dt}, z_t|z_1), \tag{9}$$

still satisfy the Kolmogorov equation. The detailed balance condition ensures that the outflow, $p_{t|1}(z_t|z_1) R_t^{\text{DB}}(z_t, z_{t+dt}|z_1)$, and inflow, $p_{t|1}(z_{t+dt}|z_1) R_t^{\text{DB}}(z_{t+dt}, z_t|z_1)$, of probability mass to any given state are perfectly balanced. Under these conditions, this additive component's contribution to the Kolmogorov equation becomes null (similar to the target guidance, as shown in the proof of of Theorem 10, in Appendix B.2).

A natural question is how to choose a suitable design for $R_t^{\text{DB}}$ from the infinite space of detailed balance rate matrices. As depicted in Figure 5, this flexibility can be leveraged to incorporate priors into the denoising model by encouraging specific transitions between states. By adjusting the sparsity of the matrix entries, additional transitions beyond those prescribed by $R_t^*$ can be introduced. In the general case, transitions between all states are possible; in the column case, a specific state centralizes all potential transitions; and in the single-entry case, only transitions between two states are permitted. These examples merely illustrate some possibilities and do not exhaust the range of potential $R_t^{\text{DB}}$ designs. The matrix entries can be structured by considering the following reorganization of terms of Eq. (9):

$$R_t^{\text{DB}}(z_{t+dt}, z_t|z_1) = \frac{p_{t|1}(z_t|z_1)}{p_{t|1}(z_{t+dt}|z_1)} R_t^{\text{DB}}(z_t, z_{t+dt}|z_1).$$

Therefore, a straightforward approach is to assign the lower triangular entries of the rate matrix as $R_t^{\text{DB}}(z_t, z_{t+dt}|z_1) = p_{t|1}(z_{t+dt}|z_1)$, and the upper triangular entries as $R_t^{\text{DB}}(z_{t+dt}, z_t|z_1) = p_{t|1}(z_t|z_1)$. The diagonal entries are computed last to ensure that $R_t(z_t, z_t) = -\sum_{z_{t+dt} \neq z_t} R_t(z_t, z_{t+dt})$.

We incorporated various types of priors into $R^{\text{DB}}$ by preserving specific rows or entries in the matrix. Specifically, we experimented with retaining the column corresponding to the state with the highest marginal distribution (Column - Max Marginal), the column corresponding to the predicted $x_1$ states (Column - $x_1$), and the columns corresponding to the state with the highest probability in $p_{t|1}$. Additionally, we tested the approach of retaining only $R^{\text{DB}}(x_t, i)$ where $i$ is the state with the highest marginal distribution (Entry - Max Marginal). For instance, under the absorbing initial distribution, this state is the one to which all data is absorbed at $t = 0$. We note that there remains significant space for exploration by adjusting the weights assigned to different positions within $R^{\text{DB}}$, as the only condition that must be satisfied is that

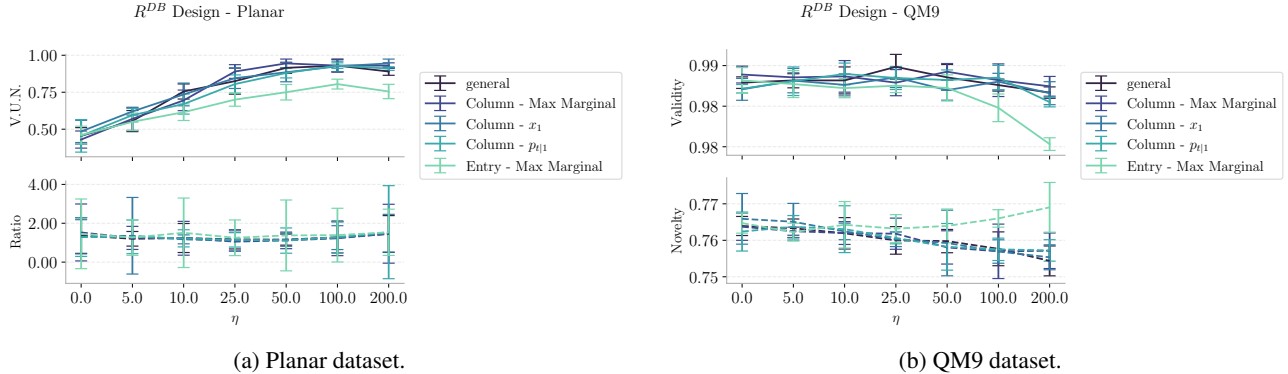

$$\begin{vmatrix} - & p_1 & p_2 \\ p_0 & - & p_2 \\ p_0 & p_1 & - \end{vmatrix} \qquad \begin{vmatrix} - & p_1 & 0 \\ p_0 & - & p_2 \\ 0 & p_1 & - \end{vmatrix} \qquad \begin{vmatrix} - & 0 & 0 \\ 0 & - & p_2 \\ 0 & p_1 & - \end{vmatrix} \qquad \begin{vmatrix} - & 0 & 0 \\ 0 & - & 0 \\ 0 & 0 & - \end{vmatrix}$$

General case          Column case          Single entry case          No stochasticity

Full matrix          Preserve one column/row          Preserve two entries          All 0          $R^{DB}$ sparsity

$$\begin{vmatrix} - & 0.7 & 0.1 \\ 0.2 & - & 0.1 \\ 0.2 & 0.7 & - \end{vmatrix} \qquad \begin{vmatrix} - & 0.7 & 0 \\ 0.2 & - & 0.1 \\ 0 & 0.7 & - \end{vmatrix} \qquad \begin{vmatrix} - & 0 & 0 \\ 0 & - & 0.1 \\ 0 & 0.7 & - \end{vmatrix} \qquad \begin{vmatrix} - & 0 & 0 \\ 0 & - & 0 \\ 0 & 0 & - \end{vmatrix}$$

Space of exploration

Figure 5: Examples of different rate matrices from the space of $3{\times}3$ matrices that satisfy the detailed balance condition. Here $p_i$ denotes $p_{t|1}(i|z_1)$.

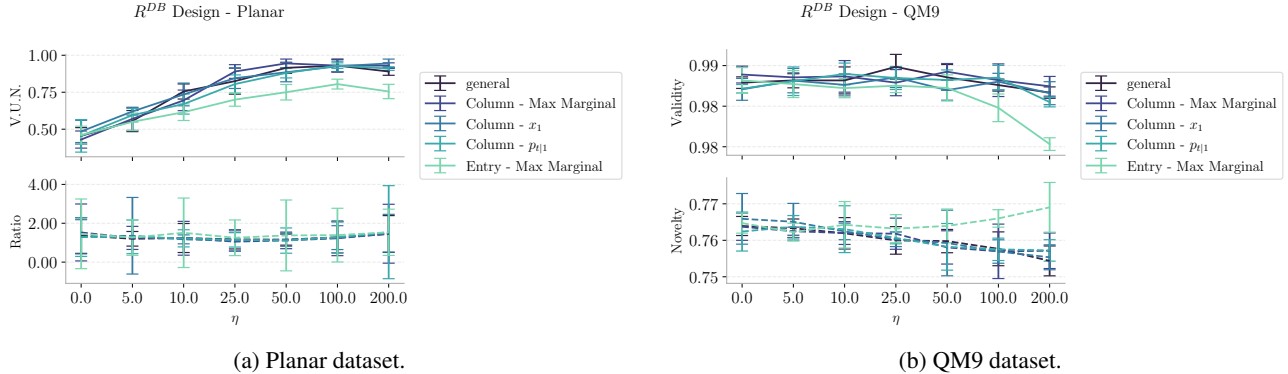

(a) Planar dataset.          (b) QM9 dataset.

Figure 6: Impact of $R^{\mathrm{DB}}$ with different level of sparsity.

symmetrical positions adhere to a specific proportionality. However, in practice, none of the specific designs illustrated in Figure 5 showed a clear advantage over others in the settings we evaluated. As a result, we chose the general case for our experiments, as it offers the most flexibility by incorporating the least prior knowledge.

Orthogonal to the design of $R_t^{\mathrm{DB}}$, we must also consider the hyperparameter $\eta$, which regulates the magnitude of stochasticity in the denoising process. Specifically, setting $\eta = 0$ (thereby relying solely on $R_t^*$) minimizes the expected number of jumps throughout the denoising trajectory under certain conditions, as shown by Campbell et al. (2024) in Proposition 3.4. However, in continuous diffusion models, some level of stochasticity has been demonstrated to enhance performance (Karras et al., 2022; Cao et al., 2023; Xu et al., 2023). Conversely, excessive stochasticity can negatively impact performance. Campbell et al. (2024) propose that there exists an optimal level of stochasticity that strikes a balance between exploration and accuracy. In our experiments, we observed varied behaviors as $\eta$ increases, resulting in different performance outcomes across datasets, as illustrated in Figure 10.

### B.4. Hyperparameter Optimization Pipeline

A significant advantage of flow matching methods is their inherently greater flexibility in the sampling process compared to diffusion models, as they are more disentangled from the training stage. Each of the proposed optimization strategies exposed in Sec. 3.2 expands the search space for optimal performance. However, conducting a full grid search across all those methodologies is impractical for the computational resources available. To address this challenge, our sampling optimization pipeline consists of, for each of the proposed optimization strategies, all hyperparameters are held constant at their default values except for the parameter controlling the chosen strategy, over which we perform a sweep. The optimal values obtained for each strategy are combined to form the final configuration. In Tab. 5, we present the final hyperparameter values obtained for each dataset. This pipeline is sufficient to achieve state-of-the-art performance, which reinforces the expressivity of DeFoG. We expect to achieve even better results if a more comprehensive search of the hyperparameter space was carried out.

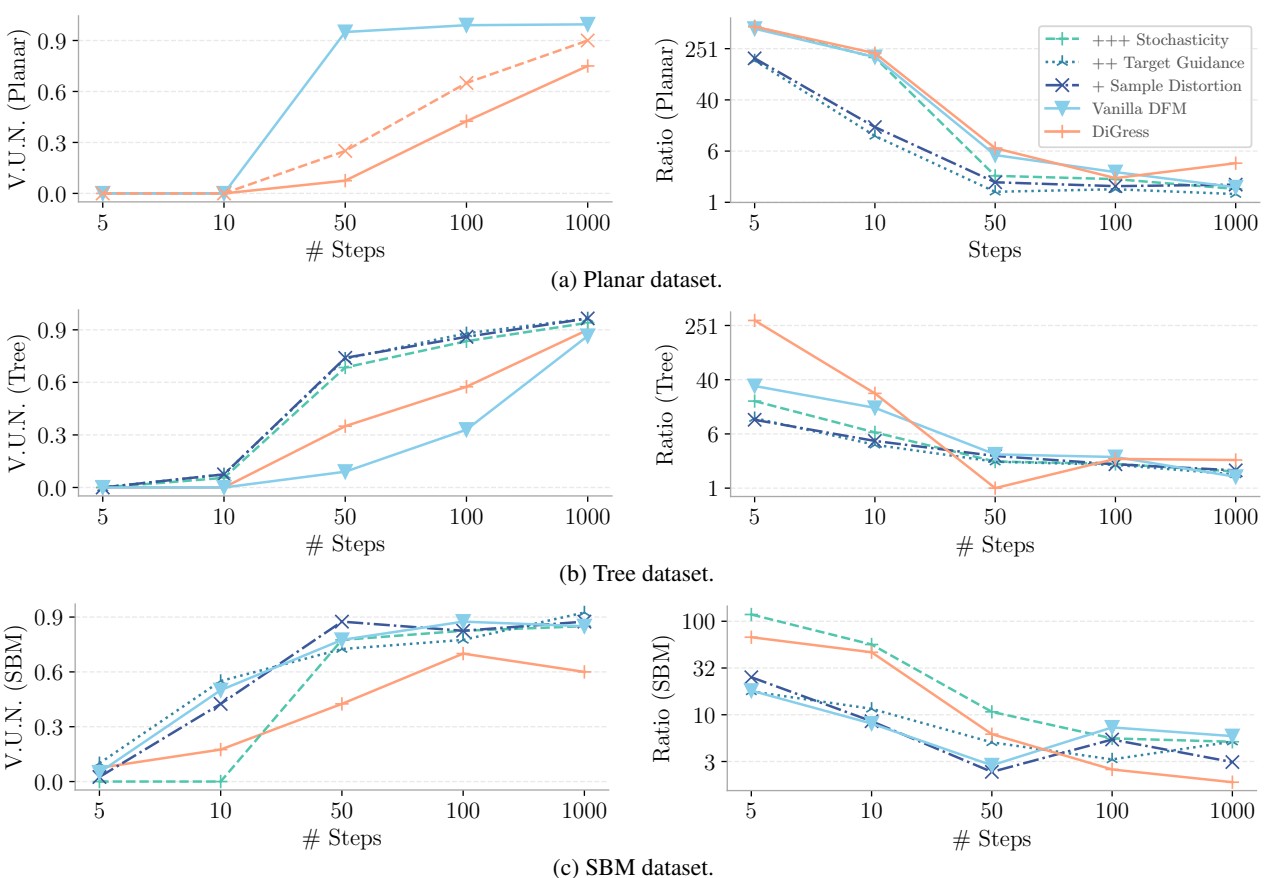

(a) Planar dataset.

(b) Tree dataset.

(c) SBM dataset.

Figure 7: Sampling efficiency improvement over all synthetic datasets.

To better mode detailedly illustrate the influence of each sampling optimization, we show in Figures 7 and 8 in present the impact of varying parameter values across the synthetic datasets Figure 7 and molecular datasets in Figure 8 used in this work.

While Figures 7 and 8 are highly condensed, we provide a more fine-grained version that specifically illustrates the influence of the hyperparameters $\eta$ and $\omega$. This version highlights their impact when generating with the full number of steps (500 and 1000 for molecular and synthetic data, respectively) and with 50 steps. As emphasized in Figures 9 and 10, the influence of these hyperparameters varies across datasets and exhibits distinct behaviors depending on the number of steps used.

Several key observations can be made here. First, since the stochasticity is designed around the detailed balance condition, which holds more rigorously with increased precision, it generally provides greater benefits with the full generation steps but leads to a more pronounced performance decrease when generating with only 50 steps. Additionally, for datasets such as Planar, MOSES, and Guacamol, the stochasticity shows an increasing-then-decreasing behavior, indicating the presence of an optimal value. Furthermore, while target guidance significantly improves validity across different datasets, it can negatively affect novelty and the average ratio when set too high. This suggests that excessive target guidance may promote overfitting to high-probability regions of the training set, distorting the overall distribution. In conclusion, each hyperparameter should be carefully chosen based on the specific objective.

To demonstrate the benefit of each designed optimization step, we report the step-wise improvements by sequentially adding each tuned step across the primary datasets — synthetic datasets in Figure 11 and molecular datasets in Figure 12 — used in this work.

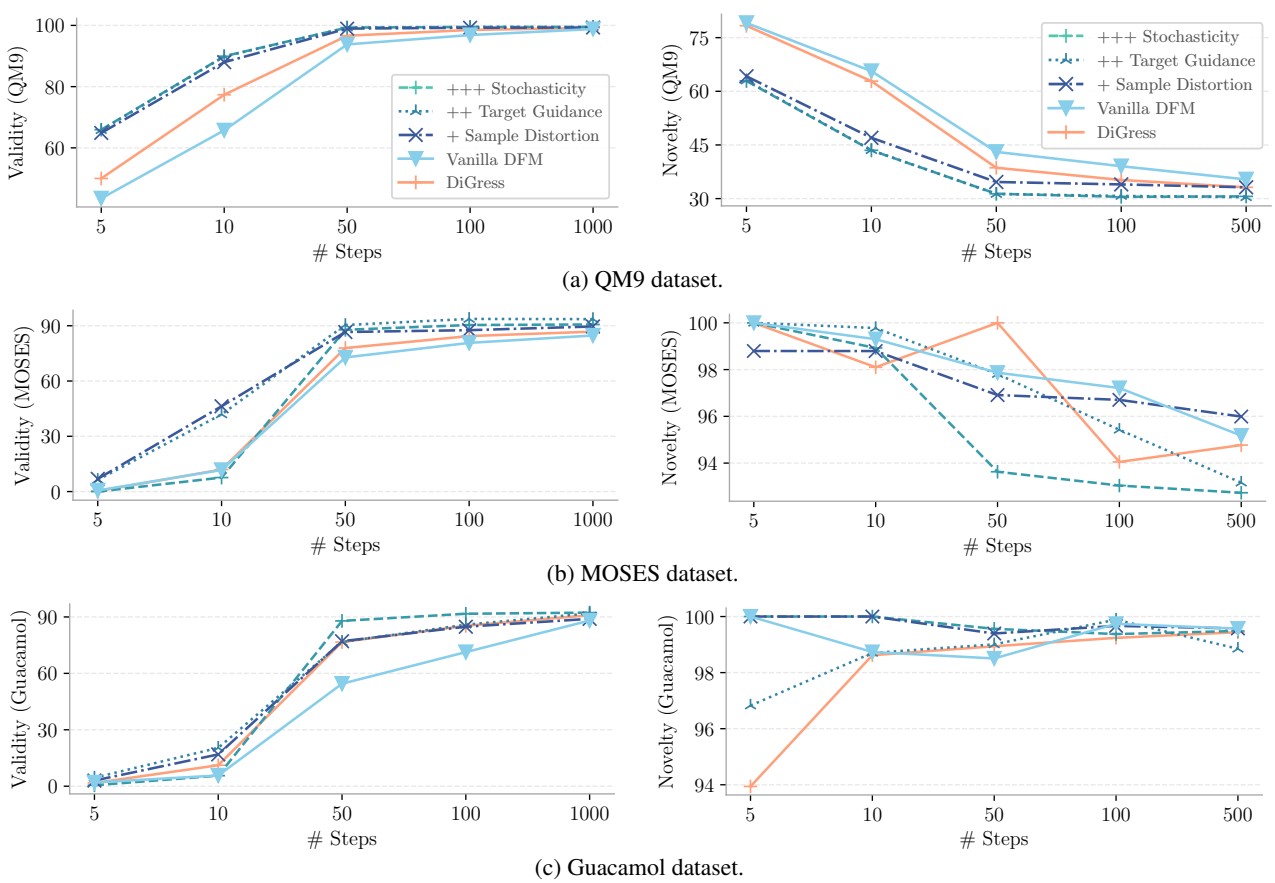

(a) QM9 dataset.

(b) MOSES dataset.

(c) Guacamol dataset.

Figure 8: Sampling efficiency improvement over all molecular datasets. The Guacamol and MOSES datasets are evaluated with 2,000 samples, and validity and novelty are computed using local implementations for efficiency instead of the original benchmarks. For the Guacamol dataset, we apply validity metrics that account for charged molecules, reflecting the dataset's intrinsic characteristic of containing a significant number of such molecules.

### B.5. Understanding the Sampling Dynamics of $R_t^*$

In this section, we aim to provide deeper intuition into the sampling dynamics imposed by the design of $R_t^*$, as proposed by (Campbell et al., 2024). The explicit formulation of $R_t^*$ can be found in Eq. (3). Notably, the denominator in the expression serves as a normalizing factor, meaning the dynamics of each sampling step are primarily influenced by the values in the numerator. Specifically, we observe the following relationship:

$$\partial_t p_{t|1}(z_t|z_1) = \delta(z_t, z_1) + p_0(z_t),$$

derived by directly differentiating Eq. (1). Based on this, the possible values of $R_t^*$ for different combinations of $z_t$ and $z_{t+dt}$ are outlined in Tab. 3.

Table 3: Values of the numerator of $R_t^*$ for different $z_t$ and $z_{t+dt}$.

| CONDITION | NUMERATOR OF $R^*(x_t, j|z_1)$ | INTUITION |
|---|---|---|
| $z_t = z_1, z_{t+dt} = z_1$ | $\mathrm{ReLU}(p_0(z_1) - p_0(z_1)) = 0$ | NO TRANSITION |
| $z_t = z_1, z_{t+dt} \neq z_1$ | $\mathrm{ReLU}(p_0(z_t) - 1 - p_0(z_{t+dt})) = 0$ | NO TRANSITION |
| $z_t \neq z_1, z_{t+dt} = z_1$ | $\mathrm{ReLU}(p_0(z_t) - p_0(z_{t+dt}) + 1) > 0$ | TRANSITION TO $z_1$ |
| $z_t \neq z_1, z_{t+dt} \neq z_1$ | $\mathrm{ReLU}(p_0(z_t) - p_0(z_{t+dt}))$ | TRANSITION TO $z_{t+dt}$ IF $p_0(z_t) > p_0(z_{t+dt})$ |

From the first two lines of Tab. 3, we observe that once the system reaches the predicted state $z_1$, it remains there. If not, $R_t^*$

only encourages transitions to other states under two conditions: either the target state is $z_1$ (third line), or the corresponding entries in the initial distribution for potential next states have smaller values than the current state (fourth line). As a result, the sampling dynamics are heavily influenced by the initial distribution, as discussed further in Appendix C.1.

For instance, with the masking distribution, the fourth line facilitates transitions to states other than the virtual "mask" state, whereas for the uniform distribution, no transitions are allowed. For the marginal distribution, transitions are directed toward less likely states. Note that while these behaviors hold when the rate matrix consists solely of $R_t^*$, additional transitions can be introduced through $R_t^{\mathrm{DB}}$ (as detailed in Appendix B.3) or by applying target guidance (see Appendix B.2).

### B.6. Performance Improvement for Undertrained Models

In this section, we present the performance of a model trained on the QM9 dataset and the Planar dataset using only $30\%$ of the epochs compared to the final model being reported. We employ the same hyperparameters as in Tab. 8 and Tab. 1 for the sampling setup, as reported in Tab. 5.

Compared to fully trained models, our model achieves 99.0 Validity (*vs* 99.3) and 96.4 Uniqueness (*vs* 96.3) on the QM9 dataset. For the Planar dataset, it attains 95.5 Validity (*vs* 99.5) and an average ratio of 1.4 (*vs* 1.6). These results demonstrate that, even with significantly fewer training epochs, the model maintains competitive performance under a well-designed sampling procedure, although extended training can still further improve performance. Notably, all metrics surpass the discrete-time diffusion benchmark DiGress (Vignac et al., 2022). As a result, the optimization in the sampling stage proves particularly beneficial when computational resources are limited, by enhancing the performance of an undertrained model.

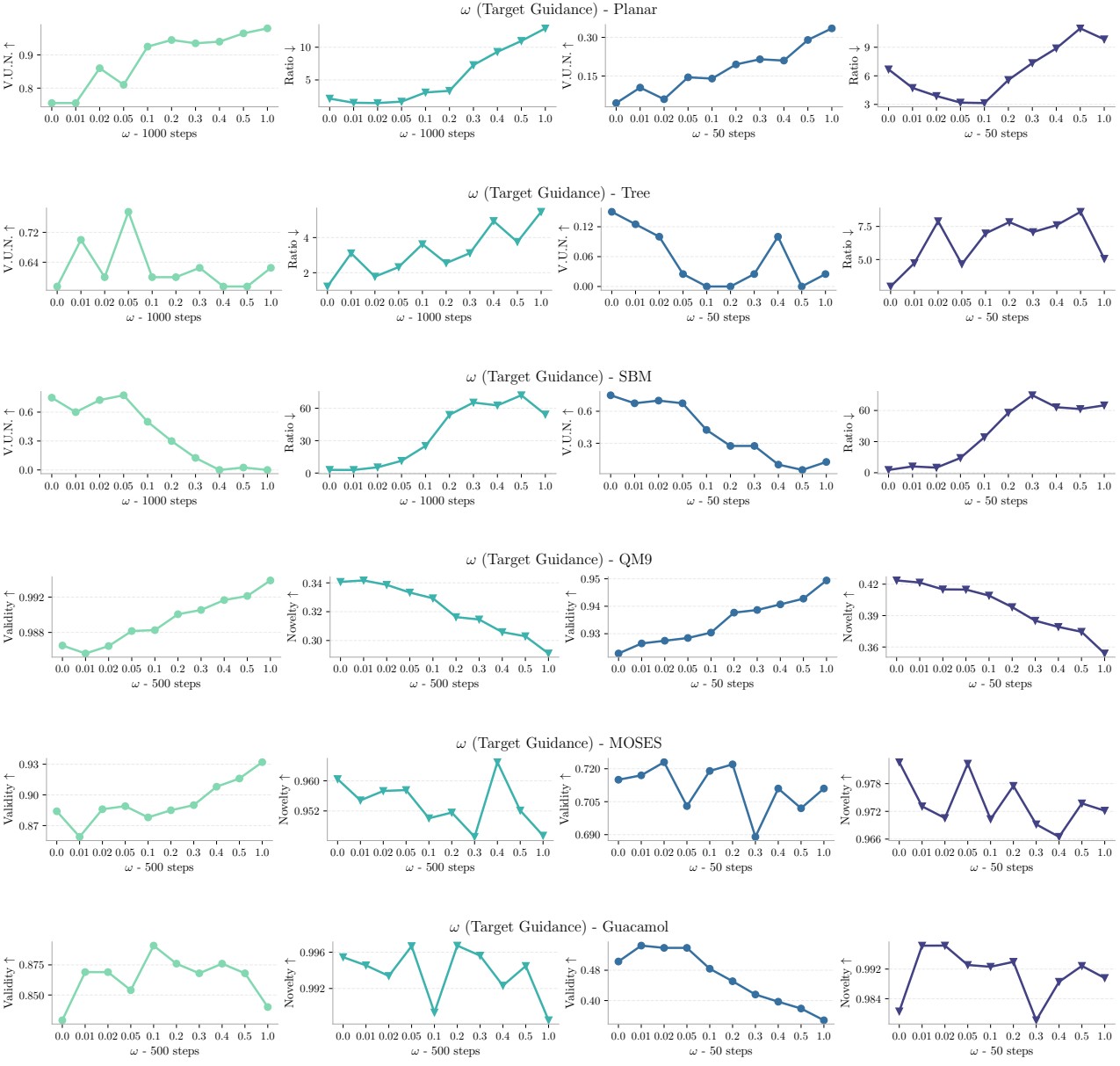

Figure 9: Influence of target guidance over all datasets.

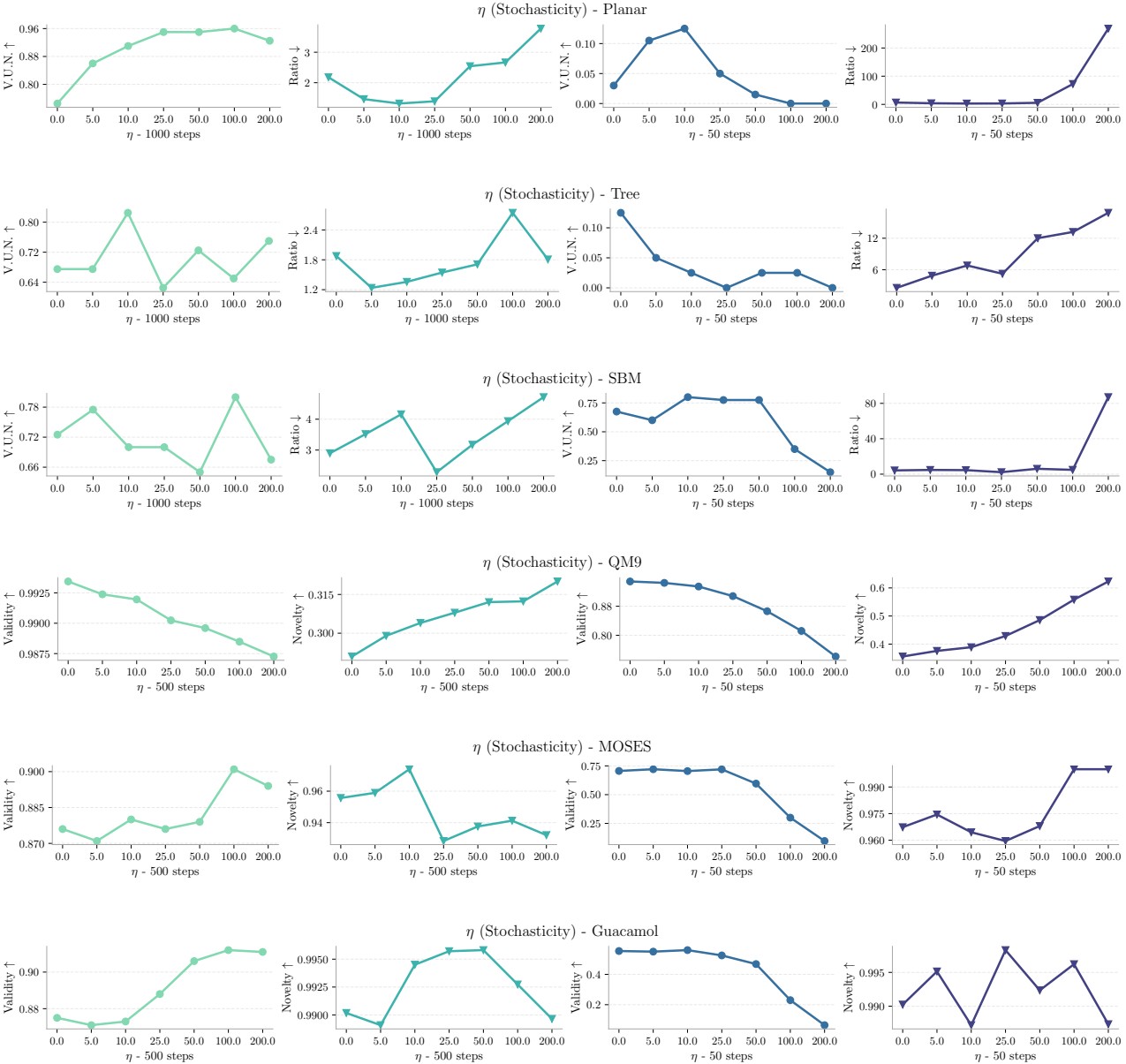

Figure 10: Influence of stochasticity level over all datasets.

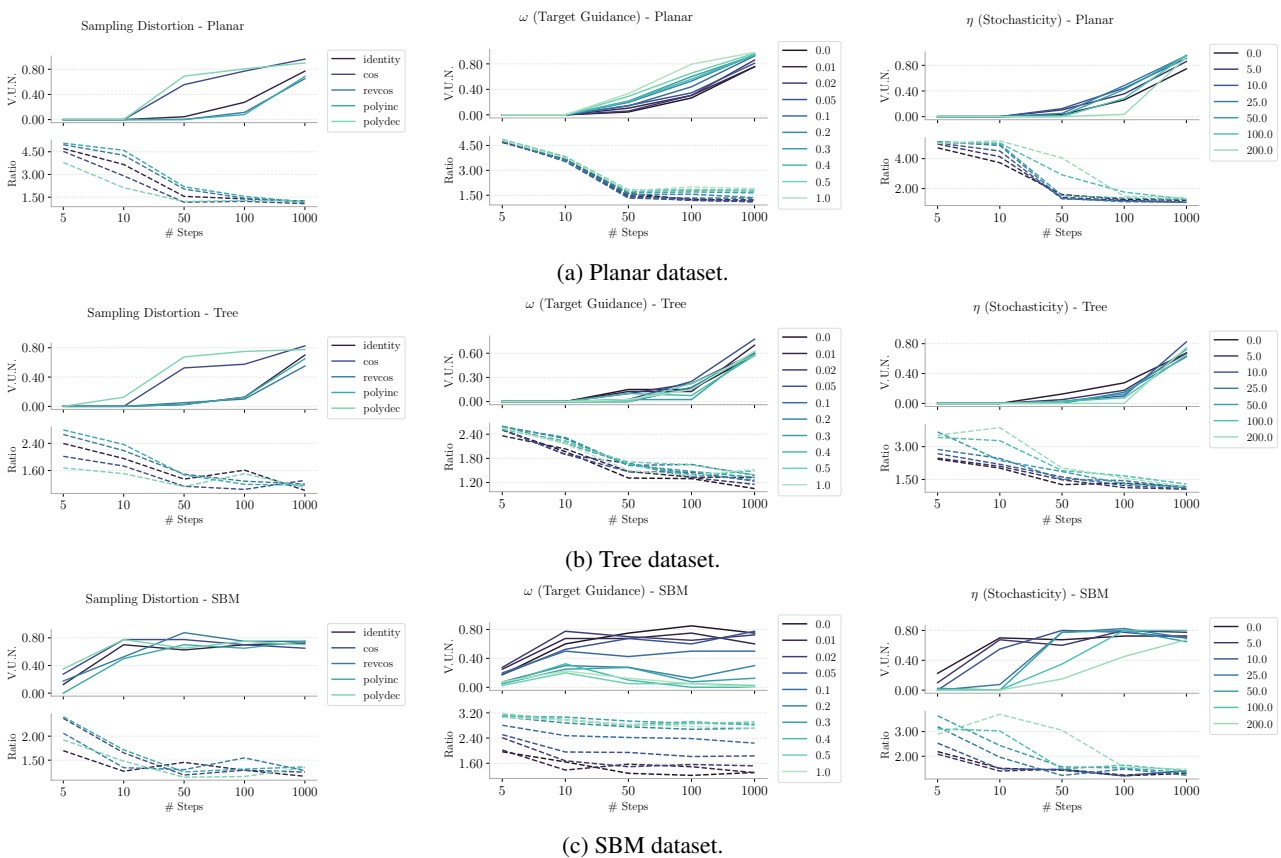

Figure 11: Stepwise parameter search for sampling optimization across synthetic datasets.

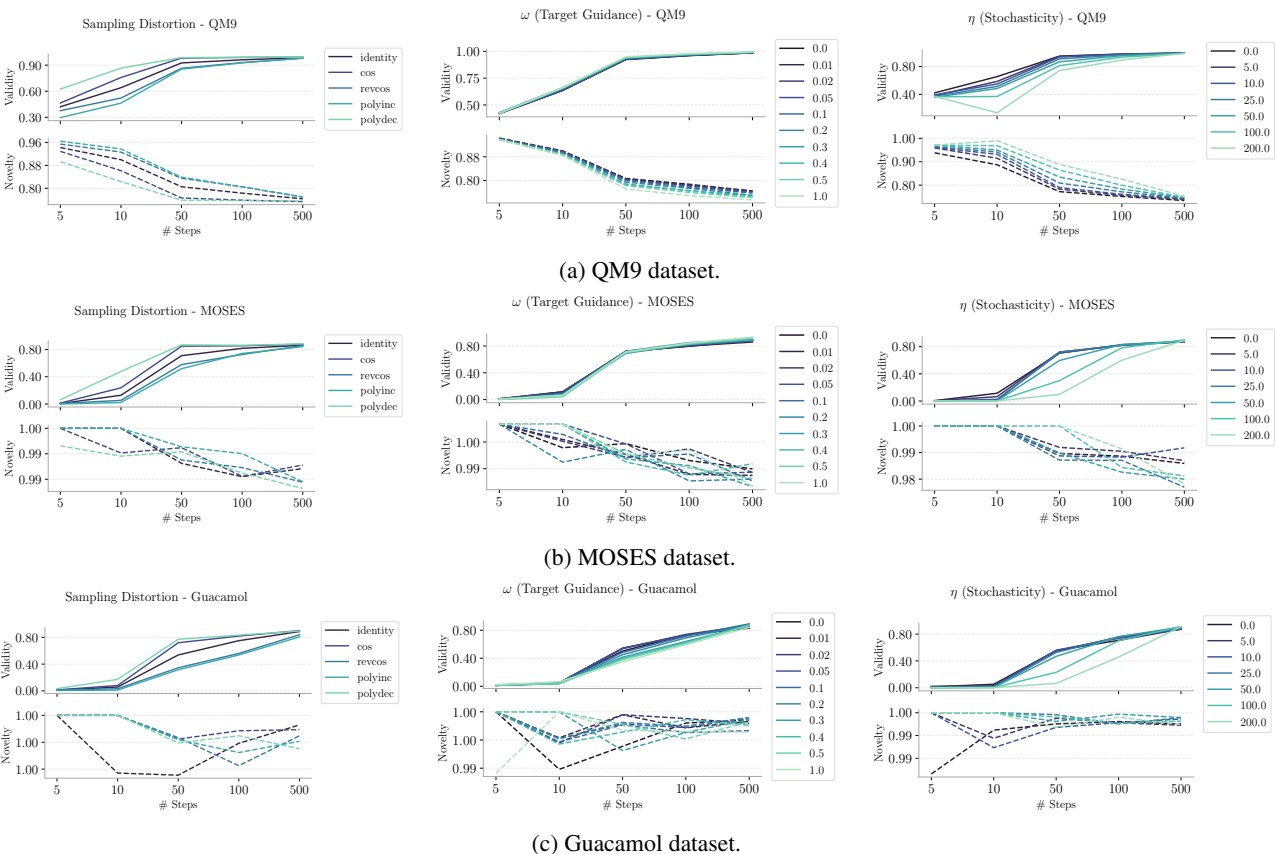

(a) QM9 dataset.

(b) MOSES dataset.

(c) Guacamol dataset.

Figure 12: Stepwise parameter search for sampling optimization across molecular datasets.

# C. Train Optimization

In this section, we provide a more detailed analysis of the influence of the various training optimization strategies introduced in Sec. 3. In Appendix C.1, we empirically demonstrate the impact of selecting different initial distributions on performance, while in Appendix C.2, we examine the interaction between training and sampling optimization.

## C.1. Initial Distributions

Under DeFoG's framework, the noising process for each dimension is modeled as a linear interpolation between the clean data distribution (the one-hot representation of the current state) and an initial distribution, $p_0$. As such, it is intuitive that different initial distributions result in varying performances, depending on the denoising dynamics they induce. In particular, they have a direct impact on the sampling dynamics through $R_t^*$ (see Appendix B.5) and may also pose tasks of varying difficulty for the graph transformer. In this paper, we explore four distinct initial distributions[2]:

**Uniform**: $p_0 = \left[\frac{1}{Z}, \frac{1}{Z}, \ldots, \frac{1}{Z}\right] \in \Delta^Z$. Here, the probability mass is uniformly distributed across all states, as proposed by (Campbell et al., 2024).

**Masking**: $p_0 = [0, 0, \ldots, 0, 1] \in \Delta^{Z+1}$. In this setting, all the probability mass collapses into a new "mask" state at $t = 0$, as introduced by (Campbell et al., 2024).

**Marginal**: $p_0 = [m_1, m_2, \ldots, m_Z] \in \Delta^Z$, where $m_i$ denotes the marginal probability of the $i$-th state in the dataset. This approach is widely used in state-of-the-art graph generation models (Vignac et al., 2022; Xu et al., 2024; Siraudin et al., 2024).

**Absorbing**: $p_0 = [0, \ldots, 1, \ldots, 0] \in \Delta^Z$, representing a one-hot encoding of the most common state (akin to applying an $\arg\max$ operator to the marginal initial distribution).

We apply each of these initial distributions to both node and edges, with the corresponding dimensionalities. In Figure 13, we present the training curves for each initial distribution for three different datasets.

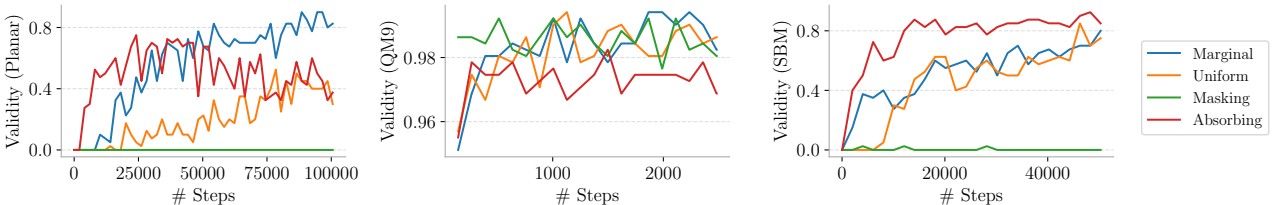

Figure 13: Influence of initial distribution over different datasets at different training steps.

We observe that the marginal distribution consistently achieves at least as good performance as the other initial distributions. This is supported by two key observations in previous work: first, Vignac et al. (2022) demonstrate that, for any given distribution, the closest distribution within the aforementioned class of factorizable initial distributions is the marginal one, thus illustrating its optimality as a prior. Second, the marginal initial distribution preserves the dataset's marginal properties throughout the noising process, maintaining graph-theoretical characteristics like sparsity (Qin et al., 2023). We conjecture that this fact facilitates the denoising task for the graph transformer. This reinforces its use as the default initial distribution for DeFoG. The only dataset where marginal was surpassed was the SBM dataset, which we attribute to its inherently different nature (stochastic, instead of deterministic). In this case, the absorbing distribution emerged as the best-performing choice. Interestingly, the absorbing distribution tends to converge faster across datasets.

Lastly, it is worth noting that in discrete diffusion models for graphs, predicting the best limit noise distribution based solely on dataset characteristics remains, to our knowledge, an open question (Tseng et al., 2023). We expect this complexity to extend to discrete flow models as well. Although this is outside the scope of our work, we view this as an exciting direction for future research.

---

[2] Recall that $Z$ represents the cardinality of the state space, and $\Delta^Z$ the associated probability simplex.

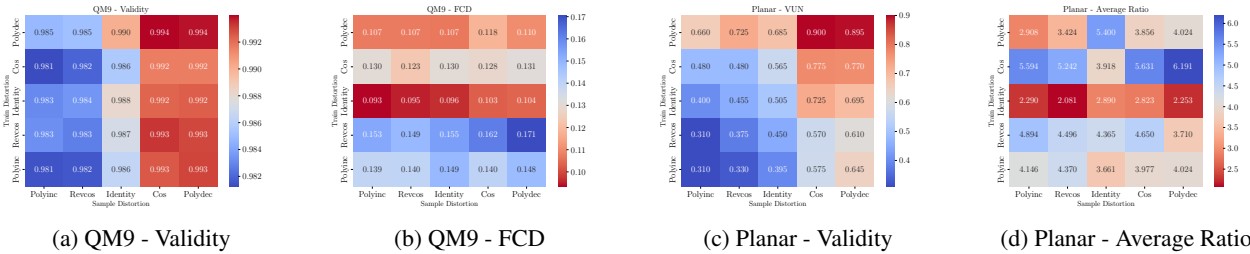

| (a) QM9 - Validity | (b) QM9 - FCD | (c) Planar - Validity | (d) Planar - Average Ratio |

Figure 14: Interaction between training and sampling time distortions.

## C.2. Interaction between Sample and Train Distortions

From Appendix B.4, we observe that time distortions applied during the sampling stage can significantly affect performance. This suggests that graph discrete flow models do not behave evenly across time and are more sensitive to specific time intervals, where generative performance benefits from finer updates achieved by using smaller time steps. Building on this observation, we extended our analysis to the training stage, exploring two main questions:

- Is there a universally optimal training time distortion for graph generation across different datasets?

- How do training and sampling time distortions interact? Is there alignment between the two? Specifically, if we understand the effect of a time distortion at one stage (training or sampling), can we infer its impact at the other?

To investigate these questions, we conducted a grid search. For two datasets, we trained five models, each with a different time distortion applied during training. Subsequently, we tested each model by applying the five different distortions at the sampling stage. The results are presented in Figure 14.

The results vary by dataset. For QM9, validity appears primarily influenced by the sampling distortion method, with a preference for distortions that encourage smaller steps at the end of the denoising process (such as *polydec* and *cos*). However, for FCD[3], the training distortion plays a more significant role.

For the planar dataset, we observe a near-perfect alignment between training and sampling distortions in terms of validity, with a clear preference for more accurate training models and finer sampling predictions closer to $t = 1$. The results for the average ratio metric, however, are less consistent and show volatility.

These findings help address our core questions: The interaction between training and sampling distortions, as well as the best training time distortion, is dataset-dependent. Nonetheless, for the particular case of the planar dataset, we observe a notable alignment between training and sampling distortions. This alignment suggests that times close to $t = 1$ which are critical for correctly generating planar graphs. We conjecture that this alignment can be attributed to planarity being a global property that arises from local constraints, as captured by Kuratowski's Theorem, which states that a graph is non-planar if and only if it contains a subgraph reducible to $K_5$ or $K_{3,3}$ through edge contraction (Kuratowski, 1930).

**Loss Tracker** To determine if the structural properties observed in datasets like the planar dataset can be detected and exploited without requiring an exhaustive sweep over all possibilities, we propose developing a metric that quantifies the difficulty of predicting the clean graph for any given time point $t \in [0, 1)$. For this, we perform a sweep over $t$ for a given model, where for each $t$, we noise a sufficiently large batch of clean graphs and evaluate the model's training loss on them. This yields a curve that shows how the training loss varies as a function of $t$. We then track how this curve evolves across epochs. To make the changes more explicit, we compute the ratio of the loss curve relative to the fully trained model's values. These curves are shown in Figure 15.

As expected, the curve of training loss as a function of $t$ (left in Figure 15) is monotonically decreasing, indicating that as graphs are decreasingly noised, the task becomes simpler. However, the most interesting insights arise from the evolution of this curve across epochs (right in Figure 15). We observe that for smaller values of $t$, the model reaches its maximum capacity early in the training process, showing no significant improvements after the initial few epochs. In contrast, for

---

[3]FCD is calculated only for valid molecules, so this metric may inherently reflect survival bias.

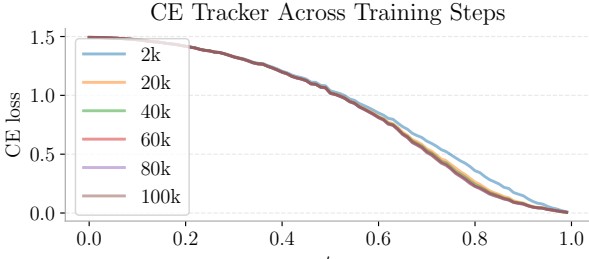 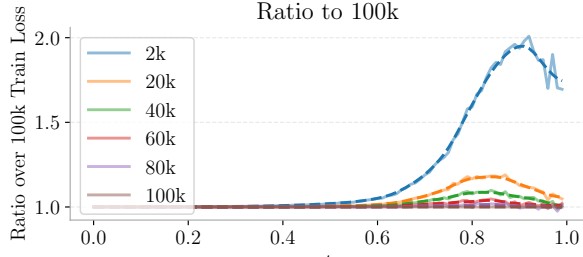

Figure 15: In the left figure, we present the cross-entropy (CE) loss used for training at different time steps across various stages of the training process. The right figure shows the ratio of each CE loss trajectory relative to the last one, illustrating the overall training trend and emphasizing which parts of the model are predominantly learned over time.

larger values of $t$ (closer to $t = 1$), the model exhibits substantial improvements throughout the training period. This suggests that the model can continue to refine its predictions in the time range where the task is easier. These findings align with those in Figure 14, reinforcing our expectation that training the model to be more precise in this range or providing more refined sampling steps will naturally enhance performance in the planar dataset.

These insights offer a valuable understanding of the specific dynamics at play within the planar dataset. Nevertheless, the unique structural characteristics of each dataset may influence the interaction between training and sampling time distortions in ways that are not captured here. Future work could explore these dynamics across a wider range of datasets to assess the generalizability of our findings.

# D. Theoretical Results

In this section, we provide the proofs of the different theoretical results of the paper. First, we provide results that are domain agnostic, i.e., that hold for general discrete data, and then instantiate them for the specific case of graphs, yielding Theorems 1 and 2. Then, we proceed to the permutation invariance/equivariance guarantees of DeFoG, and some remarks on the expressivity of the RRWP additional features.

## D.1. Theoretical Results on the Generative Framework

In Appendix D.1.1, we provide a proof of the bounded estimation error of the rate matrix for general discrete data, and instantiate it for graphs, yielding Theorem 1. Then, in Appendix D.1.2, we prove the bounded deviation of the generated distribution for general discrete data as well, and instantiate it for graphs, yielding Theorem 2.

Importantly, for general discrete data, we jointly model $D$ discrete variables, $(z^{(1)}, \ldots, z^{(D)}) = z^{1:D} \in \mathcal{Z}^D$.

### D.1.1. BOUNDED ESTIMATION ERROR OF UNCONDITIONAL MULTIVARIATE RATE MATRIX

We start by introducing two important concepts, which will reveal useful for the proof of the intended result.

**Unconditional Multivariate Rate Matrix**    As exposed in Sec. 2, the marginal distribution and the rate matrix of a CTMC are related by the Kolmogorov equation:

$$\partial_t p_t = R_t^T p_t$$
$$= \underbrace{\sum_{z_{t+\mathrm{d}t} \neq z_t} R_t(z_{t+\mathrm{d}t}, z_t) p_t(z_{t+\mathrm{d}t})}_{\text{Probability Inflow}} - \underbrace{\sum_{z_{t+\mathrm{d}t} \neq z_t} R_t(z_t, z_{t+\mathrm{d}t}) p_t(z_t)}_{\text{Probability Outflow}}.$$

The expansion in the second equality reveals the conservation law inherent in the Kolmogorov equation, illustrating that the time derivative of the marginal distribution represents the net balance between the inflow and outflow of probability mass at a given state.

Importantly, in the multivariate case, the (joint) rate matrix can be expressed through the following decomposition:

$$R_t(z_t^{1:D}, z_{t+\mathrm{d}t}^{1:D}) = \sum_{d=1}^{D} \delta(z_t^{1:D \setminus (d)}, z_{t+\mathrm{d}t}^{1:D \setminus (d)}) \, R_t^{(d)}(z_t^{1:D}, z_{t+\mathrm{d}t}^{(d)}) \tag{10}$$

$$= \sum_{d=1}^{D} \delta(z_t^{1:D \setminus (d)}, z_{t+\mathrm{d}t}^{1:D \setminus (d)}) \, \mathbb{E}_{p_{1|t}(z_1^{(d)} | z_t^{1:D})} \left[ R_t^{*(d)}(z_t^{(d)}, z_{t+\mathrm{d}t}^{(d)} | z_1^{(d)}) \right]. \tag{11}$$

In the first equality, $1 : D \setminus (d)$ refers to all dimensions except $d$ and the $\delta$ term restricts contributions to rate matrices that account for at most one dimension transitioning at a time, since the probability of two or more independently noised dimensions transitioning simultaneously is zero under a continuous time framework (Campbell et al., 2022; 2024). In the second equality, the unconditional rate matrix is retrieved by taking the expectation over the $z_1$-conditioned rate matrices. Specifically, $R_t^{*(d)}(z_t^{(d)}, z_{t+\mathrm{d}t}^{(d)} | z_1^{(d)})$ denotes the univariate rate matrix corresponding to dimension $d$ (see Eq. (3))

**Total Variation**    The total variation (TV) distance is a distance measure between probability distributions. While it can be defined more generally, this paper focuses on its application to discrete probability distributions over a finite sample space $\mathcal{Z}$. In particular, for two discrete probability distributions $P$ and $Q$, their total variation distance is defined as:

$$\|P - Q\|_{\mathrm{TV}} = \frac{1}{2} \sum_{z \in \mathcal{Z}} |P(z) - Q(z)| \tag{12}$$

We are now prepared to proceed with the proof of Theorem 11.

**Theorem 11** (Bounded estimation error of unconditional multivariate rate matrix). *Given $t \in [0, 1]$, $z_t^{1:D}, z_{t+\mathrm{d}t}^{1:D} \in \mathcal{Z}^D$, and $z_1^{1:D} \sim p_1(z_1^{1:D})$, let $R_t(z_t^{1:D}, z_{t+\mathrm{d}t}^{1:D})$ be the groundtruth rate matrix of the CTMC, which we approximate with*

$R_t^\theta(z_t^{1:D}, z_{t+\mathrm{dt}}^{1:D})$. *The corresponding estimation error is upper-bounded as follows:*

$$|R_t(z_t^{1:D}, z_{t+\mathrm{dt}}^{1:D}) - R_t^\theta(z_t^{1:D}, z_{t+\mathrm{dt}}^{1:D})|^2 \leq C_0 + C_1 \mathbb{E}_{p_1(z_1^{1:D})} \left[ p_{t|1}(z_t^{1:D}|z_1^{1:D}) \sum_{d=1}^D - \log p_{1|t}^{\theta,(d)}(z_1^{(d)}|z_t^{1:D}) \right], \quad (13)$$

*where*

- $C_0 = 2D \sup_{d \in \{1,\ldots,D\}} \{C_{z^{(d)}}^2\} \sum_{z_1^{(d)} \in \mathcal{Z}} p_{1|t}^{(d)}(z_1^{(d)}|z_t^{1:D}) \log p_{1|t}^{(d)}(z_1^{(d)}|z_t^{1:D})$;

- $C_1 = 2D \sup_{d \in \{1,\ldots,D\}} \{C_{z^{(d)}}^2\}/p_1(z_1^{1:D})$;

*with* $C_{z^{(d)}} = \delta(z_t^{1:D\backslash(d)}, z_{t+\mathrm{dt}}^{1:D\backslash(d)}) \sup_{z_1^{(d)} \in \mathcal{Z}} \{R_t^{(d)}(z_t^{(d)}, z_{t+\mathrm{dt}}^{(d)}|z_1^{(d)})\}$.

*Proof.* This proof is an adaptation of the proof of Theorem 3.3 from Xu et al. (2024) to the discrete flow matching setting. By definition (Eq. (11)), we have:

$$\begin{aligned}
R_t(z_t^{1:D}, z_{t+\mathrm{dt}}^{1:D}) &= \sum_{d=1}^D \delta(z_t^{1:D\backslash(d)}, z_{t+\mathrm{dt}}^{1:D\backslash(d)}) \, R_t^{(d)}(z_t^{1:D}, z_{t+\mathrm{dt}}^{(d)}) \\
&= \sum_{d=1}^D \delta(z_t^{1:D\backslash(d)}, z_{t+\mathrm{dt}}^{1:D\backslash(d)}) \, \mathbb{E}_{p_{1|t}^{(d)}(z_1^{(d)}|z_t^{1:D})} \left[ R_t^{(d)}(z_t^{(d)}, z_{t+\mathrm{dt}}^{(d)}|z_1^{(d)}) \right] \\
&= \sum_{d=1}^D \delta(z_t^{1:D\backslash(d)}, z_{t+\mathrm{dt}}^{1:D\backslash(d)}) \sum_{z_1^{(d)}} p_{1|t}^{(d)}(z_1^{(d)}|z_t^{1:D}) R_t^{(d)}(z_t^{(d)}, z_{t+\mathrm{dt}}^{(d)}|z_1^{(d)})
\end{aligned}$$

Thus:

$$\begin{aligned}
|R_t(z_t^{1:D}, &z_{t+dt}^{1:D}) - R_t^\theta(z_t^{1:D}, z_{t+dt}^{1:D})| = \\
&= \left| \sum_{d=1}^D \delta(z_t^{1:D\backslash(d)}, z_{t+dt}^{1:D\backslash(d)}) \sum_{z_1^{(d)}} [R_t^{(d)}(z_t^{(d)}, z_{t+dt}^{(d)}|z_1^{(d)}) \left( p_{1|t}^{(d)}(z_1^{(d)}|z_t^{1:D}) - p_{1|t}^{\theta,(d)}(z_1^{(d)}|z_t^{1:D}) \right)] \right| \\
&\leq \sum_{d=1}^D \delta(z_t^{1:D\backslash(d)}, z_{t+dt}^{1:D\backslash(d)}) \left| \sum_{z_1^{(d)}} \left[ R_t^{(d)}(z_t^{(d)}, z_{t+dt}^{(d)}|z_1^{(d)}) \left( p_{1|t}^{(d)}(z_1^{(d)}|z_t^{1:D}) - p_{1|t}^{\theta,(d)}(z_1^{(d)}|z_t^{1:D}) \right) \right] \right| \\
&\leq \sum_{d=1}^D \delta(z_t^{1:D\backslash(d)}, z_{t+dt}^{1:D\backslash(d)}) \sup_{z_1^{(d)}} \{R_t^{(d)}(z_t^{(d)}, z_{t+dt}^{(d)}|z_1^{(d)})\} \sum_{z_1^{(d)}} \left| p_{1|t}^{(d)}(z_1^{(d)}|z_t^{1:D}) - p_{1|t}^{\theta,(d)}(z_1^{(d)}|z_t^{1:D}) \right| \\
&= \sum_{d=1}^D 2\, C_{z^{(d)}} \|p_{1|t}^{(d)}(z_1^{(d)}|z_t^{1:D}) - p_{1|t}^{\theta,(d)}(z_1^{(d)}|z_t^{1:D})\|_{\mathrm{TV}} \qquad (14) \\
&\leq \sum_{d=1}^D C_{z^{(d)}} \sqrt{2 D_{\mathrm{KL}} \left( p_{1|t}^{(d)}(z_1^{(d)}|z_t^{1:D}) \| p_{1|t}^{\theta,(d)}(z_1^{(d)}|z_t^{1:D}) \right)} \qquad (15) \\
&= \sum_{d=1}^D \sqrt{2\, C_{z^{(d)}}^2 \sum_{z_1^{(d)}} p_{1|t}^{(d)}(z_1^{(d)}|z_t^{1:D}) \log \frac{p^{(d)}(_{1|t}z_1^{(d)}|z_t^{1:D})}{p_{1|t}^{\theta,(d)}(z_1^{(d)}|z_t^{1:D})}}.
\end{aligned}$$

In Eq. (14), we use the definition of TV distance as defined in Eq. (12) and Eq. (15) results from direct application of Pinsker's inequality. Now, we change the ordering of the sum and of the square root through the Cauchy-Schwarz inequality: $\sum_{d=1}^{D} \sqrt{x_d} \leq \sum_{d=1}^{D} \sqrt{x_d} \cdot 1 \leq \sqrt{\sum_{d=1}^{D} \sqrt{x_d}^2} \sqrt{\sum_{d=1}^{D} \sqrt{1}^2} \leq \sqrt{D \sum_{d=1}^{D} x_d}$.

So, we obtain:

$$
|R_t(z_t^{1:D}, z_{t+dt}^{1:D}) - R_t^\theta(z_t^{1:D}, z_{t+dt}^{1:D})| \leq \sqrt{2\,D \sum_{d=1}^{D} C_{z^{(d)}}^2 \sum_{z_1^{(d)}} p_{1|t}^{(d)}(z_1^{(d)}|z_t^{1:D}) \log \frac{p_{1|t}^{(d)}(z_1^{(d)}|z_t^{1:D})}{p_{1|t}^{\theta,(d)}(z_1^{(d)}|z_t^{1:D})}}
$$

$$
\leq \sqrt{2\,D \sup_{d \in \{1,\dots,D\}} \{C_{z^{(d)}}^2\} \sum_{d=1}^{D} \sum_{z_1^{(d)}} p_{1|t}^{(d)}(z_1^{(d)}|z_t^{1:D}) \left[\log p_{1|t}^{(d)}(z_1^{(d)}|z_t^{1:D}) - \log p_{1|t}^{\theta,(d)}(z_1^{(d)}|z_t^{1:D})\right]}
$$

$$
= \sqrt{C_2 \left(C_3 - \underbrace{\sum_{d=1}^{D} \sum_{z_1^{(d)}} p_{1|t}^{(d)}(z_1^{(d)}|z_t^{1:D}) \log p_{1|t}^{\theta,(d)}(z_1^{(d)}|z_t^{1:D})}\right)},
$$

where in the last step we rearrange the terms independent of the approximation parametrized by $\theta$ as constants: $C_2 = 2D \sup_{d \in \{1,\dots,D\}} \{C_{z^{(d)}}^2\}$, $C_3 = \sum_{z_1^{(d)} \in \mathcal{Z}} p_{1|t}^{(d)}(z_1^{(d)}|z_t^{1:D}) \log p_{1|t}^{(d)}(z_1^{(d)}|z_t^{1:D})$.

We now develop the underbraced term in the last equation[4]:

$$
\sum_{d=1}^{D} \sum_{z_1^{(d)}} p_{1|t}^{(d)}(z_1^{(d)}|z_t^{1:D}) \log p_{1|t}^{\theta,(d)}(z_1^{(d)}|z_t^{1:D}) =
$$

$$
= \sum_{d=1}^{D} \sum_{z_1^{(d)}} \frac{p(z_1^{(d)}, z_t^{1:D})}{p_t(z_t^{1:D})} \log p_{1|t}^{\theta,(d)}(z_1^{(d)}|z_t^{1:D})
$$

$$
= \frac{1}{p_t(z_t^{1:D})} \sum_{d=1}^{D} \sum_{z_1^{(d)}} \sum_{z_1^{1:D\setminus(d)}} p(z_1^{(d)}, z_t^{1:D}, z_1^{1:D\setminus(d)}) \log p_{1|t}^{\theta,(d)}(z_1^{(d)}|z_t^{1:D})
$$

$$
= \frac{1}{p_t(z_t^{1:D})} \sum_{d=1}^{D} \sum_{z_1^{1:D}} p(z_1^{1:D}, z_t^{1:D}) \log p_{1|t}^{\theta,(d)}(z_1^{(d)}|z_t^{1:D})
$$

$$
= \frac{1}{p_t(z_t^{1:D})} \sum_{d=1}^{D} \sum_{z_1^{1:D}} p_1(z_1^{1:D}) p_{t|1}(z_t^{1:D}|z_1^{1:D}) \log p_{1|t}^{\theta,(d)}(z_1^{(d)}|z_t^{1:D})
$$

$$
= \frac{1}{p_t(z_t^{1:D})} \sum_{z_1^{1:D}} p_1(z_1^{1:D}) p_{t|1}(z_t^{1:D}|z_1^{1:D}) \sum_{d=1}^{D} \log p_{1|t}^{\theta,(d)}(z_1^{(d)}|z_t^{1:D})
$$

$$
= -C_4 \, \mathbb{E}_{p_1(z_1^{1:D})} \left[ p_{t|1}(z_t^{1:D}|z_1^{1:D}) \underbrace{\sum_{d=1}^{D} -\log p_{1|t}^{\theta,(d)}(z_1^{(d)}|z_t^{1:D})}_{\text{Cross-entropy}} \right],
$$

where $C_4 = 1/p(z_1^{1:D})$.

---

[4]In this step, we omit some subscripts from joint probability distributions as they are not defined in the main paper, but they can be inferred from context.

Replacing back the obtained expression into the original equation, we obtain:

$$\left| R_t(z_t^{1:D}, z_{t+dt}^{1:D}) - R_t^\theta(z_t^{1:D}, z_{t+dt}^{1:D}) \right|^2 \leq C_2 C_3 + C_2 C_4 \mathbb{E}_{p_1(z_1^{1:D})} \left[ p_{t|1}(z_t^{1:D}|z_1^{1:D}) \sum_{d=1}^{D} - \log p_{1|t}^{\theta,(d)}(z_1^{(d)}|z_t^{1:D}) \right],$$

retrieving the intended result. $\qquad\square$

**Corollary 1** (Bounded estimation error of unconditional rate matrix for graphs). *Given $t \in [0,1]$, graphs $G_t, G_{t+dt}$, and $G_1 \sim p_1(G_1)$, let $R_t(G_t, G_{t+dt})$ be the groundtruth rate matrix of the CTMC, which we approximate with $R_t^\theta(G_t, G_{t+dt})$. Then, there exist constants $\bar{C}_0, \bar{C}_1, \bar{C}_3$ such that:*

$$|R_t(G_t, G_{t+dt}) - R_t^\theta(G_t, G_{t+dt})|^2 \leq$$

$$\leq \bar{C}_0 + \bar{C}_1 \, \mathbb{E}_{p_1(G_1)} \left[ p_{t|1}(G_t|G_1) \sum_{n=1}^{N} - \log p_{1|t}^{\theta,(n)}(x_1^{(n)}|G_t) \right]$$

$$+ \bar{C}_2 \, \mathbb{E}_{p_1(G_1)} \left[ p_{t|1}(G_t|G_1) \sum_{1 \leq i < j \leq N} - \log p_{1|t}^{\theta,(ij)}(e_1^{(ij)}|G_t) \right].$$

*Proof.* Recall that a graph $G$ is defined as the set of nodes and edges $G = (x^{1:n:N}, e^{1:i<j:N})$, such that $x^{(n)} \in \mathcal{X} = \{1, \ldots, X\}$, $e^{(ij)} \in \mathcal{E} = \{1, \ldots, E\}$, so we can develop Eq. (11) as follows:

$$R_t(G_t, G_{t+dt}) = \underbrace{\sum_{n=1}^{N} \delta\left(G_t^{\backslash(n)}, G_{t+dt}^{\backslash(n)}\right) \mathbb{E}_{p_{1|t}(x_1^{(n)}|G_t)} \left[ R_t^{(n)}(x_t^{(n)}, x_{t+dt}^{(n)}|x_1^{(n)}) \right]}_{R_t(x_t^{1:n:N}, x_{t+dt}^{1:n:N})}$$

$$+ \underbrace{\sum_{1 \leq i < j \leq X} \delta\left(G_t^{\backslash(ij)}, G_{t+dt}^{\backslash(ij)}\right) \mathbb{E}_{p_{1|t}(e_1^{(ij)}|G_t)} \left[ R_t^{(ij)}(e_t^{(ij)}, e_{t+dt}^{(ij)}|e_1^{(ij)}) \right]}_{R_t(e_t^{1:i<j:N}, e_{t+dt}^{1:i<j:N})}, \qquad (16)$$

where $G^{\backslash(n)}$ and $G^{\backslash(ij)}$ denote the whole graph except the node $n$ and the edge connecting node $i$ to node $j$, respectively.

Therefore, the result from Theorem 11 implies:

$$|R_t(G_t, G_{t+dt}) - R_t^\theta(G_t, G_{t+dt})| \leq$$

$$\leq |R_t(x_t^{1:n:N}, x_{t+dt}^{1:n:N}) - R_t^\theta(x_t^{1:n:N}, x_{t+dt}^{1:n:N})| + |R_t(e_t^{1:i<j:N}, e_{t+dt}^{1:i<j:N}) - R_t^\theta(e_t^{1:i<j:N}, e_{t+dt}^{1:i<j:N})|$$

$$\leq \sqrt{C_0^{\mathcal{X}} + C_1^{\mathcal{X}} \, \mathbb{E}_{p_1(G_1)} \left[ p_{t|1}(G_t|G_1) \sum_{n=1}^{N} - \log p_{1|t}^{\theta,(n)}(x_1^{(n)}|G_t) \right]}$$

$$+ \sqrt{C_2^{\mathcal{E}} + C_3^{\mathcal{E}} \, \mathbb{E}_{p_1(G_1)} \left[ p_{t|1}(G_t|G_1) \sum_{1 \leq i < j \leq N} - \log p_{1|t}^{\theta,(ij)}(e_1^{(ij)}|G_t) \right]}$$

We now apply the identity $\sum_{d=1}^{D} \sqrt{x_d} \leq \sqrt{D \sum_{d=1}^{D} x_d}$, derived from the Cauchy-Schwarz inequality in the proof of Theorem 11, with $D = 2$ to obtain:

$$|R_t(G_t, G_{t+dt}) - R_t^\theta(G_t, G_{t+dt})| \leq$$

$$\leq \left( \bar{C}_0 + \bar{C}_1 \, \mathbb{E}_{p_1(G_1)} \left[ p_{t|1}(G_t|G_1) \sum_{n=1}^{N} - \log p_{1|t}^{\theta,(n)}(x_1^{(n)}|G_t) \right] \right.$$

$$\left. + \bar{C}_2 \, \mathbb{E}_{p_1(G_1)} \left[ p_{t|1}(G_t|G_1) \sum_{1 \leq i < j \leq N} - \log p_{1|t}^{\theta,(ij)}(e_1^{(ij)}|G_t) \right] \right)^{1/2},$$

with $\bar{C}_0 = 2(C_0^{\mathcal{X}} + C_2^{\mathcal{E}})$, $\bar{C}_1 = 2C_1^{\mathcal{X}}$, and $\bar{C}_2 = 2C_3^{\mathcal{E}}$, yielding the intended result. $\qquad\square$

D.1.2. BOUNDED DEVIATION OF THE GENERATED DISTRIBUTION

As in Appendix D.1.1, we start by introducing the necessary concepts that will reveal useful for the proof of the intended result.

**On the Choice of the CTMC Sampling Method**   Generating new samples using DFM amounts to simulate a multivariate CTMC according to:

$$p_{t+\mathrm{d}t|t}(z_{t+\mathrm{d}t}^{1:D}|z_t^{1:D}) = \delta(z_t^{1:D}, z_{t+\mathrm{d}t}^{1:D}) + R_t(z_t^{1:D}, z_{t+\mathrm{d}t}^{1:D})\mathrm{d}t, \tag{17}$$

where $R_t(z_t^{1:D}, z_{t+\mathrm{d}t}^{1:D})$ denotes the unconditional multivariate rate matrix defined in Eq. (11). This process can be simulated exactly using Gillespie's Algorithm (Gillespie, 1976; 1977). However, such an algorithm does not scale for large $D$ (Campbell et al., 2022). Although $\tau$-leaping is a widely adopted approximate algorithm to address this limitation (Gillespie, 2001), it requires ordinal discrete state spaces, which is suitable for cases like text or images but not for graphs. Therefore, we cannot apply it in the context of this paper. Additionally, directly replacing the infinitesimal step $\mathrm{d}t$ in Eq. (11) with a finite time step $\Delta t$ *à la* Euler method is inappropriate, as $R_t(z_t^{1:D}, z_{t+\mathrm{d}t}^{1:D})$ prevents state transitions in more than one dimension per step under the continuous framework. Instead, Campbell et al. (2024) propose an approximation where the Euler step is applied independently to each dimension, as seen in Eq. (4).

In this section, we theoretically demonstrate that, despite its approximation, the independent-dimensional Euler sampling method error remains bounded and can be made arbitrarily small by reducing the step size $\Delta t$ or by reducing the estimation error of the rate matrix.

**Markov Kernel of a CTMC**   For this proof, we also introduce the notion of *Markov kernel* of a CTMC. The Markov kernel, $\mathcal{R}_{t \to t+\Delta t}$, is a function that provides the transition probabilities between states over a *finite time interval*, $\Delta t$:

$$p_{t+\Delta t} = \mathcal{R}_{t \to t+\Delta t}^\top p_t. \tag{18}$$

For example, for a univariate CTMC with a state space $\mathcal{Z}$ of cardinality $Z$, the Markov kernel $\mathcal{R}_{t \to t+\Delta t}$ is a matrix where each entry $\mathcal{R}_{t \to t+\Delta t}^{(ij)}$ represents the probability that the single variable transitions from state $i$ at time $t$ to state $j$ at time $t + \Delta t$, i.e., $p_{t+\Delta t|t}(j|i)$.

The definition of Markov kernel contrasts with the one of *rate matrix* (or *generator*), $R_t$, which instead characterizes the *infinitesimal* transition rates between states at a given time $t$. Thus, while the rows of the rate matrix sum to 0, the Markov kernel matrices are stochastic, i.e., $\sum_{j \in Z} \mathcal{R}_{t \to t+\Delta t}^{(ij)} = 1, \ \forall i$. This contrasts with the rate matrix where rows sum to 1. Additionally, Markov kernels must also respect the initial condition $\mathcal{R}_{t \to t} = I$, where $I$ denotes the identity matrix.

Recall from Sec. 2 that the evolution of a CTMC is governed by the rate matrix through the equation:

$$\partial_t p_t = R^\top p_t,$$

which represents a first-order differential equation. Here, we focus on the time-homogeneous case, where $R_t = R$ in the time interval $[t; t + \Delta t]$, i.e., the rate matrix remains constant within the time interval. In that case, its solution is given by:

$$p_{t+\Delta t} = e^{R^\top \Delta t} p_t$$

Therefore the result above sets, by definition, the corresponding Markov kernel of a constant rate matrix in a finite time interval $\Delta t$ as:

$$\mathcal{R}_{t \to t+\Delta t} = (e^{R^\top \Delta t})^\top = e^{R \Delta t}, \tag{19}$$

where the second equality is a direct consequence of the definition of the matrix exponential as a series expansion.

We are now in conditions of proceeding to the proof of Theorem 8. We start by first proving that, in the univariate case, the time derivatives of the *conditional* rate matrices are upper bounded.

**Lemma 5** (Upper bound time derivative of conditional univariate rate matrix). *For $t \in (0,1)$, $z_t, z_{t+\mathrm{d}t}, z_1 \in \mathbb{Z}$, with $z_t \neq z_{t+\mathrm{d}t}$, then we have:*

$$|\partial_t R_t^*(z_t, z_{t+\mathrm{d}t}|z_1)| \leq \frac{2}{p_{t|1}(z_t|z_1)^2}.$$

*Proof.* Recall that $p_{t|1}(z_t|z_1) = t\,\delta(z_t, z_1) + (1-t)\,p_0(z_t)$ (from Eq. (1)). Two different cases must then be considered.

In the first case, $p_{t|1}(z_t|z_1) = 0$. This implies that both extremes of the linear interpolation are 0. In that case, the linear interpolation will be identically 0 for $t \in (0, 1)$. Thus, by definition, $R_t^*(z_t, z_{t+dt}|z_1) = 0$ for $t \in (0, 1)$, which implies that $|\partial_t R_t^*(z_t, z_{t+dt}|z_1)| = 0$.

Otherwise $(p_{t|1}(z_t|z_1) > 0)$, we recall that $R_t^*(z_t, z_{t+dt}|z_1)$ with $z_t \neq z_{t+dt}$ has the following form:

$$R_t^*(z_t, z_{t+dt}|z_1) = \frac{\text{ReLU}\left(\partial_t p_{t|1}(z_{t+dt}|z_1) - \partial_t p_{t|1}(z_t|z_1)\right)}{\mathbb{Z}_t^{>0} p_{t|1}(z_t|z_1)}, \tag{20}$$

where $\mathbb{Z}_t^{>0} = |\{z_t : p_{t|1}(z_t|z_1) > 0\}|$.

By differentiating the explicit form of $p_{t|1}(z_t|z_1)$, we have that $\partial^2 p_{t|1}(z_t|z_1) = 0$. As a consequence, the numerator of Eq. (20) has zero derivative. Additionally, we also note that $\mathbb{Z}_t^{>0}$ is constant. Again, since $p_{t|1}(z_t|z_1)$ is a linear interpolation between $z_1$ and $p_0$ and, therefore, it is impossible for $p_{t|1}(z_t|z_1)$ to suddenly become 0 for $t \in (0, 1)$.

Consequently, we have:

$$\partial_t R_t^*(z_t, z_{t+dt}|z_1) = \frac{\text{ReLU}\left(\partial_t p_{t|1}(z_{t+dt}|z_1) - \partial_t p_{t|1}(z_t|z_1)\right)}{\mathbb{Z}_t^{>0}} \partial_t \left(\frac{1}{p_{t|1}(z_t|z_1)}\right)$$

$$= -\frac{\text{ReLU}\left(\partial_t p_{t|1}(z_{t+dt}|z_1) - \partial_t p_{t|1}(z_t|z_1)\right)}{\mathbb{Z}_t^{>0}} \frac{\partial_t p_{t|1}(z_t|z_1)}{p_{t|1}(z_t|z_1)^2}.$$

We necessarily have $|\partial_t p_{t|1}(z_t|z_1)| = |\delta(z_t, z_1) - p_0(z_t)| \leq 1$, $\text{ReLU}\left(\partial_t p(z_{t+dt}|z_1) - \partial_t p(z_t|z_1)\right) \leq 2$, $\mathbb{Z}_t^{>0} \geq 1$, and, necessarily, $p(z_t|z_1) > 0$. Thus:

$$|\partial_t R_t^*(z_t, z_{t+dt}|z_1)| \leq \frac{\text{ReLU}\left(\partial_t p_{t|1}(z_{t+dt}|z_1) - \partial_t p_{t|1}(z_t|z_1)\right)}{\mathbb{Z}_t^{>0}} \frac{|\delta(z_t, z_1) - p_0(z_t)|}{p_{t|1}(z_t|z_1)^2}$$

$$\leq \frac{2}{p_{t|1}(z_t|z_1)^2}.$$

$\square$

We now upper bound the time derivative of the *unconditonal* multivariate rate matrix. We use Theorem 5 as an intermediate result to accomplish so. Additionally, we consider the following assumption.

**Assumption 6.** For $z_t^{1:D} \in \mathcal{Z}^D$, $z_1^{(d)} \in \mathcal{Z}$ and $t \in [0, 1]$, for each variable $z^{(d)}$ of a joint variable $z^{1:D}$, there exists a constant $B_t^{(d)} > 0$ such that $p_{1|t}(z_1^{(d)}|z_t^{1:D}) \leq B_t^{(d)} p_{t|1}(z_t^{(d)}|z_1^{(d)})^2$.

This assumption states that the denoising process is upper bounded by a quadratic term on the noising process. This assumption is reasonable because, while the noising term applies individually to each component of the data, the denoising process operates on the joint variable, allowing for a more comprehensive and interdependent correction that reflects the combined influence of all components.

**Proposition 7** (Upper bound time derivative of unconditional multivariate rate matrix)*. For $z_t^{1:D}, z_{t+dt}^{1:D} \in \mathcal{Z}^D$ and $t \in (0, 1)$, under Theorem 6, we have:*

$$|\partial_t R_t^{1:D}(z_t^{1:D}, z_{t+dt}^{1:D})| \leq 2B_t ZD,$$

*with $B_t = \sup_{d \in 1,\dots,D} B_t^{(d)}$.*

*Proof.* From Eq. (11), the unconditional rate matrix is given by:

$$R_t(z_t^{1:D}, z_{t+dt}^{1:D}) = \sum_{d=1}^{D} \delta(z_t^{1:D\setminus(d)}, z_{t+dt}^{1:D\setminus(d)}) \, R_t^{(d)}(z_t^{1:D}, z_{t+dt}^{(d)})$$

$$= \sum_{d=1}^{D} \delta(z_t^{1:D\setminus(d)}, z_{t+dt}^{1:D\setminus(d)}) \, \mathbb{E}_{p_{1|t}^{(d)}(z_1^{(d)}|z_t^{1:D})} \left[ R_t^{*(d)}(z_t^{(d)}, z_{t+dt}^{(d)}|z_1^{(d)}) \right]$$

$$= \sum_{d=1}^{D} \delta(z_t^{1:D\setminus(d)}, z_{t+dt}^{1:D\setminus(d)}) \sum_{z_1^{(d)} \in \mathcal{Z}} p_{1|t}^{(d)}(z_1^{(d)}|z_t^{1:D}) R_t^{*(d)}(z_t^{(d)}, z_{t+dt}^{(d)}|z_1^{(d)}).$$

So, by linearity of the time derivative, we have:

$$\left| \partial_t R_t(z_t^{1:D}, z_{t+dt}^{1:D}) \right| = \left| \sum_{d=1}^{D} \delta(z_t^{1:D\setminus(d)}, z_{t+dt}^{1:D\setminus(d)}) \sum_{z_1^{(d)} \in \mathcal{Z}} p_{1|t}^{(d)}(z_1^{(d)}|z_t^{1:D}) \, \partial_t R_t^{*(d)}(z_t^{(d)}, z_{t+dt}^{(d)}|z_1^{(d)}) \right|$$

$$\leq \sum_{d=1}^{D} \delta(z_t^{1:D\setminus(d)}, z_{t+dt}^{1:D\setminus(d)}) \sum_{z_1^{(d)} \in \mathcal{Z}} p_{1|t}^{(d)}(z_1^{(d)}|z_t^{1:D}) \left| \partial_t R_t^{*(d)}(z_t^{(d)}, z_{t+dt}^{(d)}|z_1^{(d)}) \right|$$

$$\leq \sum_{d=1}^{D} \delta(z_t^{1:D\setminus(d)}, z_{t+dt}^{1:D\setminus(d)}) \sum_{z_1^{(d)} \in \mathcal{Z}} B_t^{(d)} \, p_{t|1}(z_t^{(d)}|z_1^{(d)})^2 \frac{2}{p_{t|1}(z_t^{(d)}|z_1^{(d)})^2}$$

$$\leq \sum_{d=1}^{D} \delta(z_t^{1:D\setminus(d)}, z_{t+dt}^{1:D\setminus(d)}) \, 2 B_t Z$$

$$\leq 2 B_t Z D,$$

where in the first inequality triangular we apply triangular inequality; in the second inequality, we use Theorem 5 and Theorem 6 to upper bound $|\partial_t R_t^{*(d)}(z_t^{(d)}, z_{t+dt}^{(d)}|z_1^{(d)})|$ and $p_{1|t}(z_1^{(d)}|z_t^{1:D})$, respectively. $\square$

Now, we finally start the proof of Theorem 8.

**Theorem 8** (Bounded deviation of the generated distribution). *Let $\{z_t^{1:D}\}_{t\in[0,1]} \in \mathcal{Z}^D \times [0,1]$ be a CTMC starting with $p_0(z_0^{1:D}) = p_\epsilon$ and ending with $p_1(z_1^{1:D}) = p_{data}$, whose groundtruth rate matrix is $R_t$. Additionally, let $(y_k^{1:D})_{k=0,\frac{1}{K},...,1}$ be a Euler sampling approximation of that CTMC, with maximum step size $\Delta t = \sup_k \Delta t_k$ and an approximate rate matrix $R_t^\theta$. Then, under Theorem 6, the following total variation bound holds:*

$$\|p(y_1^{1:D}) - p_{data}\|_{TV} \leq UZD + B(ZD)^2 \Delta t + O(\Delta t),$$

*where* $U = \sup\limits_{\substack{t\in[0,1], \\ z_t^{1:D}, \, z_{t+dt}^{1:D} \in \mathcal{Z}^D}} \sqrt{C_0 + C_1 \mathbb{E}_{p_1(z_1^{1:D})} \left[ p_{t|1}(z_t^{1:D}|z_1^{1:D}) \sum_{d=1}^{D} -\log p_{1|t}^\theta(z_1^{(d)}|z_t^{1:D}) \right]}$ *and* $B = \sup\limits_{\substack{t\in[0,1], z_1^{(d)} \in \mathcal{Z} \\ z_t^{1:D} \in \mathcal{Z}^D}} B_t^{(d)}.$

*Proof.* We start the proof by clarifying the notation for the Euler sampling approximation process. We denote its discretization timesteps by $0 = t_0 < t_1 < \ldots < t_K = 1$, with $\Delta t_k = t_k - t_{k-1}$. It is initiated at the same limit distribution as the groundtruth CTMC, $p_\epsilon$, and the bound to be proven will quantify the deviation that the approximated procedure incurs in comparison to the groundtruth CTMC. To accomplish so, we define $\mathcal{R}_k^{\theta,E} = \mathcal{R}_{t_{k-1} \to t_k}^{\theta,E}$ as the Markov kernel that corresponds to applying Euler sampling with the approximated rate matrix $R_t^\theta$, moving from $t_{k-1}$ to $t_k$. Therefore, $\mathcal{R}^{\theta,E} = \mathcal{R}_1^{\theta,E} \mathcal{R}_2^{\theta,E} \ldots \mathcal{R}_K^{\theta,E}$ and $p(y_K^{1:D}) = \mathcal{R}^{\theta,E^T} p_\epsilon$.

We first apply the same decomposition to the left-hand side of Theorem 8, as (Campbell et al., 2022), Theorem 1:

$$
\begin{aligned}
\|p(y_K^{1:D}) - p_{\text{data}}\|_{\text{TV}} &= \|\mathcal{R}^{\theta,E^{\top}} p_\epsilon - p_{\text{data}}\|_{\text{TV}} \\
&\leq \|\mathcal{R}^{\theta,E^{\top}} p_\epsilon - \mathbb{P}_{1|0}^{\top} p_\epsilon\|_{\text{TV}} + \|p_\epsilon - \underbrace{p(z_0^{1:D})}_{=p_\epsilon}\|_{\text{TV}} \quad (21) \\
&\leq \|\mathcal{R}^{\theta,E^{\top}} p_\epsilon - \mathbb{P}_{1|0}^{\top} p_\epsilon\|_{\text{TV}} \\
&\leq \sum_{k=1}^{K} \sup_{\nu} \|\mathcal{R}_k^{\theta,E^{\top}} \nu - \mathcal{P}_k^{\top} \nu\|_{\text{TV}}, \quad (22)
\end{aligned}
$$

where, in Eq. (21), $\mathbb{P}_{1|0}$ denotes the path measure of the exact groundtruth CTMC and the difference between limit distributions (second term from Eq. (21)) is zero since in flow matching the convergence to the limit distribution via linear interpolation is not asymptotic, in constrast to diffusion models (Austin et al., 2021), but actually attained at $t = 0$. In Eq. (22), we introduce the stepwise path measure, i.e., $\mathcal{P}_k = \mathbb{P}_{t_k|t_{k-1}}$, such that $\mathbb{P}_{T|0} = \mathcal{P}_1 \mathcal{P}_2 \ldots \mathcal{P}_K$. Therefore, finding the intended upper bound amounts to establish bounds on the total variation distance for each interval $[t_{k-1}, t_k]$.

For any distribution $\nu$:

$$
\begin{aligned}
\|\mathcal{R}_k^{\theta,E^{\top}} \nu - \mathcal{P}_k^{\top} \nu\|_{\text{TV}} &\leq \|\mathcal{R}_k^{\theta,E^{\top}} \nu - \mathcal{R}_k^{\theta^{\top}} \nu + \mathcal{R}_k^{\theta^{\top}} \nu - \mathcal{P}_k^{\top} \nu\|_{\text{TV}} \\
&\leq \|\mathcal{P}_k^{\top} \nu - \mathcal{R}_k^{\theta^{\top}} \nu\|_{\text{TV}} + \|\mathcal{R}_k^{\theta^{\top}} \nu - \mathcal{R}_k^{\theta,E^{\top}} \nu\|_{\text{TV}}, \quad (23)
\end{aligned}
$$

where $\mathcal{R}_k^{\theta}$ denotes the resulting Markov kernel of running a CTMC with constant rate matrix $R_{t_{k-1}}^{\theta}$ between $t_{k-1}$ and $t_k$.

For the first term, we use Proposition 5 from (Campbell et al., 2022) to relate the total variation distance imposed by the Markov kernels with the difference between the corresponding rate matrices:

$$
\begin{aligned}
\|\mathcal{P}_k^{\top} \nu - \mathcal{R}_k^{\theta^{\top}} \nu\|_{\text{TV}} &\leq \int_{t_{k-1}}^{t_k} \sup_{z_t^{1:D} \in \mathcal{Z}^D} \left\{ \sum_{z_{t+dt}^{1:D} \neq z_t^{1:D}} \left| R_t(z_t^{1:D}, z_{t+dt}^{1:D}) - R_{t_{k-1}}^{\theta}(z_t^{1:D}, z_{t+dt}^{1:D}) \right| \right\} dt \\
&\leq \underbrace{\int_{t_{k-1}}^{t_k} \sup_{z_t^{1:D} \in \mathcal{Z}^D} \left\{ \sum_{z_{t+dt}^{1:D} \neq z_t^{1:D}} \left| R_t(z_t^{1:D}, z_{t+dt}^{1:D}) - R_{t_{k-1}}(z_t^{1:D}, z_{t+dt}^{1:D}) \right| \right\} dt}_{\text{Discretization Error}} \\
&\quad + \underbrace{\int_{t_{k-1}}^{t_k} \sup_{z_t^{1:D} \in \mathcal{Z}^D} \left\{ \sum_{z_{t+dt}^{1:D} \neq z_t^{1:D}} \left| R_{t_{k-1}}(z_t^{1:D}, z_{t+dt}^{1:D}) - R_{t_{k-1}}^{\theta}(z_t^{1:D}, z_{t+dt}^{1:D}) \right| \right\} dt}_{\text{Estimation Error}}
\end{aligned}
$$

The first term consists of the discretization error, where we compare the chain with groundtruth rate matrix changing continuously between $t_{k-1}$ and $t_k$ with its discretized counterpart, i.e., a chain where the rate matrix is held constant to its value at the beginning of the interval. The second corresponds to the estimation error, where we compare the chain generated by the discretized groundtruth rate matrix with an equally discretized chain but that uses an estimated rate matrix instead. For the former, we have:

$$\int_{t_{k-1}}^{t_k} \sup_{z_t^{1:D} \in \mathcal{Z}^D} \left\{ \sum_{z_{t+\mathrm{d}t}^{1:D} \neq z_t^{1:D}} \left| R_t(z_t^{1:D}, z_{t+\mathrm{d}t}^{1:D}) - R_{t_{k-1}}(z_t^{1:D}, z_{t+\mathrm{d}t}^{1:D}) \right| \right\} \mathrm{d}t$$

$$\leq \int_{t_{k-1}}^{t_k} \sup_{z_t^{1:D} \in \mathcal{Z}^D} \left\{ \sum_{z_{t+\mathrm{d}t}^{1:D} \neq z_t^{1:D}} \left| \partial_t R_{t_c}(z_t^{1:D}, z_{t+\mathrm{d}t}^{1:D})(t - t_{k-1})) \right| \right\} \mathrm{d}t \tag{24}$$

$$\leq \int_{t_{k-1}}^{t_k} ZD \sup_{z_t^{1:D}, z_{t+\mathrm{d}t}^{1:D} \in \mathcal{Z}^D} \left\{ \left| \partial_t R_{t_c}(z_t^{1:D}, z_{t+\mathrm{d}t}^{1:D}) \right| \right\} |t - t_{k-1}| \, \mathrm{d}t \tag{25}$$

$$\leq 2Z^2 D^2 \int_{t_{k-1}}^{t_k} B_t \, |t - t_{k-1}| \, \mathrm{d}t, \tag{26}$$

$$= B_k (ZD\Delta t_k)^2, \tag{27}$$

where, in Eq. (24), we use the Mean Value Theorem, with $t_c \in (t_{k-1}, t_k)$; in Eq. (25), we use the fact that there are $ZD$ values of $z_{t+\mathrm{d}t}^{1:D}$ that differ at most in only one coordinate from $z_t^{1:D}$; in Eq. (26), we use the result from Theorem 7 to upper bound the time derivative of the multivariate unconditional rate matrix; and finally, in Eq. (27), we define $B_k = \sup_{t \in (t_{k-1}, t_k)} B_t$.

For the estimation error term, we have:

$$\int_{t_{k-1}}^{t_k} \sup_{z_t^{1:D} \in \mathcal{Z}^D} \left\{ \sum_{z_{t+\mathrm{d}t}^{1:D} \neq z_t^{1:D}} \left| R_{t_{k-1}}(z_t^{1:D}, z_{t+\mathrm{d}t}^{1:D}) - R_{t_{k-1}}^\theta(z_t^{1:D}, z_{t+\mathrm{d}t}^{1:D}) \right| \right\} \mathrm{d}t$$

$$\leq \int_{t_{k-1}}^{t_k} U_k ZD \, \mathrm{d}t, \tag{28}$$

$$\leq U_k ZD\Delta t_k, \tag{29}$$

where, in Eq. (28), we use again the fact that there are $ZD$ values of $z_{t+\mathrm{d}t}^{1:D}$ that differ at most in only one coordinate from $z_t^{1:D}$ along with the estimation error upper bound from Theorem 11. In particular, we consider $U_k = \sup_{\substack{t \in [t_{k-1}, t_k], \\ z_t^{1:D}, \, z_{t+\mathrm{d}t}^{1:D} \in \mathcal{Z}^D}} U_k^{z_t^{1:D} \to z_{t+\mathrm{d}t}^{1:D}}$, with:

$$U_k^{z_t^{1:D} \to z_{t+\mathrm{d}t}^{1:D}} = \sqrt{C_0 + C_1 \mathbb{E}_{p_1(z_1^{1:D})} \left[ p_{t|1}(z_t^{1:D}|z_1^{1:D}) \sum_{d=1}^{D} -\log p_{1|t}^{\theta,(d)}(z_1^{(d)}|z_t^{1:D}) \right]},$$

i.e., the square root of the right-hand side of Eq. (13).

It remains to bound the second term from Eq. (23). We start by analyzing the Markov kernel $\mathcal{R}_k^\theta$ corresponding to a Markov chain with *constant* rate matrix $R_{t_{k-1}}^\theta$ between $t_{k-1}$ and $t_k$. In that case, from Eq. (19) we obtain:

$$\mathcal{R}_k^\theta = e^{R_{t_{k-1}}^\theta \Delta t_k}$$

$$= \sum_{i=0}^{\infty} \frac{(R_{t_{k-1}}^\theta \Delta t_k)^i}{i!}$$

$$= I + R_{t_{k-1}}^\theta \Delta t_k + \frac{(R_{t_{k-1}}^\theta \Delta t_k)^2}{2!} + \frac{(R_{t_{k-1}}^\theta \Delta t_k)^3}{3!} + \cdots$$

On the other hand, we have from Eq. (4) that sampling with the Euler approximation in multivariate Markov chain corresponds to:

$$\tilde{p}_{t_k|t_{k-1}}(z_{t_k}^{1:D}|z_{t_{k-1}}^{1:D}) =$$

$$= \prod_{d=1}^{D} \tilde{p}_{t_k|t_{k-1}}^{(d)}(z_{t_k}^{(d)}|z_{t_{k-1}}^{1:D})$$

$$= \prod_{d=1}^{D} \delta(z_{t_{k-1}}^{(d)}, z_{t_k}^{(d)}) + R_t^{\theta,(d)}\left(z_{t_{k-1}}^{1:D}, z_{t_k}^{(d)}\right) \Delta t_k \tag{30}$$

$$= \delta\left(z_{t_{k-1}}^{1:D}, z_{t_k}^{1:D}\right) + \Delta t_k \sum_{d=1} \delta\left(z_{t_{k-1}}^{1:D\backslash(d)}, z_{t_k}^{1:D\backslash(d)}\right) R_t^{\theta,(d)}\left(z_{t_{k-1}}^{1:D}, z_{t_k}^{(d)}\right) + O(\Delta t_k^2),$$

where in Eq. (30) we have that the approximated transition rate matrix is computed according to Eq. (11) but using $p_{1|t}^{\theta,(d)}(z_1^{(d)}|z_t^{1:D})$ instead of $p_{1|t}^{(d)}(z_1^{(d)}|z_t^{1:D})$. To obtain $\mathcal{R}_k^{\theta,E}$, we just need to convert $\tilde{p}_{t_k|t_{k-1}}(z_{t_k}^{1:D}|z_{t_{k-1}}^{1:D})$ into matrix form. Note that:

- $\delta(z_{t_{k-1}}^{1:D}, z_{t_k}^{1:D})$ corresponds to $I$ in matrix form;

- From Eq. (10), $\sum_{d=1} \delta\left(z_{t_{k-1}}^{1:D\backslash(d)}, z_{t_k}^{1:D\backslash(d)}\right) R_t^{\theta,(d)}\left(z_{t_{k-1}}^{1:D}, z_{t_k}^{(d)}\right)$ corresponds to $R_{t_{k-1}}^\theta$ in matrix form.

These correspondences yield:

$$\mathcal{R}_k^{\theta,E} = I + \Delta t_k\, R_{t_{k-1}}^\theta + O\left(\Delta t_k^2\right)$$

Consequently, we have:

$$\left\|\mathcal{R}_k^{\theta\top}\nu - \mathcal{R}_k^{\theta,E\top}\nu\right\|_{\text{TV}} \in O\left(\Delta t_k^2\right). \tag{31}$$

Therefore, we get the intended result by gathering the results from Eq. (27), Eq. (29), and Eq. (31).

$$\|p(y_1^{1:D}) - p_{\text{data}}\|_{\text{TV}} \leq \sum_{k=1}^{K} \left(U_k ZD\Delta t_k + B_k(ZD\Delta t_k)^2 + O(\Delta t_k^2)\right)$$

$$\leq UZD + B(ZD)^2\Delta t + O(\Delta t),$$

where $\Delta t = \sup_k \Delta t_k$ is the maximum step size, $\sum_{k=1}^{K} \Delta t_k = 1$, $U = \sup_k U_k$ and $B = \sup_k B_k$. □

Below we provide a corollary of Theorem 8, which consists of its instantiation for graphs.

**Corollary 2** (Bounded deviation of the generated graph distribution). *Let $\{G_t\}_{t\in[0,1]}$ be a CTMC over graphs starting with $p_0(G_0) = p_\epsilon$ and ending with $p_1(G_1) = p_{data}$, whose groundtruth rate matrix is $R_t$. Additionally, let $(\bar{G}_k)_{k=0,1,\ldots,K}$ be a Euler sampling approximation of that CTMC. Then, we have:*

$$\|p(G_1) - p_{data}\|_{TV} \leq \bar{U}\left(XN + E\frac{N(N-1)}{2}\right) + \bar{B}\left(XN + E\frac{N(N-1)}{2}\right)^2 \Delta t + O(\Delta t)$$

*where $\bar{B}$ is defined similarly to $B$ from Theorem 8 but for graphs:* $\bar{B} = \sup\limits_{t\in[0,1]} \sup\limits_{\substack{x_1^{(n)}\in\mathcal{X}, e_1^{(ij)}\in\mathcal{E} \\ G_t}} \{\bar{B}_t^{(n)}, \bar{B}_t^{(ij)}\}$, *with*

$\bar{B}_t^{(n)}, \bar{B}_t^{(ij)} > 0$ *defined according to Theorem 6:*

$$p_{1|t}(x_1^{(n)}|G_t) \leq \bar{B}_t^{(n)} p_{t|1}(x_t^{(n)}|x_1^{(n)})^2$$

$$p_{1|t}(e_1^{(ij)}|G_t) \leq \bar{B}_t^{(ij)} p_{t|1}(e(ij)_t|e_1^{(ij)})^2,$$

*and*

$$U = \sup_{\substack{t \in [0,1], \\ G_t,\, G_{t+dt} \in \mathcal{Z}^D}} \left( \bar{C}_0 + \bar{C}_1 \, \mathbb{E}_{p_1(G_1)} \left[ p_{t|1}(G_t|G_1) \sum_{n=1}^N -\log p_{1|t}^{\theta,(n)}(x_1^{(n)}|G_t) \right] \right.$$

$$\left. + \bar{C}_2 \, \mathbb{E}_{p_1(G_1)} \left[ p_{t|1}(G_t|G_1) \sum_{1 \le i < j \le N} -\log p_{1|t}^{\theta,(ij)}(e_1^{(ij)}|G_t) \right] \right)^{\frac{1}{2}}$$

*Proof.* Considering the definition of a graph as $G = \left( x^{1:n:N, e^{1 \le i < j \le N}} \right)$, we proceed through the different steps of the proof of Theorem 8.

In particular, for the discretization error, we obtain:

$$\int_{t_{k-1}}^{t_k} \sup_{G_t} \left\{ \sum_{G_{t+dt} \ne G_t} \left| R_t(G_t, G_{t+dt}) - R_{t_{k-1}}(G_t, G_{t+dt}) \right| \right\} dt = \bar{B}_k \left( XN + E \frac{N(N-1)}{2} \right)^2 \Delta t_k^2,$$

since there are $XN + E\frac{N(N-1)}{2}$ values of $G_{t+dt}$ that differ at most in only one coordinate (node or edge) from $G_t$, with $\bar{B}_k = \sup_{t \in (t_{k-1}, t_k)} \bar{B}_t$.

For the estimation error:

$$\int_{t_{k-1}}^{t_k} \sup_{G_t} \left\{ \sum_{G_{t+dt} \ne G_t} \left| R_{t_{k-1}}(G_t, G_{t+dt}) - R_{t_{k-1}}^\theta(G_t, G_{t+dt}) \right| \right\} dt \le \bar{U}_k \left( XN + E \frac{N(N-1)}{2} \right) \Delta t_k,$$

since, again, there are $XN + E\frac{N(N-1)}{2}$ values of $G_{t+dt}$ that differ at most in only one coordinate from $G_t$ and $U_k = \sup_{\substack{t \in [t_{k-1}, t_k], \\ G_t,\, G_{t+dt}}} \bar{U}_k^{G_t \to G_{t+dt}}$, where:

$$\bar{U}_k^{G_t \to G_{t+dt}} = \left( \bar{C}_0 + \bar{C}_1 \, \mathbb{E}_{p_1(G_1)} \left[ p_{t|1}(G_t|G_1) \sum_{n=1}^N -\log p_{1|t}^{\theta,(n)}(x_1^{(n)}|G_t) \right] \right.$$

$$\left. + \bar{C}_2 \, \mathbb{E}_{p_1(G_1)} \left[ p_{t|1}(G_t|G_1) \sum_{1 \le i < j \le N} -\log p_{1|t}^{\theta,(ij)}(e_1^{(ij)}|G_t) \right] \right)^{\frac{1}{2}},$$

i.e., the square root of the right-hand side of Theorem 1.

The term $\|\mathcal{R}_k^{\theta\top}\nu - \mathcal{R}_k^{\theta,E\top}\nu\|_{\text{TV}}$ remains $O\left(\Delta t_k^2\right)$.

Therefore, by finally aggregating the terms above as in Theorem 8, we obtain:

$$\|p(G_1) - p_{\text{data}}\|_{\text{TV}} \le \bar{U} \left( XN + E \frac{N(N-1)}{2} \right) + \bar{B} \left( XN + E \frac{N(N-1)}{2} \right)^2 \Delta t + O(\Delta t)$$

$\square$

### D.1.3. CRITICAL ANALYSIS AND POSITIONING OF THEOREM 1 AND THEOREM 2

Theorem 1 establishes that minimizing the cross-entropy (CE) loss directly corresponds to minimizing an upper bound on the rate matrix estimation error. This result provides a direct and principled justification for using the CE loss, as it promotes accurate sampling from the underlying CTMC.

Prior work derive an ELBO loss and motivate using only the CE loss based on approximations upon the derived ELBO by dropping the rate matrix-dependent terms (see Appendix C.2 in (Campbell et al., 2024)). In contrast, our bounds do

not require such simplifications and are agnostic to rate matrix design choices (e.g., stochasticity level), reinforcing our training-sampling decoupling claim. Additionally, we note that concurrent work also derives a tractable ELBO (Shaul et al., 2025); however, it addresses a different family of conditional rate matrices.

Theorem 2 complements our theoretical analysis by bounding the deviation of the generated distribution from the target distribution, combining two sources of error: the discretization error from CTMC sampling, which is $\mathcal{O}(\Delta t)$, and the rate matrix estimation error, captured by Theorem 1. This type of bound is a standard objective in generative modeling, aiming to guarantee that the generation process remains faithful to the underlying data distribution. For instance, Theorem 1 in Campbell et al. (2022) presents an analogous bound in the context of discrete diffusion, though with significant differences (e.g., they assume a bounded rate matrix estimation error, while we explicitly prove it).

## D.2. Theoretical Results on Architectural Expressivity

We now proceed to the graph specific theoretical results.

### D.2.1. NODE PERMUTATION EQUIVARIANCE AND INVARIANCE PROPERTIES

The different components of a graph generative model have to respect different graph symmetries. For example, the permutation equivariance of the model architecture ensures the output changes consistently with any reordering of input nodes, while permutation-invariant loss evaluates the model's performance consistently across isomorphic graphs, regardless of node order. We provide a proof for related properties included in Lemma 3 as follows.

**Lemma 3** (Node Permutation Equivariance and Invariance Properties of DeFoG). *For any permutation-equivariant denoising neural network, the loss function of DeFoG is permutation invariant, and its sampling probability is permutation invariant.*

*Proof.* Recall that we denote an undirected graph with $N$ nodes by $G = (x^{1:n:N}, e^{1:i<j:N})$. Here, each node variable is represented as $x^n \in \mathcal{X} = \{1, \dots, X\}$, and each edge variable as $e^{(ij)} \in \mathcal{E} = \{1, \dots, E\}$. We also treat $G$ as a multivariate data point consisting of $D$ discrete variables including all nodes and all edges.

We then consider a permutation function $\sigma$, which is applied to permute the graph's node ordering. Under this permutation, the index $n$ will be mapped to $\sigma(n)$. We denote the ordered set of nodes and edges in the original ordering by $x^{(1:n:N)}$ and $e^{(1:i<j:N)}$, respectively, and by $x'^{(1:n:N)}$ and $e'^{(1:i<j:N)}$ after permutation. Additionally, $x^{(n)}$ denotes the $n$-th entry of the corresponding ordered set (and analogously for edges). By definition, the relationship between the original and permuted entries of the ordered sets is given by: $x'^{(n)} = x^{\sigma^{-1}(n)}$ and $e'^{(ij)} = e^{(\sigma^{-1}(i), \sigma^{-1}(j))}$.

**Permutation Equivariant Model**  We begin by proving that the DeFoG architecture is permutation-equivariant, including the network architecture and the additional features employed.

- Permutation Equivariance of RRWP Features: Recall that the RRWP features until $K - 1$ steps are defined as $\text{RRWP}(M) = P = [I, M, \dots, M^{K-1}] \in \mathbb{R}^{n \times n \times K}$, where $M^k = (D^{-1}A)^k$, $0 \le k < K$.

  We first prove that $M(A) = D^{-1}A$ is permutation equivariant:

$$\begin{aligned}
M(A')^{(ij)} &= (D'^{-1}A')^{(ij)} \\
&= (1/(D')^{(ii)})(A')^{(ij)} \\
&= (1/D^{(\sigma^{-1}(i), \sigma^{-1}(i))})(A)^{(\sigma^{-1}(i), \sigma^{-1}(j))} \\
&= (D^{-1}A)^{(\sigma^{-1}(i), \sigma^{-1}(j))} \\
&= (M(A)')^{(ij)}.
\end{aligned}$$

  To facilitate notation, in the following proofs, we consider the matrix $\pi \in \{0, 1\}^{N \times N}$ representing the same permutation function $\sigma$, with the permuted features represented as $\pi M \pi^T$. We then prove that RRWP is permutation

equivariant.

$$
\begin{aligned}
P(\pi M \pi^T) &= [\pi I \pi^T, \pi M \pi^T, \ldots, (\pi M \pi^T)^{K-1}] \\
&= [\pi I \pi^T, \pi M \pi^T, \ldots, (\pi M^{K-1} \pi^T)], \quad \text{since } \pi^T \pi = \mathbf{I}_N \\
&= \pi [I, M, \ldots, M^{K-1}] \pi^T \\
&= \pi P(M) \pi^T.
\end{aligned}
$$

- Permutation Equivariance of Model Layers: The model layers (MLP, FiLM, PNA, and self-attention) preserve permutation equivariance, as shown in prior work (e.g., Vignac et al. (2022) in Lemma 3.1).

Hence, since all of its components are permutation-equivariant, so is the DeFoG full architecture. The next results hold for any permutation equivariant denoising neural network.

**Permutation Invariant Loss Function**  DeFoG's loss consists of summing the cross-entropy loss between the predicted clean graph and the true clean graph (node and edge-wise). Vignac et al. (2022), Lemma 3.2, provide a concise proof that this loss is permutation invariant.

**Permutation Invariant Sampling Probability**  Given a noisy graph $G_0$ sampled from the initial distribution, $p_0$, the rate matrix at any time point $t \in [0, 1]$ defines the denoising process.

Recall that the conditional rate matrix for a variable $z_t$ at each time step $t$ is defined as:

$$
R_t^*(z_t, z_{t+\mathrm{d}t}|z_1) = \frac{\text{ReLU}\left[\partial_t p_{t|1}(z_{t+\mathrm{d}t}|z_1) - \partial_t p_{t|1}(z_t|z_1)\right]}{Z_t^{>0} \, p_{t|1}(z_t|z_1)}.
$$

In our multivariate formulation, we compute this rate matrix independently for each variable inside the graph. We denote the concatenated rate matrix entries for all nodes with $R_t^*(x_t'^{(1:N)}, x_{t+\mathrm{d}t}'^{(1:N)}|x_1'^{(1:N)})$.

In the following part, we demonstrate the node permutation equivariance of the rate matrix predicted by the trained equivariant network, denoted by $f_\theta$. The proof for edges follows a similar logic. Suppose that the noisy graph $G_t$ is permuted, and the permuted graph has nodes denoted by $x_t'^{(1:N)}$, i.e., $x_t'^{(n)} = x_t^{\sigma^{-1}(n)}$. We have:

$$
\begin{aligned}
& R_t^*\left(x_t'^{(1:N)}, x_{t+\mathrm{d}t}'^{(1:N)}|x_1'^{(1:N)}\right)^{(n)} \\
=& R_t^*\left(x_t'^{(n)}, x_{t+\mathrm{d}t}'^{(n)}|x_1'^{(n)}\right) \\
=& \frac{\text{ReLU}\left[\partial_t p_{t|1}(x_{t+\mathrm{d}t}'^{(n)}|x_1'^{(n)}) - \partial_t p_{t|1}(x_t'^{(n)}|x_1'^{(n)})\right]}{Z_t^{>0} \, p_{t|1}(x_t'^{(n)}|x_1'^{(n)})} \\
=& \frac{\text{ReLU}\left[\partial_t p_{t|1}(x_{t+\mathrm{d}t}^{\sigma^{-1}(n)}|x_1'^{(n)}) - \partial_t p_{t|1}(x_t^{\sigma^{-1}(n)}|x_1'^{(n)})\right]}{Z_t^{>0} \, p_{t|1}(x_t^{\sigma^{-1}(n)}|x_1'^{(n)})}, & (32) \\
=& \frac{\text{ReLU}\left[\partial_t p_{t|1}(x_{t+\mathrm{d}t}^{\sigma^{-1}(n)}|x_1^{\sigma^{-1}(n)}) - \partial_t p_{t|1}(x_t^{\sigma^{-1}(n)}|x_1^{\sigma^{-1}(n)})\right]}{Z_t^{>0} \, p_{t|1}(x_t^{\sigma^{-1}(n)}|x_1^{\sigma^{-1}(n)})}, & (33) \\
=& R_t^*(x_t^{\sigma^{-1}(n)}, x_{t+\mathrm{d}t}^{\sigma^{-1}(n)}|x_1^{\sigma^{-1}(n)}) \\
=& R_t^*(x_t^{(1:N)}, x_{t+\mathrm{d}t}^{(1:N)}|x_1^{(1:N)})^{\sigma^{-1}(n)}.
\end{aligned}
$$

In Eq. (32), we use the definition of permuted ordered set for $x_t'$ and $x_{t+\mathrm{d}t}'$, and, in Eq. (33), we use that $f_\theta$ is equivariant.

Furthermore, the transition probability at each time step $t$ is given by:

$$\tilde{p}_{t+\Delta t|t}(G_{t+\Delta t}^{1:D}|G_t^{1:D}) = \prod_{d=1}^{D} \tilde{p}_{t+\Delta t|t}^{(d)}(G_{t+\Delta t}^{(d)}|G_t^{1:D})$$

$$= \prod_{d=1}^{D} \left( \delta(G_t^{(d)}, G_{t+\Delta t}^{(d)}) + \mathbb{E}_{p^\theta(G_1^{(d)}|G_t^{1:D})} \left[ R_t^{(d)}(G_t^{(d)}, G_{t+\Delta t}^{(d)}|G_1^{(d)}) \right] \Delta t \right).$$

The transition probability $\tilde{p}_{t+\Delta t|t}(G_{t+\Delta t}^{1:D}|G_t^{1:D})$ is expressed as a product over all nodes and edges, an operation that is inherently a permutation invariant function with respect to node ordering. Furthermore, as demonstrated earlier, the term $\mathbb{E}_{p^\theta(G_1^{(d)}|G_t^{1:D})} \left[ R_t^{(d)}(G_t^{(d)}, G_{t+\Delta t}^{(d)}|G_1^{(d)}) \right]$ is permutation equivariant since $R_t^{(d)}$ and $p_{t|1}^\theta(G_1^{(d)}|G_t^{1:D})$ are both permutation equivariant if model $f_\theta$ is permutation equivariant. Consequently, since the composition of these components yields a permutation invariant function, we conclude that the transition probability of the considered CTMC is permutation invariant.

We finally verify the final sampling probability, $\tilde{p}_1(G_1^{1:D})$, is permutation-invariant. To simulate $G_1^{1:D}$ over $K$ time steps $[0 = t_0, t_1, \ldots, t_K = 1]$, we can marginalize it first by taking the expectation over the state at the last time step $t_{K-1}$. Specifically, we have $p_1(G_1^{1:D}) = \mathbb{E}_{p_{t_{K-1}}(G_{t_{K-1}}^{1:D})} \left[ \tilde{p}_{1|t_{K-1}}(G_1^{1:D}|G_{t_{K-1}}^{1:D}) \right]$. Since the process is Markovian, this expression can be sequentially extended over the $T$ steps through successive expectations.

$$p_1(G_1^{1:D}) = \mathbb{E}_{p_{t_{K-1}}(G_{t_{K-1}}^{1:D})} \left[ \tilde{p}_{1|t_{K-1}}(G_1^{1:D}|G_{t_{K-1}}^{1:D}) \right]$$

$$= \mathbb{E}_{p_{t_{K-2}}(G_{t_{K-2}}^{1:D})} \left[ \underbrace{\mathbb{E}_{\tilde{p}_{t_{K-1}|t_{K-2}}(G_{t_{K-1}}^{1:D}|G_{t_{K-2}}^{1:D})} \left[ \tilde{p}_{1|t_{K-1}}(G_1^{1:D}|G_{t_{K-1}}^{1:D}) \right]}_{\tilde{p}_{1|t_{K-2}}(G_1^{1:D}|G_{t_{K-2}}^{1:D})} \right]$$

$$\ldots$$

$$= \mathbb{E}_{p_0(G_0^{1:D})} \left[ \mathbb{E}_{\tilde{p}_{t_1|0}(G_{t_1}^{1:D}|G_0^{1:D})} \left[ \ldots \mathbb{E}_{\tilde{p}_{t_{K-1}|t_{K-2}}(G_{t_{K-1}}^{1:D}|G_{t_{K-2}}^{1:D})} \left[ \tilde{p}_{1|t_{K-1}}(G_1^{1:D}|G_{t_{K-1}}^{1:D}) \right] \right] \right]$$

Due to the fact that each function in the sequence is itself permutation-invariant and that the initial distribution $p_0(G_0^{1:D})$ is permutation invariant (see Appendix C.1), the composition of permutation-invariant functions preserves this invariance throughout. Thus, the final sampling probability is invariant over isomorphic graphs. $\square$

### D.2.2. ADDITIONAL FEATURES EXPRESSIVITY

This section explains the expressivity of the RRWP features used in DeFoG. We summarize the findings of Ma et al. (2023) in Theorem 7, who establish that, by encoding random walk probabilities, the RRWP positional features can be used to arbitrarily approximate several essential graph properties when fed into an MLP. Specifically, point 1 shows that RRWP with $K-1$ steps encodes all shortest path distances for nodes up to $K-1$ hops. Additionally, points 2 and 3 indicate that RRWP features effectively capture diverse graph propagation dynamics.

**Proposition 7** (Expressivity of an MLP with RRWP encoding (Ma et al., 2023)). *For any $n \in \mathbb{N}$, let $G_n \subseteq \{0,1\}^{n \times n}$ denote all adjacency matrices of $n$-node graphs. For $K \in \mathbb{N}$, and $A \in \mathbb{G}_n$, consider the RRWP:*

$$P = [I, M, \ldots, M^{K-1}] \in \mathbb{R}^{n \times n \times K}$$

*Then, for any $\epsilon > 0$, there exists an* $\mathrm{MLP} : \mathbb{R}^{K-1} \to \mathbb{R}$ *acting independently across each $n$ dimension such that* $\mathrm{MLP}(P)$ *approximates any of the following to within $\epsilon$ error:*

1. $\mathrm{MLP}(P)_{ij} \approx SPD_{K-1}(i,j)$

2. $\mathrm{MLP}(P) \approx \sum_{k=0}^{K-1} \theta_k (D^{-1}A)^k$

3. $\mathrm{MLP}(P) \approx \theta_0 I + \theta_1 A$

*in which $SPD_{K-1}(i, j)$ is the $K - 1$ truncated shortest path distance, and $\theta_k \in \mathbb{R}$ are arbitrary coefficients.*

Siraudin et al. (2024) experimentally validate the effectiveness of RRWP features for graph diffusion models and propose extending their proof to additional graph properties that GNNs fail to capture (Xu et al., 2019; Morris et al., 2019). For example, an MLP with input $M^k$ with $k = N - 1$ for an $N$-node graph can approximate the connected component of each node and the number of vertices in the largest connected component. Additionally, RRWP features can be used to capture cycle-related information.

# E. Conditional Generation

In this section, we describe how to seamlessly integrate DeFoG with existing methods for CTMC-based conditioning mechanisms. In this setting, all the examples are assumed to have a label. The objective of conditional generation is to steer the generative process based on that label, so that at sampling time we can guide the model to which class of samples we are interested in obtaining.

We focus on classifier-free guidance methods, as these models streamline training by avoiding task-specific classifiers. This approach has been widely adopted for continuous state-space models, e.g., for image generation, where it has been shown to enhance the generation quality of the generative model (Ho & Salimans, 2021; Sanchez et al., 2024). Recently, Nisonoff et al. (2025) extended this method to discrete flow matching in a principled manner. In this paper, we adopt their formulation.

In this framework, 90% of the training is performed with the model having access to the label of each noisy sample. This allows the model to learn the conditional rate matrix, $R_t^\theta(z_t, z_{t+dt}|y)$. In the remaining 10% of the training procedure, the labels of the samples are masked, forcing the model to learn the unconditional generative rate matrix $R_t^\theta(z_t, z_{t+dt})$. The conditional training enables targeted and accurate graph generation, while the unconditional phase ensures robustness when no conditions are specified. The combination of both conditional and unconditional training offers a more accurate pointer to the conditional distribution, typically described by the distance between the conditional and unconditional prediction. In our framework, this pointer is defined through the ratio between the conditional and unconditional rate matrices, as follows:

$$R_t^{\theta,\gamma}(x,\tilde{x}|y) = R_t^\theta(x,\tilde{x}|y)^\gamma R_t^\theta(x,\tilde{x})^{1-\gamma} = R_t^\theta(x,\tilde{x}|y)\left(\frac{R_t^\theta(x,\tilde{x}|y)}{R_t^\theta(x,\tilde{x})}\right)^{\gamma-1},$$

where $\gamma$ denotes the guidance weight. In particular, the case with $\gamma = 1$ corresponds to standard conditional generation, while $\gamma = 0$ represents standard unconditional generation. As $\gamma$ increases, the conditioning effect described by $\left(\frac{R_t^\theta(x,\tilde{x}|y)}{R_t^\theta(x,\tilde{x})}\right)^{\gamma-1}$ is strengthened, thereby enhancing the quality of the generated samples. We observed $\gamma = 2.0$ to be the best performing value for our digital pathology experiments (Appendix G.1), as detailed in Tab. 5.

Overall, conditional generation is pivotal for guiding models to produce graphs that meet specific requirements, offering tailored solutions for complex real-world tasks. The flexibility of DeFoG, being well-suited for conditional generation, marks an important step forward in advancing this direction, promising greater adaptability and precision in future graph-based applications.

# F. Experimental Details

This section provides further details on the experimental settings used in the paper.

## F.1. Architecture and Additional Features

**Denoising Neural Architecture**  DeFoG's denoising neural network takes a noisy graph $G_t$ as input and predicts the clean marginal probability for each node $x^{(n)}$ via $p_{1|t}^{\theta,(n)}(\cdot|G_t)$ and for each edge $e^{(ij)}$ via $p_{1|t}^{\theta,(ij)}(\cdot|G_t)$. This formulation boils down the graph generative task to a graph-to-graph mapping. While both message-passing layers and graph transformers can be used for this task, graph transformers have empirically outperformed message-passing layers in graph generation (Qin et al., 2023). DeFoG thus adopts the transformer architecture of Vignac et al. (2022), using multi-head attention layers which encode node, edge, and graph-level features while preserving node permutation equivariance.

**Enhancing Model Expressivity**  Graph neural networks, including graph transformers, have inherent limitations in their expressive power (Xu et al., 2019; Zhu et al., 2023). An usual approach to overcome the limited representation power of graph neural networks consists of explicitly augmenting the inputs with features that the networks would otherwise struggle to learn. We adopt Relative Random Walk Probabilities (RRWP) encodings that are proved to be expressive for both discriminative (Ma et al., 2023) and generative settings (Siraudin et al., 2024). RRWP encodes the likelihood of traversing from one node to another in a graph through random walks of varying lengths, offering insights into graph dynamics across different hop distances. In particular, given a graph with an adjacency matrix $A$, we generate $K-1$ powers of its degree-normalized adjacency matrix, $M = D^{-1}A$, i.e., $[I, M, M^2, \ldots, M^{K-1}]$. We concatenate the diagonal entries of each power to their corresponding node embedding, while combining and appending the non-diagonal to their corresponding edge embeddings. In addition to their enhanced expressiveness, RRWP features are easy to compute and thus offer a notably more efficient and scalable alternative to the resource-intensive spectral and cycle features (Vignac et al., 2022) commonly employed in prior works (see Appendix G.4).

## F.2. Dataset Details

### F.2.1. SYNTHETIC DATASETS

Here, we describe the datasets employed in our experiments and outline the specific metrics used to evaluate model performance on each dataset. Additional visualizations of example graphs from each dataset, along with generated graphs, are provided in Figures 16 to 18.

**Description**  We use three synthetic datasets with distinct topological structures. The first is the *planar* dataset (Martinkus et al., 2022), which consists of connected planar graphs—graphs that can be drawn on a plane without any edges crossing. The second dataset, *tree* (Bergmeister et al., 2023), contains tree graphs, which are connected graphs with no cycles. Lastly, the *Stochastic Block Model (SBM)* dataset (Martinkus et al., 2022) features synthetic clustering graphs where nodes within the same cluster have a higher probability of being connected.

The *planar* and *tree* datasets exhibit well-defined deterministic graph structures, while the *SBM* dataset, commonly used in the literature, stands out due to its stochasticity, resulting from the random sampling process that governs its connectivity.

**Metrics**  We follow the evaluation procedures described by Martinkus et al. (2022); Bergmeister et al. (2023), using both dataset-agnostic and dataset-specific metrics.

First, dataset-agnostic metrics assess the alignment between the generated and training distributions for specific general graph properties. We map the graphs to their node degrees (Deg.), clustering coefficients (Clus.), orbit count (Orbit), eigenvalues of the normalized graph Laplacian (Spec.), and statistics derived from a wavelet graph transform (Wavelet). We then compute the distance to the corresponding statistics calculated for the test graphs. For each statistic, we measure the distance between the empirical distributions of the generated and test sets using Maximum Mean Discrepancy (MMD). These distances are aggregated into the *Ratio* metric. To compute this, we first calculate the MMD distances between the training and test sets for the same graph statistics. The final Ratio metric is obtained by dividing the average MMD distance between the generated and test sets by the average MMD distance between the training and test sets. A Ratio value of 1 is ideal, as the distance between the training and test sets represents a lower-bound reference for the generated data's performance.

Next, we report dataset-specific metrics using the V.U.N. framework, which assesses the proportion of graphs that are valid (V), unique (U), and novel (N). Validity is assessed based on dataset-specific properties: the graph must be planar, a tree, or statistically consistent with an SBM for the planar, tree, and SBM datasets, respectively. Uniqueness captures the proportion of non-isomorphic graphs within the generated graphs, while novelty measures how many of these graphs are non-isomorphic to any graph in the training set.

### F.2.2. MOLECULAR DATASETS

**Description**   Molecular generation is a key real-world application of graph generation. It poses a challenging task to current graph generation models to their rich chemistry-specific information, involving several nodes and edges classes and leaning how to generate them jointly, and more complex evaluation pipelines. To assess DeFoG's performance on molecular datasets, we use three benchmarks that progressively increase in molecular complexity and size.

First, we use the QM9 dataset (Wu et al., 2018), a subset of GDB9 (Ruddigkeit et al., 2012), which contains molecules with up to 9 heavy atoms.

Next, we evaluate DeFoG on the Moses benchmark (Polykovskiy et al., 2020), derived from the ZINC Clean Leads collection (Sterling & Irwin, 2015), featuring molecules with 8 to 27 heavy atoms, filtered by specific criteria.

Then, we include the Guacamol benchmark (Brown et al., 2019), based on the ChEMBL 24 database (Mendez et al., 2019). This dataset comprises synthesized molecules, tested against biological targets, with sizes ranging from 2 to 88 heavy atoms.

Lastly, we also include the ZINC250k dataset (Sterling & Irwin, 2015), which contains 249,455 molecules with up to 38 heavy atoms from 9 element types. We evaluate DeFoG's performance under the same setting of previous works (Jo et al., 2024).

**Metrics**   For the QM9 dataset, we follow the dataset splits and evaluation metrics outlined by Vignac et al. (2022). For the Moses and Guacamol benchmarks, we adhere to the training setups and evaluation metrics proposed by Polykovskiy et al. (2020) and Brown et al. (2019), respectively. Note that Guacamol includes molecules with charges; therefore, the generated graphs are converted to charged molecules based on the relaxed validity criterion used by Jo et al. (2022) before being translated to their corresponding SMILES representations. The validity, uniqueness, and novelty metrics reported by the Guacamol benchmark are actually V, V.U., and V.U.N., and are referred to directly as V, V.U., and V.U.N. in the table for clarity. For ZINC250k, we adopt the standard evaluation metrics commonly used for this benchmark (see, e.g., Jo et al. (2024); Eijkelboom et al. (2024)), which comprise a subset of the metrics described above.

### F.2.3. DIGITAL PATHOLOGY DATASETS

**Description**   Graphs, with their natural ability to represent relational data, are widely used to capture spatial biological dependencies in tissue images. This approach has proven successful in digital pathology tasks such as microenvironment classification (Wu et al., 2022), cancer classification (Pati et al., 2022), and decision explainability (Jaume et al., 2020). More recently, graph-based methods have been applied to generative tasks (Madeira et al., 2023), and an open-source dataset was made available by Madeira et al. (2024). This dataset consists of cell graphs where the nodes represent biological cells, categorized into 9 distinct cell types (node classes), and edges model local cell-cell interactions (a single class). For further details, refer to Madeira et al. (2024).

**Metrics**   Each cell graph in the dataset can be mapped to a TLS (Tertiary Lymphoid Structure) embedding, denoted as $\kappa = [\kappa_0, \ldots, \kappa_5] \in \mathbb{R}^6$, which quantifies its TLS content. A graph $G$ is classified as having low TLS content if $\kappa_1(G) < 0.05$, and high TLS content if $\kappa_2(G) > 0.05$. Based on these criteria, the dataset is split into two subsets: high TLS and low TLS. In prior work, TLS generation accuracy was evaluated by training generative models on these subsets separately, and verifying if the generated graphs matched the corresponding TLS content label. We compute TLS accuracy as the average accuracy across both subsets. For DeFoG, we conditionally train it on both subsets simultaneously, as described in Appendix E, and compute TLS accuracy based on whether the generated graphs adhere to the conditioning label. Additionally, we report the V.U.N. metric (valid, unique, novel), similar to what is done for the synthetic datasets (see Appendix F.2.1). A graph is considered valid in this case if it is a connected planar graph, as the graphs in these datasets were constructed using Delaunay triangulation.

Table 4: Training and sampling time on each dataset.

| Dataset | Min Nodes | Max Nodes | Training Time (h) | Graphs Sampled | Sampling Time (h) |
|---------|-----------|-----------|-------------------|----------------|-------------------|
| Planar | 64 | 64 | 29 | 40 | 0.07 |
| Tree | 64 | 64 | 8 | 40 | 0.07 |
| SBM | 44 | 187 | 75 | 40 | 0.07 |
| QM9 | 2 | 9 | 6.5 | 10000 | 0.2 |
| QM9(H) | 3 | 29 | 55 | 10000 | 0.4 |
| Moses | 8 | 27 | 46 | 25000 | 5 |
| Guacamol | 2 | 88 | 141 | 10000 | 7 |
| TLS | 20 | 81 | 38 | 80 | 0.15 |
| ZINC250k | 6 | 38 | 14 | 10000 | 4.8 |

Table 5: Training and sampling parameters for full-step sampling (500 or 1000 steps for molecular and synthetic datasets, respectively).

| | Train | | Sampling | | | Conditional |
|---------|---------------------|-----------------|-------------------|------------------------|----------------------|--------|
| Dataset | Initial Distribution | Train Distortion | Sample Distortion | $\omega$ (Target Guidance) | $\eta$ (Stochasticity) | $\gamma$ |
| Planar | Marginal | Identity | Polydec | 0.05 | 50 | — |
| Tree | Marginal | Polydec | Polydec | 0.0 | 0.0 | — |
| SBM | Absorbing | Identity | Identity | 0.0 | 0.0 | — |
| QM9 | Marginal | Identity | Polydec | 0.0 | 0.0 | — |
| QM9(H) | Marginal | Identity | Polydec | 0.05 | 0.0 | — |
| Moses | Marginal | Polydec | Polydec | 0.5 | 200 | — |
| Guacamol | Marginal | Polydec | Polydec | 0.1 | 300 | — |
| TLS | Marginal | Identity | Polydec | 0.05 | 0.0 | 2.0 |
| ZINC250k | Marginal | Identity | Polydec | 0.0 | 200 | — |

### F.3. Resources

The training and sampling times for the different datasets explored in this paper are provided in Tab. 4. All the experiments in this work were run on a single NVIDIA A100-SXM4-80GB GPU.

DeFoG's memory usage matches existing diffusion models, with quadratic complexity in node number due to complete-graph modeling. Rate matrix overhead is negligible, and RRWP features are more efficient to compute than previous alternatives (Vignac et al., 2022; Xu et al., 2024), as shown in Appendix G.4.

### F.4. Hyperparameter Tuning

The default hyperparameters for training and sampling for each dataset can be found in the provided code repository. In Tab. 5, we specifically highlight their values for the proposed training and sampling strategies (Sec. 3 and Appendix C.1), and conditional guidance parameter (see Appendix E). As the training process is by far the most computationally costly stage, we aim to minimize changes to the default model training configuration. Nevertheless, we demonstrate the effectiveness of these modifications on certain datasets:

1. SBM performs particularly well with absorbing distributions, likely due to its distinct clustering structure, which differs from other graph properties. Additionally, when tested with a marginal model, SBM can achieve a V.U.N. of 80.5% and an average ratio of 2.5, which also reaches state-of-the-art performance.

2. Guacamol and MOSES are trained directly with polydec distortion to accelerate convergence, as these datasets are very large and typically require a significantly longer training period.

3. For the tree dataset, standard training yielded suboptimal results (85.3% for V.U.N. and 1.8 for average ratio). However, a quick re-training using polydec distortion achieved state-of-the-art performance with 7 hours of training.

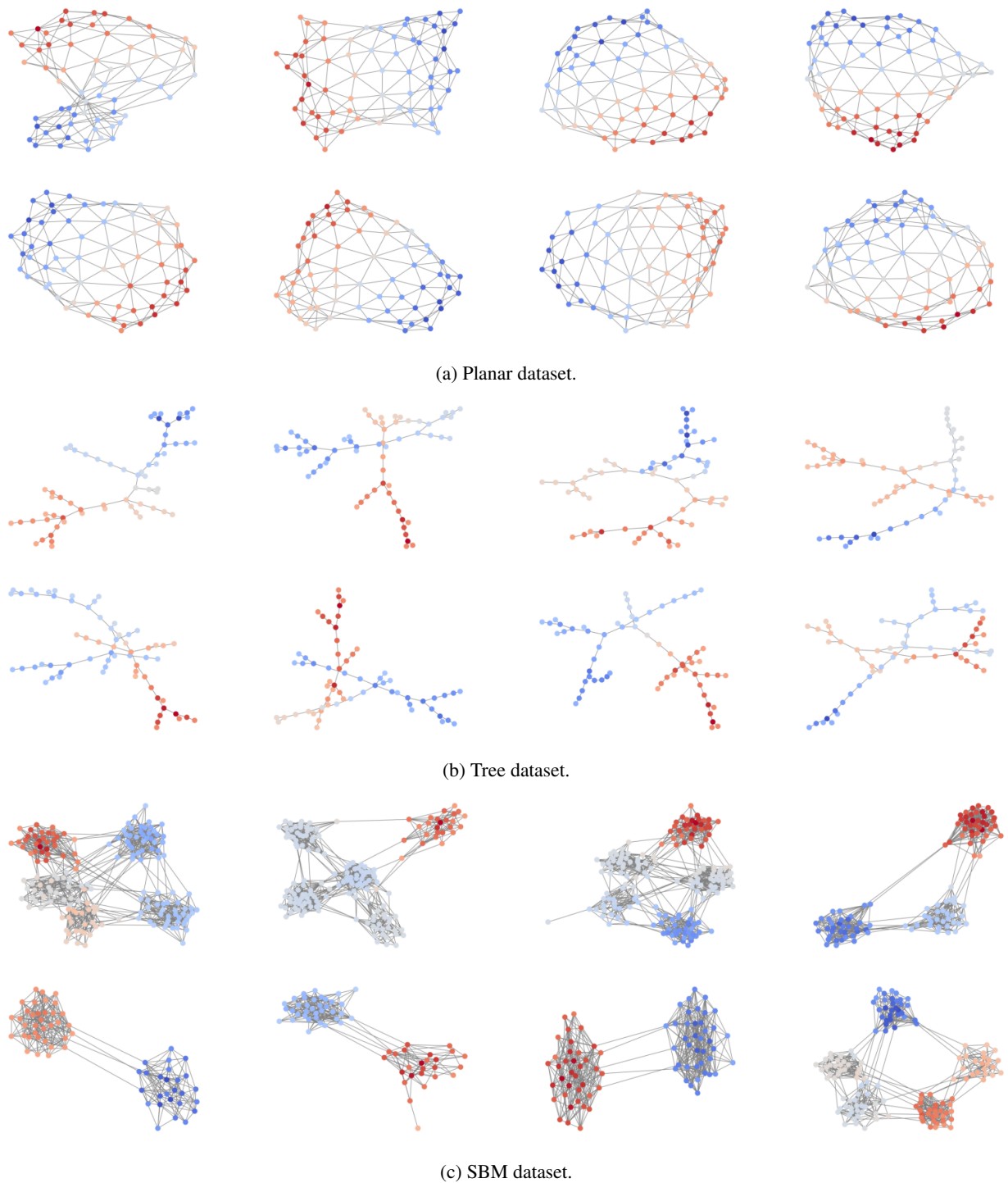

(a) Planar dataset.

(b) Tree dataset.

(c) SBM dataset.

Figure 16: Uncurated set of dataset graphs (top) and generated graphs by DeFoG (bottom) for the synthetic datasets.

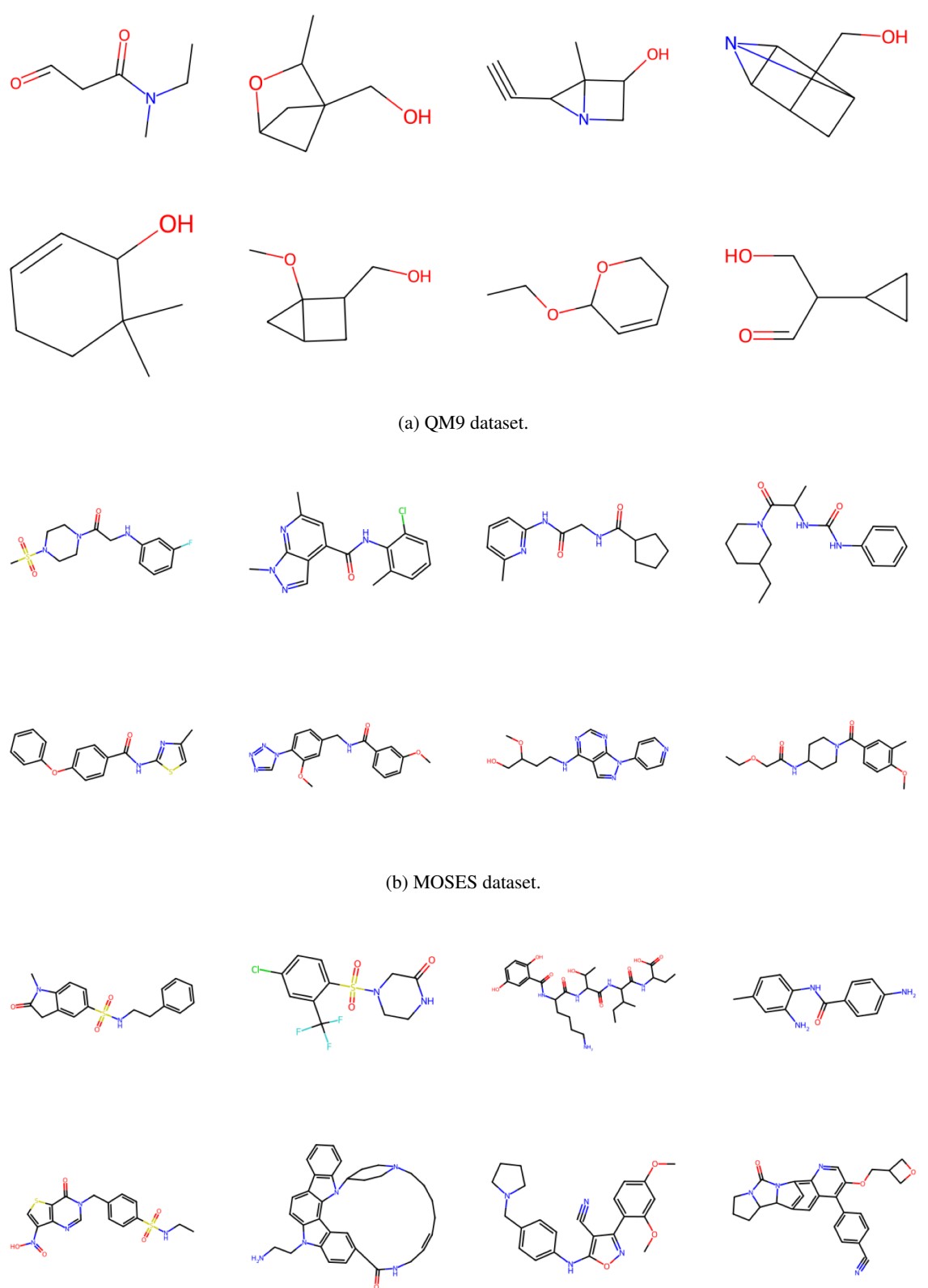

(a) QM9 dataset.

(b) MOSES dataset.

(c) Guacamol dataset.

Figure 17: Uncurated set of dataset graphs (top) and generated graphs by DeFoG (bottom) for the molecular datasets.

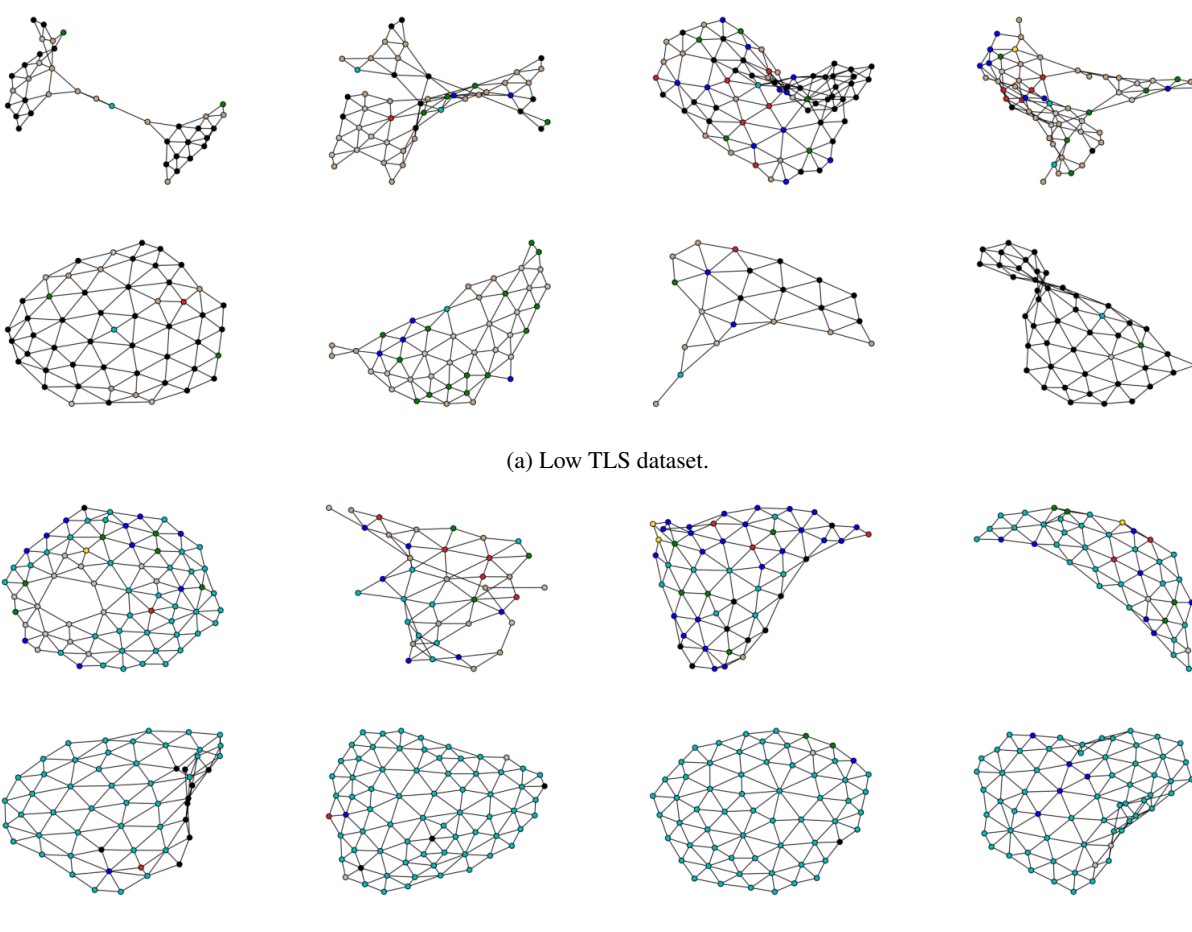

(a) Low TLS dataset.

(b) High TLS dataset.

Figure 18: Uncurated set of dataset graphs (top) and generated graphs by DeFoG (bottom) for the TLS dataset.

# G. Additional Results

First, we discuss conditional generation for real-world digital pathology graph generation. Next, we provide comprehensive tables detailing results on synthetic datasets in Appendix G.2. Then, Appendix G.3 focuses on molecular generation tasks, including results for the QM9 dataset and the full tables for MOSES and Guacamol. Finally, in Appendix G.4, we analyze the time complexity of various additional features that enhance expressivity in graph diffusion models.

## G.1. Conditional Generation

**Setup** Tertiary Lymphoid Structure (TLS) graph datasets have been recently released with graphs built from digital pathology data (Madeira et al., 2024). These graphs are split into two subsets — low TLS and high TLS — based on their TLS content, a biologically informed metric reflecting the cell organization in the graph structure. Previously, models were trained and evaluated separately for each subset. To demonstrate the flexibility of DeFoG, we conditionally train it across both datasets simultaneously, using the low/high TLS content as a binary label for each graph. More details on the used conditional framework in Appendix E.

Table 6: TLS conditional generation results.

| Model | TLS Dataset | |
| --- | --- | --- |
| | V.U.N. ↑ | TLS Val. ↑ |
| Train set | 0.0 | 100 |
| GraphGen (Goyal et al., 2020) | 40.2±3.8 | 25.1±1.2 |
| BiGG (Dai et al., 2020) | 0.6±0.4 | 16.7±1.6 |
| SPECTRE (Martinkus et al., 2022) | 7.9±1.3 | 25.3±0.8 |
| DiGress+ (Madeira et al., 2024) | 13.2±3.4 | 12.6±3.0 |
| ConStruct (Madeira et al., 2024) | **99.1**±1.1 | 92.1±1.3 |
| DeFoG (# steps = 50) | 44.5±4.2 | 93.0±5.6 |
| DeFoG (# steps = 1,000) | 94.5±1.8 | **95.8**±1.5 |

We evaluate two main aspects: first, how frequently the conditionally generated graphs align with the provided labels (TLS Validity); and, second, the validity, uniqueness, and novelty of the generated graphs (V.U.N.). Graphs are considered valid if they are planar and connected. For comparison, we report the average results of existing models across the two subsets, as they were not trained conditionally.

**Results** From Tab. 6, DeFoG significantly outperforms the unconstrained models (all but ConStruct). Notably, we outperform ConStruct on TLS validity with even 50 steps. For V.U.N., while ConStruct is hard-constrained to achieve 100% graph planarity, making it strongly biased toward high validity, DeFoG remarkably approaches these values without relying on such rigid constraints.

## G.2. Synthetic Graph Generation

In Tab. 7, we present the full results for DeFoG for the three different datasets: planar, tree, and SBM.

Table 7: Graph Generation Performance on Synthetic Graphs. We present the results of DeFoG across five sampling runs, each generating 40 graphs, reported as mean ± standard deviation. The remaining values are obtained from Bergmeister et al. (2023). Additionally, we include results for Cometh (Siraudin et al., 2024) and DisCo (Xu et al., 2024). For the average ratio computation, we adhere to the method outlined by (Bergmeister et al., 2023), excluding statistics where the training set MMD is zero.

| | Planar Dataset | | | | | | | | | |
| Model | Deg.↓ | Clus.↓ | Orbit↓ | Spec.↓ | Wavelet↓ | Ratio↓ | Valid↑ | Unique↑ | Novel↑ | V.U.N.↑ |
|---|---|---|---|---|---|---|---|---|---|---|
| Train set | 0.0002 | 0.0310 | 0.0005 | 0.0038 | 0.0012 | 1.0 | 100 | 100 | 0.0 | 0.0 |
| GraphRNN (You et al., 2018) | 0.0049 | 0.2779 | 1.2543 | 0.0459 | 0.1034 | 490.2 | 0.0 | 100 | 100 | 0.0 |
| GRAN (Liao et al., 2019) | 0.0007 | 0.0426 | 0.0009 | 0.0075 | 0.0019 | 2.0 | 97.5 | 85.0 | 2.5 | 0.0 |
| SPECTRE (Martinkus et al., 2022) | 0.0005 | 0.0785 | 0.0012 | 0.0112 | 0.0059 | 3.0 | 25.0 | 100 | 100 | 25.0 |
| DiGress (Vignac et al., 2022) | 0.0007 | 0.0780 | 0.0079 | 0.0098 | 0.0031 | 5.1 | 77.5 | 100 | 100 | 77.5 |
| EDGE (Chen et al., 2023) | 0.0761 | 0.3229 | 0.7737 | 0.0957 | 0.3627 | 431.4 | 0.0 | 100 | 100 | 0.0 |
| BwR (Diamant et al., 2023) | 0.0231 | 0.2596 | 0.5473 | 0.0444 | 0.1314 | 251.9 | 0.0 | 100 | 100 | 0.0 |
| BiGG (Dai et al., 2020) | 0.0007 | 0.0570 | 0.0367 | 0.0105 | 0.0052 | 16.0 | 62.5 | 85.0 | 42.5 | 5.0 |
| GraphGen (Goyal et al., 2020) | 0.0328 | 0.2106 | 0.4236 | 0.0430 | 0.0989 | 210.3 | 7.5 | 100 | 100 | 7.5 |
| HSpectre (one-shot) (Bergmeister et al., 2023) | 0.0003 | 0.0245 | 0.0006 | 0.0104 | 0.0030 | 1.7 | 67.5 | 100 | 100 | 67.5 |
| HSpectre (Bergmeister et al., 2023) | 0.0005 | 0.0626 | 0.0017 | 0.0075 | 0.0013 | 2.1 | 95.0 | 100 | 100 | 95.0 |
| GruM (Jo et al., 2024) | 0.0004 | 0.0301 | 0.0002 | 0.0104 | 0.0020 | 1.8 | — | — | — | 90.0 |
| CatFlow (Eijkelboom et al., 2024) | 0.0003 | 0.0403 | 0.0008 | — | — | — | — | — | — | 80.0 |
| DisCo (Xu et al., 2024) | 0.0002 ±0.0001 | 0.0403 ±0.0155 | 0.0009 ±0.0004 | — | — | — | 83.6 ±2.1 | 100.0 ±0.0 | 100.0 ±0.0 | 83.6 ±2.1 |
| Cometh - PC (Siraudin et al., 2024) | 0.0006 ±0.0005 | 0.0434 ±0.0093 | 0.0016 ±0.0006 | 0.0049 ±0.0008 | — | — | 99.5 ±0.9 | 100.0 ±0.0 | 100.0 ±0.0 | **99.5** ±0.9 |
| DeFoG | 0.0005 ±0.0002 | 0.0501 ±0.0149 | 0.0006 ±0.0004 | 0.0072 ±0.0011 | 0.0014 ±0.0002 | **1.6** ±0.4 | 99.5 ±1.0 | 100.0 ±0.0 | 100.0 ±0.0 | **99.5** ±1.0 |

| | Tree Dataset | | | | | | | | | |
| Train set | 0.0001 | 0.0000 | 0.0000 | 0.0075 | 0.0030 | 1.0 | 100 | 100 | 0.0 | 0.0 |
|---|---|---|---|---|---|---|---|---|---|---|
| GRAN (Liao et al., 2019) | 0.1884 | 0.0080 | 0.0199 | 0.2751 | 0.3274 | 607.0 | 0.0 | 100 | 100 | 0.0 |
| DiGress (Vignac et al., 2022) | 0.0002 | 0.0000 | 0.0000 | 0.0113 | 0.0043 | **1.6** | 90.0 | 100 | 100 | 90.0 |
| EDGE (Chen et al., 2023) | 0.2678 | 0.0000 | 0.7357 | 0.2247 | 0.4230 | 850.7 | 0.0 | 7.5 | 100 | 0.0 |
| BwR (Diamant et al., 2023) | 0.0016 | 0.1239 | 0.0003 | 0.0480 | 0.0388 | 11.4 | 0.0 | 100 | 100 | 0.0 |
| BiGG (Dai et al., 2020) | 0.0014 | 0.0000 | 0.0000 | 0.0119 | 0.0058 | 5.2 | 100 | 87.5 | 50.0 | 75.0 |
| GraphGen (Goyal et al., 2020) | 0.0105 | 0.0000 | 0.0000 | 0.0153 | 0.0122 | 33.2 | 95.0 | 100 | 100 | 95.0 |
| HSpectre (one-shot) (Bergmeister et al., 2023) | 0.0004 | 0.0000 | 0.0000 | 0.0080 | 0.0055 | 2.1 | 82.5 | 100 | 100 | 82.5 |
| HSpectre (Bergmeister et al., 2023) | 0.0001 | 0.0000 | 0.0000 | 0.0117 | 0.0047 | 4.0 | 100 | 100 | 100 | **100** |
| DeFoG | 0.0002 ±0.0001 | 0.0000 ±0.0000 | 0.0000 ±0.0000 | 0.0108 ±0.0028 | 0.0046 ±0.0004 | **1.6** ±0.4 | 96.5 ±2.6 | 100.0 ±0.0 | 100.0 ±0.0 | 96.5 ±2.6 |

| | Stochastic Block Model ($n_{max} = 187$, $n_{avg} = 104$) | | | | | | | | | |
| Model | Deg.↓ | Clus.↓ | Orbit↓ | Spec.↓ | Wavelet↓ | Ratio↓ | Valid↑ | Unique↑ | Novel↑ | V.U.N.↑ |
|---|---|---|---|---|---|---|---|---|---|---|
| Training set | 0.0008 | 0.0332 | 0.0255 | 0.0027 | 0.0007 | 1.0 | 85.9 | 100 | 0.0 | 0.0 |
| GraphRNN (You et al., 2018) | 0.0055 | 0.0584 | 0.0785 | 0.0065 | 0.0431 | 14.7 | 5.0 | 100 | 100 | 5.0 |
| GRAN (Liao et al., 2019) | 0.0113 | 0.0553 | 0.0540 | 0.0054 | 0.0212 | 9.7 | 25.0 | 100 | 100 | 25.0 |
| SPECTRE (Martinkus et al., 2022) | 0.0015 | 0.0521 | 0.0412 | 0.0056 | 0.0028 | 2.2 | 52.5 | 100 | 100 | 52.5 |
| DiGress (Vignac et al., 2022) | 0.0018 | 0.0485 | 0.0415 | 0.0045 | 0.0014 | 1.7 | 60.0 | 100 | 100 | 60.0 |
| EDGE (Chen et al., 2023) | 0.0279 | 0.1113 | 0.0854 | 0.0251 | 0.1500 | 51.4 | 0.0 | 100 | 100 | 0.0 |
| BwR (Diamant et al., 2023) | 0.0478 | 0.0638 | 0.1139 | 0.0169 | 0.0894 | 38.6 | 7.5 | 100 | 100 | 7.5 |
| BiGG (Dai et al., 2020) | 0.0012 | 0.0604 | 0.0667 | 0.0059 | 0.0370 | 11.9 | 10.0 | 100 | 100 | 10.0 |
| GraphGen (Goyal et al., 2020) | 0.0550 | 0.0623 | 0.1189 | 0.0182 | 0.1193 | 48.8 | 5.0 | 100 | 100 | 5.0 |
| HSpectre (one-shot) (Bergmeister et al., 2023) | 0.0141 | 0.0528 | 0.0809 | 0.0071 | 0.0205 | 10.5 | 75.0 | 100 | 100 | 75.0 |
| HSpectre (Bergmeister et al., 2023) | 0.0119 | 0.0517 | 0.0669 | 0.0067 | 0.0219 | 10.2 | 45.0 | 100 | 100 | 45.0 |
| GruM (Jo et al., 2024) | 0.0015 | 0.0589 | 0.0450 | 0.0077 | 0.0012 | **1.1** | — | — | — | 85.0 |
| CatFlow (Eijkelboom et al., 2024) | 0.0012 | 0.0498 | 0.0357 | — | — | — | — | — | — | 85.0 |
| DisCo (Xu et al., 2024) | 0.0006 ±0.0002 | 0.0266 ±0.0133 | 0.0510 ±0.0128 | — | — | — | 66.2 ±1.4 | 100.0 ±0.0 | 100.0 ±0.0 | 66.2 ±1.4 |
| Cometh (Siraudin et al., 2024) | 0.0020 ±0.0003 | 0.0498 ±0.0000 | 0.0383 ±0.0051 | 0.0024 ±0.0003 | — | — | 75.0 ±3.7 | 100.0 ±0.0 | 100.0 ±0.0 | 75.0 ±3.7 |
| DeFoG | 0.0006 ±0.0023 | 0.0517 ±0.0012 | 0.0556 ±0.0739 | 0.0054 ±0.0012 | 0.0080 ±0.0024 | 4.9 ±1.3 | 90.0 ±5.1 | 90.0 ±5.1 | 90.0 ±5.1 | **90.0** ±5.1 |

## G.3. Molecular Graph Generation

For the molecular generation tasks, we begin by examining the results for QM9, considering both implicit and explicit hydrogens (Vignac et al., 2022). In the implicit case, hydrogen atoms are inferred to complete the valencies, while in the explicit case, hydrogens must be explicitly modeled, making it an inherently more challenging task. The results are presented in Tab. 8. Notably, DeFoG achieves training set validity in both scenarios, representing the theoretical maximum. Furthermore, DeFoG consistently outperforms other models in terms of FCD. Remarkably, even with only 10% of the sampling steps, DeFoG surpasses many existing methods.

Table 8: Molecule generation on QM9. We present the results over five sampling runs of 10000 generated graphs each, in the format mean ± standard deviation. We include the results of Relaxed Validity, which accounts for charged molecules to facilitate comparison, as different methods may report varying types of validity.

| Model | Without Explicit Hydrogenes | | | | With Explicit Hydrogenes | | | |
|---|---|---|---|---|---|---|---|---|
| | Valid ↑ | Relaxed Valid ↑ | Unique ↑ | FCD ↓ | Valid ↑ | Relaxed Valid ↑ | Unique ↑ | FCD ↓ |
| Training set | 99.3 | 99.5 | 99.2 | 0.03 | 97.8 | 98.9 | 99.9 | 0.01 |
| SPECTRE (Martinkus et al., 2022) | 87.3 | — | 35.7 | — | — | — | — | — |
| GraphNVP (Madhawa et al., 2019) | 83.1 | — | 99.2 | — | — | — | — | — |
| GDSS (Jo et al., 2022) | 95.7 | — | 98.5 | 2.9 | — | — | — | — |
| DiGress (Vignac et al., 2022) | $99.0_{\pm0.0}$ | — | $96.2_{\pm0.1}$ | — | $95.4_{\pm1.1}$ | — | $\mathbf{97.6_{\pm0.4}}$ | — |
| GruM(Jo et al., 2024) | 99.2 | — | 96.7 | **0.11** | — | — | — | — |
| CatFlow(Eijkelboom et al., 2024) | — | **99.8** | **100.0** | 0.44 | — | — | — | — |
| DisCo (Xu et al., 2024) | $99.3_{\pm0.6}$ | — | — | — | — | — | — | — |
| Cometh (Siraudin et al., 2024) | $\mathbf{99.6_{\pm0.1}}$ | — | $96.8_{\pm0.2}$ | $0.25_{\pm0.01}$ | — | — | — | — |
| DeFoG (# sampling steps = 50) | $98.9_{\pm0.1}$ | $99.2_{\pm0.0}$ | $96.2_{\pm0.2}$ | $0.26_{\pm0.00}$ | $\underline{97.1_{\pm0.0}}$ | $\underline{98.1_{\pm0.0}}$ | $94.8_{\pm0.0}$ | $\underline{0.31_{\pm0.00}}$ |
| DeFoG (# sampling steps = 500) | $\underline{99.3_{\pm0.0}}$ | $\underline{99.4_{\pm0.1}}$ | $96.3_{\pm0.3}$ | $\underline{0.12_{\pm0.00}}$ | $\mathbf{98.0_{\pm0.0}}$ | $\mathbf{98.8_{\pm0.0}}$ | $\underline{96.7_{\pm0.0}}$ | $\mathbf{0.05_{\pm0.00}}$ |

Additionally, we provide the complete version of Tab. 2, presenting the results for MOSES, Guacamol and ZINC250k datasets separately in Tab. 9, Tab. 10 and Tab. 11, respectively. We include models from classes beyond diffusion models to better contextualize the performance achieved by DeFoG. We analyze the performance of diffusion and flow-based methods on MOSES and Guacamol in the main paper (see Sec. 5.1). Here, we focus on the same analysis for ZINC250k. As shown in Tab. 11, DeFoG achieves state-of-the-art performance on this benchmark. Notably, it also attains superior FCD scores with only 50 sampling steps, surpassing existing diffusion and flow-based methods.

Table 9: Molecule generation on MOSES.

| Model | Class | Val.↑ | Unique. ↑ | Novelty↑ | Filters ↑ | FCD ↓ | SNN ↑ | Scaf ↑ |
|---|---|---|---|---|---|---|---|---|
| Training set | — | 100.0 | 100.0 | 0.0 | 100.0 | 0.01 | 0.64 | 99.1 |
| VAE (Kingma & Welling, 2013) | Smiles | 97.7 | 99.8 | 69.5 | **99.7** | **0.57** | **0.58** | 5.9 |
| JT-VAE (Jin et al., 2018) | Fragment | **100.0** | **100.0** | **99.9** | 97.8 | 1.00 | 0.53 | 10.0 |
| GraphInvent (Mercado et al., 2021) | Autoreg. | 96.4 | 99.8 | —- | 95.0 | 1.22 | 0.54 | 12.7 |
| DiGress (Vignac et al., 2022) | One-shot | 85.7 | **100.0** | 95.0 | 97.1 | 1.19 | 0.52 | 14.8 |
| DisCo (Xu et al., 2024) | One-shot | 88.3 | **100.0** | 97.7 | 95.6 | 1.44 | 0.50 | 15.1 |
| Cometh (Siraudin et al., 2024) | One-shot | 90.5 | 99.9 | 92.6 | 99.1 | 1.27 | 0.54 | 16.0 |
| DeFoG (# sampling steps = 50) | One-shot | 83.9 | 99.9 | 96.9 | 96.5 | 1.87 | 0.50 | **23.5** |
| DeFoG (# sampling steps = 500) | One-shot | 92.8 | 99.9 | 92.1 | 98.9 | 1.95 | 0.55 | 14.4 |

Table 10: Molecule generation on GuacaMol. We present the results over five sampling runs of 10000 generated graphs each, in the format mean ± standard deviation.

| Model | Class | Val. ↑ | V.U. ↑ | V.U.N. ↑ | KL div↑ | FCD↑ |
|---|---|---|---|---|---|---|
| Training set | — | 100.0 | 100.0 | 0.0 | 99.9 | 92.8 |
| LSTM (Hochreiter & Schmidhuber, 1997) | Smiles | 95.9 | 95.9 | 87.4 | **99.1** | **91.3** |
| NAGVAE (Kwon et al., 2020) | One-shot | 92.9 | 88.7 | 88.7 | 38.4 | 0.9 |
| MCTS (Brown et al., 2019) | One-shot | **100.0** | **100.0** | 95.4 | 82.2 | 1.5 |
| DiGress (Vignac et al., 2022) | One-shot | 85.2 | 85.2 | 85.1 | 92.9 | 68.0 |
| DisCo (Xu et al., 2024) | One-shot | 86.6 | 86.6 | 86.5 | 92.6 | 59.7 |
| Cometh (Siraudin et al., 2024) | One-shot | 98.9 | 98.9 | 97.6 | 96.7 | 72.7 |
| DeFoG (# steps = 50) | One-shot | 91.7 | 91.7 | 91.2 | 92.3 | 57.9 |
| DeFoG (# steps = 500) | One-shot | 99.0 | 99.0 | **97.9** | 97.7 | 73.8 |

Table 11: Molecular generation on ZINC250k dataset.

| Model | Val. ↑ | Uniqueness ↑ | FCD ↓ | NSPDK ↓ | Scaffold ↑ |
|---|---|---|---|---|---|
| GruM (Jo et al., 2024) | 98.65 | – | 2.257 | 0.0015 | 0.5299 |
| GBD (Liu et al., 2024) | 97.87 | – | 2.248 | 0.0018 | 0.5042 |
| CatFlow (Eijkelboom et al., 2024) | 99.21 | **100.00** | 13.211 | – | – |
| DeFoG (50 steps) | 96.65±0.16 | 99.99±0.01 | 2.123±0.029 | 0.0022±0.0001 | 0.4245±0.0109 |
| DeFoG | **99.22**±0.08 | 99.99±0.01 | **1.425**±0.022 | **0.0008**±0.0001 | **0.5903**±0.0099 |

Table 12: Performance comparison of RRWP-based graph encoding within the DiGress framework.

| Method | Planar | | SBM | | QM9 | | |
|---|---|---|---|---|---|---|---|
| | V.U.N. ↑ | Ratio ↓ | V.U.N. ↑ | Ratio ↓ | Valid ↑ | Unique ↑ | FCD ↓ |
| DiGress | 77.5 | 5.1 | 60.0 | 1.7 | $99.0 \pm 0.0$ | $96.2 \pm 0.1$ | - |
| DiGress (RRWP) | 90.0 | 4.0 | 70.0 | 1.7 | $99.1 \pm 0.1$ | $96.6 \pm 0.2$ | - |
| DeFoG (RRWP, 50 steps) | $95.0 \pm 3.2$ | $3.2 \pm 1.1$ | $86.5 \pm 5.3$ | $2.2 \pm 0.3$ | $98.9 \pm 0.1$ | $96.2 \pm 0.2$ | $0.26 \pm 0.00$ |
| DeFoG (RRWP) | $99.5 \pm 1.0$ | $1.6 \pm 0.4$ | $90.0 \pm 5.1$ | $4.9 \pm 1.3$ | $99.3 \pm 0.0$ | $96.3 \pm 0.3$ | $0.12 \pm 0.00$ |

Table 13: Computation time for different additional features. The RRWP features are computed with 12 steps.

| Dataset | Min Nodes | Max Nodes | RRWP (ms) | Cycles (ms) | Spectral (ms) |
|---|---|---|---|---|---|
| Moses | 8 | 27 | 0.6 | 1.2 | 2.5 |
| Planar | 64 | 64 | 0.5 | 1.2 | 93.0 |
| SBM | 44 | 187 | 0.6 | 2.7 | 146.9 |

### G.4. Impact of Additional Features

In graph diffusion methods, the task of graph generation is decomposed into a mapping of a graph to a set of marginal probabilities for each node and edge. This problem is typically addressed using a Graph Transformer architecture, which is augmented with additional features to capture structural aspects that the base architecture might struggle to model effectively (Vignac et al., 2022; Xu et al., 2024; Siraudin et al., 2024) otherwise.

In this section, we evaluate the impact of using RRWP encodings as opposed to the spectral and cycle encodings (up to 6-cycles) proposed in DiGress (Vignac et al., 2022).

In Tab. 12, we present a performance comparison of these two variants across three datasets: QM9, Planar, and SBM. The results show that RRWP achieves comparable or superior performance within the DiGress framework, validating its effectiveness as a graph encoding method. Notably, despite these improvements, DiGress's performance remains significantly below that of DeFoG on the Planar and SBM datasets, while achieving similar validity and uniqueness on the QM9 dataset.

To further demonstrate the impact of using RRWP on sampling efficiency, we compare the performances of DeFoG, DiGress, and DiGress augmented with RRWP (replacing the original additional features) across a varying number of sampling steps. These results are shown in Figure 19.

We observe that, while RRWP provides improvements on the Planar and SBM datasets with fewer generation steps, it is still significantly outperformed by the optimized DeFoG framework. This highlights that although RRWP is an efficient and effective graph encoding method, the primary performance gains of DeFoG stem from its continuous-time formulation featuring fully decoupled training and sampling stages.

We then perform a time complexity analysis of these methods. While both cycle and RRWP encodings primarily involve matrix multiplications, spectral encodings require more complex algorithms for eigenvalue and eigenvector computation. As shown in Tab. 13, cycle and RRWP encodings are more computationally efficient, particularly for larger graphs where eigenvalue computation becomes increasingly costly. These results also support the use of RRWP encodings over the combined utilization of cycle and spectral features.

For the graph sizes considered in this work, the additional feature computation time remains relatively small compared to the model's forward pass and backpropagation. However, as graph sizes increase - a direction beyond the scope of this paper - this computational gap could become significant, making RRWP a suitable encoding for scalable graph generative models.

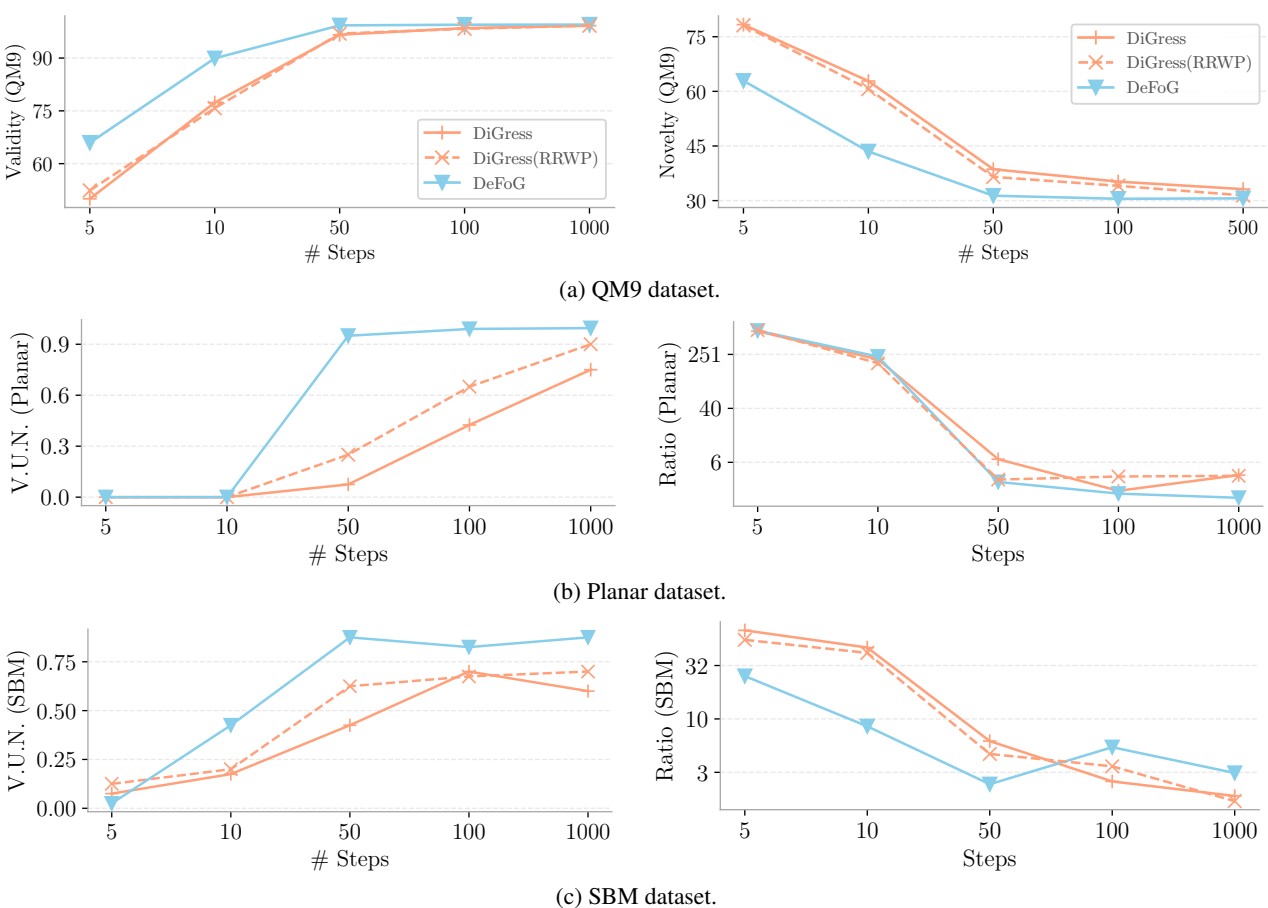

(a) QM9 dataset.

(b) Planar dataset.

(c) SBM dataset.

Figure 19: Impact of RRWP features for sampling efficiency.

