# OpenReview forum: "DeFoG: Discrete Flow Matching for Graph Generation"
_ICML.cc/2025/Conference — ICML 2025 oral_

### Official Review · Reviewer_kK3Q · 2025-03-13

**Overall Recommendation:** 4

**Summary:**

This paper introduces **DeFoG**, a novel **graph generative framework** that decouples the **training and sampling** processes to improve efficiency and flexibility. The key innovation is the **discrete flow-matching (DFM) formulation**, which ensures **node permutation equivariance** and allows more expressive sampling methods tailored for graph structures. The paper provides **theoretical guarantees** linking training loss optimization to improved sampling dynamics. Extensive experiments across synthetic, molecular, and digital pathology datasets show that DeFoG achieves **state-of-the-art performance**, significantly reducing the required number of refinement steps compared to graph diffusion models.

The main contributions of the paper are:
- **A novel training-sampling disentanglement framework** for graph generative models.
- **Theoretical justification** demonstrating that DeFoG faithfully replicates ground truth graph distributions.
- **Exploration of novel sampling strategies**, improving efficiency while maintaining performance.
- **Empirical validation** across various datasets, demonstrating superior performance compared to existing diffusion models.

## update after rebuttal
The authors have addressed my concerns, so I raise my score.

**Claims And Evidence:**

The paper makes strong claims about:
1. **Training-Sampling Disentanglement**: Theoretical grounding is provided, but additional empirical comparisons with diffusion-based methods under different training configurations would strengthen the claim.
2. **Improved Sampling Efficiency**: The results show significant reductions in sampling steps (5–10% of diffusion models), but additional ablations on time-adaptive methods would clarify the practical efficiency gains.
3. **State-of-the-Art Performance**: DeFoG outperforms most existing diffusion models across various datasets. However, detailed breakdowns on dataset-specific improvements would be helpful.

Overall, the claims are well-supported, though further empirical comparisons and ablations could enhance the robustness of the results.

**Essential References Not Discussed:**

Some essential references about graph generative models are not discussed, such as the following references:

[1] Han X, Chen X, Ruiz F J R, et al. Fitting autoregressive graph generative models through maximum likelihood estimation[J]. Journal of Machine Learning Research, 2023, 24(97): 1-30.

[2] Xu M, Liu M, Jin W, et al. Graph and geometry generative modeling for drug discovery[C]//Proceedings of the 29th ACM SIGKDD Conference on Knowledge Discovery and Data Mining. 2023: 5833-5834.

**Experimental Designs Or Analyses:**

The experimental setup is well-structured, but there are some areas for improvement:
- **Scalability Analysis**: Given the paper's emphasis on scalability, it would be useful to include an explicit study on how DeFoG scales with increasing graph size.
- **Ablation Studies**: The ablation studies justify the importance of each component, but additional experiments on varying training configurations would be informative.
- **Generalization to Out-of-Distribution Data**: Evaluating how DeFoG performs on unseen graph structures would enhance its applicability.

**Methods And Evaluation Criteria:**

The proposed method and evaluation criteria make sense for the task:
- **Synthetic, molecular, and pathology datasets** are well-chosen, covering diverse graph generation scenarios.
- **Comparison with strong baselines**, including diffusion-based models, ensures fairness.
- **Metrics such as validity, uniqueness, and novelty** are appropriate for generative models.

However, additional analysis on the impact of different hyperparameters and computational trade-offs (e.g., memory and runtime comparisons) would be beneficial.

**Other Comments Or Suggestions:**

1. **Clarify the computational efficiency trade-offs**—while fewer sampling steps are needed, does DeFoG require significantly more compute per step?
2. **Include additional experiments on scalability**—how does DeFoG perform on very large graphs compared to diffusion models?
3. **Compare with other flow-based and autoregressive models**—this would help position DeFoG within the broader space of graph generation.

**Other Strengths And Weaknesses:**

### Strengths:
- **Novel formulation** that decouples training and sampling, offering greater flexibility.
- **Theoretical justification** enhances confidence in the approach.
- **Empirical validation across multiple domains**, demonstrating SOTA performance.

### Weaknesses:
- **Scalability analysis is missing**, despite claims of efficiency.
- **Limited analysis on generalization** beyond the tested datasets.
- **Additional hyperparameter studies** would further validate robustness.

**Questions For Authors:**

1. **How does DeFoG handle highly imbalanced graph structures?** Some datasets may have extreme degree distributions—does this affect the sampling process?
2. **What are the computational trade-offs of training-sampling decoupling?** While DeFoG reduces sampling steps, does it increase per-step complexity?
3. **How sensitive is DeFoG to hyperparameter choices?** Particularly in terms of the interpolation and rate matrices.

**Relation To Broader Scientific Literature:**

The paper is well-grounded in the existing literature on **graph generative models**, particularly diffusion-based methods. However, connections to recent advances in **graph normalizing flows and autoregressive models** could be further discussed.

**Theoretical Claims:**

The paper provides **mathematical derivations** linking the discrete flow-matching formulation to graph generation performance. The proof structure appears sound, but verifying its correctness in practical implementations (e.g., sensitivity to noise schedules) would further validate its robustness.

---

> ### Author Rebuttal · Authors · 2025-04-01
>
> We thank the reviewer for positively assessing our framework’s novelty, theoretical grounding, and empirical results. Below, we address the raised concerns:
>
> **I - Methods And Evaluation Criteria**:
> 1) *Hyperparameters sensitivity*: In the paper, we perform extensive hyperparameter sensitivity analyses covering $R^\omega$, $R^\text{DB}$, target guidance, stochasticity level, initial distributions, and training-sampling distortion (Sec. 5.3, App. B.1–B.4, C.1–C.2, Figs. 9, 10, 13, 14), as acknowledged by Reviewers 8xvx and 2q2N. These analyses sufficiently cover the key factors affecting generation and we provide supporting intuitions (Sec. 3.2, 5.3).
> 2) *Computational trade-offs*: We report training/sampling runtimes for DeFoG across datasets (App. F.3) and compare them with SOTA methods in Sec. 5.2. Note that step count scales linearly with runtime. Regarding memory usage, we propose to clarify in App. F.3:
> ```
> "DeFoG’s memory usage matches existing diffusion models, with quadratic complexity in node number due to complete-graph modeling. Rate matrix overhead is negligible, and RRWP features are more efficient to compute than previous alternatives [3,4]."
> ```
>
> **II - Theoretical Claims**:
> 1) Whether "noise schedules" refers to the choice of initial distribution or distortion functions, our theoretical results explicitly link training loss to sampling accuracy. Strong empirical results validate the practical relevance of these results, and the observed performance deterioration with fewer sampling steps confirms the theoretical predictions (Tables 1,2; Figs. 7,8,11,12).
>
> **III - Experimental Designs Or Analyses**:
> 1) *Scalability*: We believe our efficiency claims are well-supported: DeFoG reduces sampling steps to 5–10% of the original total (Tables 1,2) and uses more efficient additional features (Table 12). Moreover, DeFoG's memory usage matches existing diffusion models (see I.2), enabling similar graph scales as competing methods. Scaling these methods to larger graphs is an interesting and challenging problem by itself (e.g., [5] leverages sparsity to scale existing graph diffusion models, an approach also applicable to DeFoG), thus beyond this work's scope.
> 2) *Ablations*: See I.1.
> 3) *OOD Generalization*: Unlike conventional OOD tasks, graph generation typically aims to maintain distributional similarity [3,4] rather than handle distributional shifts. Thus, while an interesting direction, it is beyond our scope. For standard generalization, see VII-2.
>
> **IV - Supplementary Material**:
> 1) We ensure reproducibility via the provided code and detailed hyperparameters (App. B.4, F.4).
>
> **V - Broader Literature**:
> 1) *Graph flows/autoregressive models:* We extensively discuss key autoregressive models (GraphRNN, GRAN, BiGG, GraphGen) (Sec.4, App.A.1, Table 1). Regarding normalizing flows, we acknowledge their relevance (Sec. 4) but are unaware of recent graph-specific advances directly relevant to our work; any suggestions are welcome.
>
> **VI - Essential References**:
> 1) We included [1] in our related work, where we already discuss similar (autoregressive) models.
> 2) [2] is only a tutorial description on generative models for molecular graphs. Arguably, it is not an “essential reference'' on graph generative models. In case this was not a mistake, we kindly invite the reviewer to motivate this request.
>
> **VII - Other Strengths And Weaknesses**:
> 1) *Scalability*: See III.1.
> 2) *Generalization*: DeFoG generalizes across diverse domains (synthetic, molecular, digital pathology), going beyond typical benchmarks in graph generation studies [4,5]. The additional ZINC250k results requested by Reviewer 2q2N further support this, making our generalization analysis comprehensive.
> 3) *Hyperparameter studies*: See I.1
>
> **VIII - Other Comments or Suggestions**:
> 1) *Efficiency*: DeFoG incurs no additional overhead per sampling step (Alg. 2), and RRWP features are computed more efficiently than in prior work [3,4] (App. G.4).
> 2) *Scalability*: See III.1.
> 3) *Flow/autoregressive comparison*: See V.
>
> **IX - Questions For Authors**:
> 1) *Imbalanced graph structures*: DeFoG implicitly captures distributional heterogeneity, confirmed by superior performance on very diverse structural graph statistics (degree, clustering, orbits, spectral, wavelet distributions; e.g., Table 7) and datasets.
> 2) *Computational decoupling trade-offs*: Per-step sampling efficiency matches/exceeds existing models (See VIII.1). Additionally, training-sampling decoupling enables optional independent tuning, potentially further improving performance at minimal extra cost via an optimized hyperparameter selection pipeline (App.B.4).
> 3) *Hyperparameter sensitivity*: See I.1.
>
> [3] - Digress: Discrete denoising diffusion for graph generation, Vignac et al., ICLR 2023
>
> [4] - Discrete-state Continuous-time Diffusion for Graph Generation, Xu et al., NeurIPS 2024
>
> [5] - Sparse Training of Discrete Diffusion Models for Graph Generation, Qin et al., ArXiv 2023

---

> > ### Comment · Reviewer_kK3Q · 2025-04-06
> >
> > The authors have adequately addressed my concerns, and I am willing to raise my score.

---

### Official Review · Reviewer_2q2N · 2025-03-13

**Overall Recommendation:** 4

**Summary:**

This paper proposed a novel graph generative model via discrete flow matching. This framework provides flexible and efficient training and sampling methods. The paper also provides theoretical guarantee for this disentanglement framework. With rich empirical validation, the proposed DeFoG shows powerful modeling ability and robust generation quality.

**Claims And Evidence:**

The paper is easy to follow and extensive experiments demonstrate its effectiveness on graph modeling.

**Essential References Not Discussed:**

This paper has discussed many related works in diffusion / flow matching based graph generative models. Recently I notice an interesting graph modeling method, which applies beta diffusion on graph modeling. I think the author could discuss it a little bit because the beta diffusion can handle both of continuous and discrete elements in graph, while your method aims to disentangle the graph modeling process with flexible training and sampling method.
- Advancing Graph Generation through Beta Diffusion ICLR'25

**Experimental Designs Or Analyses:**

This paper provides rich experiments to validate its powerful graph modeling ability. The high V.U.N. performance on general graph shows its robustness in generation quality. This paper also conducted extensive ablation on training and sampling efficiency.

**Methods And Evaluation Criteria:**

The proposed method follows the widely used evaluation process.

**Other Comments Or Suggestions:**

How is the sampling distortion specific to the graph modeling, instead of usage in DFM.

**Other Strengths And Weaknesses:**

## Strengths
- Well discussion of related literature and connection to previous results.
- Important and useful empirical / theoretical results and their discussion.
- Extensive discussion about related works on continuous-time discrete diffusion and discrete flow matching.
- Sufficient analysis on sampling optimization.

## Weaknesses
- Could you provide some results on ZINC250k, I think this benchmark is important for molecule graph modeling task.
- Could you provide mean and standard deviation for synthetic graph generation. I'd like to know how stable the proposed method is.

**Questions For Authors:**

All of my questions are above.

**Relation To Broader Scientific Literature:**

Graph generation has always been a meaningful topic in machine learning / generation tasks and it has a broader impact on scientific discovery. This paper provides a practical implementation in graph modeling and theoretical foundation for this as well.

**Theoretical Claims:**

I verified that the sampling and training algorithms are consistent with Equations 4 and 5.
For the permutation invariance in graph modeling, I check the Appendix D.2.1 and the derivation is correct.

---

> ### Author Rebuttal · Authors · 2025-04-01
>
> We thank the reviewer for their thoughtful feedback. Below, we address the raised concerns in detail.
>
> **Essential Reference**:
> We thank the reviewer for proposing this interesting reference. We will expand our related work section to include a discussion over methods that support both continuous and discrete data. Specifically, we propose to integrate the following in the revised manuscript:
>
> ```
> Integrating continuous and categorical data within graph generative models is an important challenge, as many real-world applications involve heterogeneous data types (e.g., molecular graphs containing atomic coordinates alongside categorical atom and bond types). A recent example addressing this challenge is GBD [1], which incorporates beta diffusion to jointly model both continuous and discrete variables. Similarly, DeFoG is amenable to formulations involving mixed data types by leveraging an approach akin to MiDi [2], independently factorizing continuous and discrete variables. However, explicitly exploring this integration is beyond the scope of this work.
> ```
>
> **Results on ZINC250 dataset**:
> To further support our empirical findings on real-world datasets, we provide results on the ZINC250k. We evaluate DeFoG’s performance on this dataset after 14 hours of training under the same setting of previous works [3] and compare it, to the best of our knowledge, against the strongest-performing methods currently reported in the literature [1,3,4].
>
> | Model | Validity ($\uparrow$) | Uniqueness ($\uparrow$) | FCD ($\downarrow$) | NSPDK ($\downarrow$) | Scaffold ($\uparrow$) |
> |---|---|---|---|---|---|
> | GruM | 98.65 | - | 2.257 | 0.0015 | 0.5299 |
> | GBD | 97.87 | - | 2.248 | 0.0018 | 0.5042 |
> | CatFlow | 99.21 | **100.00** | 13.211 | - | - |
> | DeFoG (50 steps) | 96.65 ± 0.16 | 99.99 ± 0.01 | 2.123 ± 0.029 | 0.0022 ± 0.0001 | 0.4245 ± 0.0109 |
> | DeFoG | **99.22** ± 0.08 | 99.99 ± 0.01 | **1.425** ± 0.022 | **0.0008** ± 0.0001 | **0.5903** ± 0.0099 |
>
> DeFoG accomplishes state-of-the-art performance for this dataset. Notably, it also attains superior FCD performance using only 50 sampling steps, outperforming existing methods. We thank the reviewer for suggesting the ZINC250k benchmark. These results further demonstrate DeFoG’s generalization capabilities in molecular graph modeling, strengthening our contribution, and, thus, have been included into the updated manuscript.
>
> **Standard Deviation for Synthetic Graphs** In the Experiments section, we report VUN and the average ratio of MMDs for all the synthetic datasets, with both mean and standard deviation (std). We can observe DeFoG’s stable performance, in particular for a large number of steps. Additionally, we provide the full results for all the original graph-specific metrics, including Degree, Orbit, Wavelet, Spectral, and Cluster MMDs, with mean and std, in Table 7.
>
> **Sampling Distortion** The original DFM paper [6] considers evenly spaced sampling timesteps without any distortion. In contrast, we investigate how breaking this uniformity can yield more refined generative trajectories, especially at timesteps crucial for capturing specific graph properties.
> Our results (Sec. 5.3 – Sampling Distortion) demonstrate that, for graphs with strict structural constraints (e.g., planarity), it is beneficial to emphasize refinement at later steps, as categorical variables may abruptly transition as $t \rightarrow 1$, potentially violating the required structure. Refining these late steps thus helps detect and correct such errors. Conversely, for datasets without strict constraints (e.g., SBM), this refinement is not necessary.
> Moreover, we provide insights into the interplay between training and sampling distortions, observing that aligning these distortions typically yields the best generative performance, although this is not universally true. We also propose a simple heuristic to determine the optimal sampling distortion (see App C.2.).
> Finally, similar beneficial effects of distorted scheduling have been reported for other data modalities, such as image generation and language modeling [6].
>
> Overall, sampling distortion allows us to exploit DeFoG's training-sampling decoupling to further enhance generative performance by adapting the generative process to different graph characteristics, refining the corresponding crucial timesteps.
>
> [1] - Advancing Graph Generation through Beta Diffusion, Liu et al., ICLR 2025
>
> [2] - Midi: Mixed graph and 3d denoising diffusion for molecule generation, Vignac et al., ECML-PKDD 2023
>
> [3] - Graph Generation with Diffusion Mixture, Jo et al., ICML 2024
>
> [4] - Variational Flow Matching for Graph Generation, Eijkelboom et al., NeurIPS 2024
>
> [5] - Digress: Discrete denoising diffusion for graph generation, Vignac et al., ICLR 2023
>
> [6] - Generative Flows on Discrete State-Spaces: Enabling Multimodal Flows with Applications to Protein Co-Design, Campbell et al., ICML 2024
>
> [7] - Discrete Flow Matching, Gat et al., NeurIPS 2024

---

> > ### Comment · Reviewer_2q2N · 2025-04-04
> >
> > Thank you for your response, which has clarified several aspects of your work. In particular the 'Essential Reference' modeling discrete and continuous data at the same from a joint view or decomposition view is interesting. I will raise my score.

---

### Official Review · Reviewer_8xvx · 2025-03-16

**Overall Recommendation:** 4

**Summary:**

The authors adapt discrete flow matching for graph generation, replacing the usual SOTA discrete diffusion framework. This authors also utilize the flexibility of flow matching to further tune the sampling process to make it more efficient and generate higher quality samples in much fewer steps. This is primarily achieved by tuning the influence of target guidance and sample distortion (de-noising schedule)

**Claims And Evidence:**

The authors perform extensive benchmarks and ablation studies. They clearly support their claims of improved efficiency and sample quality.

**Essential References Not Discussed:**

References look good to me.

**Experimental Designs Or Analyses:**

All the experimental design is quite standard and looks good. The experiments and ablations are extensive and cover all introduced modifications

**Methods And Evaluation Criteria:**

Authors rely on well established benchmarks for graph generation with standard datasets and metrics that match best practices in the literature.

**Other Comments Or Suggestions:**

None

**Other Strengths And Weaknesses:**

I already covered the main strengths and weaknesses above. While novelty is not super high, as it's just a combination of existing ideas, not straying off the common path, I think it's a very valuable work in that it does a great job refining this current standard recipe and noticeably improving the resuls.

**Questions For Authors:**

None

**Relation To Broader Scientific Literature:**

The paper tackles a very important and widely studied problem of graph generation. It builds on the SOTA approaches (discrete diffusion) changing the diffusion framework with discrete flow matching. While the model is overall similar to previous discrete diffusion approaches and while discrete flow matching has been used in other contexts before, the multiple various minor improvements presented in the paper, do noticeably improve upon existing SOTA. The potential choices are extensively experimentally evaluated, which is very valuable for the community, extending our knowledge of how best to build these graph generative models in practice. If the paper comes with a nicely written and easy to work with and understand codebase I can see it becoming a go-to graph/molecule generative model people develop upon, replacing DiGress in this role, that is currently used as a main building block in variety of graph and molecule generation papers.

**Theoretical Claims:**

I didn't check the proofs/derivations in the appendix in detail, but the theory in the paper heavily relies on previous established works and looks sound.

---

> ### Author Rebuttal · Authors · 2025-03-31
>
> We thank the reviewer for their constructive feedback, as well as for recognizing the importance of our algorithmic improvements, the thoroughness of our experimental validation, and the practical value of our contributions to the graph generation community. Aligned with the reviewer's perspective, we are committed to releasing a clean, well-documented, and easy-to-use codebase upon publication to facilitate adoption and further developments by the community.
>
> We remain open to any further suggestions or questions that may arise.

---

### Official Review · Reviewer_kmNp · 2025-03-22

**Overall Recommendation:** 4

**Summary:**

The authors apply discrete flow matching to graph generation.

**Claims And Evidence:**

Claimed contribution 1:
> We introduce DeFoG, a novel flow-based graph generative model that effectively disentangles training and sampling for improved flexibility and efficiency;

I feel this is misleading. DFM already decouples training and sampling (see [1] and [2]).

In fact, the authors themselves dismiss their own contribution later in the introduction (lines 145 to 147): "While the DFM paradigm enables training-sampling disentanglement, it lacks a complete formulation and empirical validation on graph data. [...]".

I will happily adjust my score if the authors adjust their claims, or can provide a convincing argument that the reasoning above is incorrect.

[1] https://arxiv.org/abs/2402.04997

[2] https://arxiv.org/abs/2412.03487

[3] https://arxiv.org/abs/2412.06264

[4] https://arxiv.org/abs/2407.15595

**Essential References Not Discussed:**

None.

**Experimental Designs Or Analyses:**

The experiments make sense.

**Methods And Evaluation Criteria:**

Yes.

**Other Comments Or Suggestions:**

See other boxes.

**Other Strengths And Weaknesses:**

See other boxes.

**Questions For Authors:**

> 3.1. Learning Discrete Flows over Graphs

The notation in this section is a bit odd. It seems like the node and edge sets are completely unrelated? Both nodes and edges are integers. I would expect the edge set to be something like $\mathcal{X}^2$.

Given the definition of $p_{t|1}$ in this section, it is theoretically possible to have “floating” edges at any time t. In other words, edges that are not connected to any nodes. Is this correct?

**Relation To Broader Scientific Literature:**

This relates to the flow matching, as well as graph generation, literature.

**Theoretical Claims:**

Somewhat.

> Corollary 1 and 2

I don’t fully understand the need for these statements, given what is already known about sampling errors in the CTMC literature. Is having $O(\Delta t)$ error any different from the usual $O(h)$ error from using Euler’s method (see [2], [3] and [4])? To be more specific: the DeFoG loss is an evidence lower bound (ELBO) (see [2]).

---

> ### Author Rebuttal · Authors · 2025-03-31
>
> We thank the reviewer for their time and comments. We address the raised concerns below:
>
> **Claims And Evidence**.
>
> We agree with the reviewer that the training-sampling disentanglement is a contribution of DFM. In this regard, our contribution lies in making this decoupling effective for graph generation, where it had not been previously formulated, implemented, or empirically validated (as acknowledged in lines 145–150). Crucially, notice that applying existing DFM methods to graphs does not lead to improved performance (see Fig. 2 ablations). DeFoG addresses this by providing a graph tailored formulation that *effectively* leverages the decoupling between training and sampling, resulting in significant improvements in both sampling flexibility and efficiency.
>
> We understand that the original phrasing could be misleading and so we propose the revised formulation:
> ```
> We introduce DeFoG, a novel graph generative model that effectively exploits the training-sampling decoupling inherited from its flow-based formulation, significantly enhancing sampling flexibility and efficiency.
> ```
> We hope this effectively addresses the reviewer’s concerns and remain available for further discussion.
>
> **Theoretical claims**:
>
> - Our Corollary 1 does not relate to the CTMC sampling error. Instead, it establishes that minimizing the CE loss directly corresponds to minimizing an upper bound on the rate matrix estimation error, directly promoting accurate sampling. This justifies the use of the CE loss without requiring ELBO-derived approximations (more on this below).
>
> - Corollary 2 is not solely about the discretization error resulting from the CTMC sampling - which, as correctly noted by the reviewer is $O(\Delta t)$ (same as $O(h)$). Instead, it combines this term with the estimation error that results from Corollary 1 to bound the deviation of the generated graph distribution. Notice that this type of result is commonly sought in generative modeling to guarantee generation accuracy. For instance, Theorem 1 in [5] presents an analogous bound in the context of discrete diffusion, though with significant differences (e.g., they assume a bounded rate matrix estimation error, while we explicitly prove it).
>
> Regarding the connection between DeFoG’s loss (CE) and ELBO, [1] derives an ELBO loss suitable to our chosen rate matrices (decomposed into three terms weighted CE, KL divergence, rate matrix regularizer). However, they motivate using only the CE loss based on approximations upon the derived ELBO by dropping the rate matrix-dependent terms (see App. C.2 from [1]). In contrast, our bounds do not require such simplifications and are agnostic to rate matrix design choices (e.g., stochasticity level), reinforcing our training-sampling decoupling claim. Additionally, [2], which is a contemporaneous submission under ICML guidelines (alongside [3]), also establishes an ELBO loss; however, on our understanding, it applies to a different family of conditional rate matrices.
>
> We have included the discussion above, as well as the mentioned references, into the revised manuscript. We hope this clarifies the raised concern; if not, we remain open to further feedback.
>
> **Questions For Authors**:
>
> We use $x^{(n)}$ for the $n$-th node ($1 \leq n \leq N$) and $e^{(ij)}$ for the edge between nodes $x^{i}$ and $x^{j}$ ($1 \leq i < j \leq N$). Nodes are gathered as $x^{1:n:N}$, edges as $e^{1:i<j:N}$. Node and edge state spaces are denoted by $\mathcal{X}$ and $\mathcal{E}$ (cardinalities $X$, $E$, respectively), meaning nodes take values in $\\{ 1,\dots,X \\}$ and edges in $\\{ 1,\dots,E \\}$. Following standard practice in the field [6,7,8], we keep the number of nodes fixed throughout trajectories and explicitly model a complete graph, connecting all node pairs by edges. During diffusion trajectories, only node and edge classes change, not their structural existence. Crucially, one of the edge classes represents the absence of an edge ("non-existing” edge), but there is no “non-existing” class for nodes. Hence, each edge is always associated with its two vertices (there can be “floating” nodes, but no “floating edges”). We thank the reviewer for raising this important remark; we have made this more explicit in the updated manuscript.
>
> Overall, we appreciate the reviewer’s constructive feedback, and we think their questions improved our work. We remain open to any further suggestions.
>
>
> [5] - A Continuous Time Framework for Discrete Denoising Models, Campbell et al., NeurIPS 2022
>
> [6] - Digress: Discrete denoising diffusion for graph generation, Vignac et al., ICLR 2023
>
> [7] - Discrete-state Continuous-time Diffusion for Graph Generation, Xu et al., NeurIPS 2024
>
> [8] - Cometh: a Continuous-time Discrete-state Graph Diffusion Model, Siraudin et al., ArXiv 2024

---

### Decision · Program_Chairs · 2025-05-01

**Decision:**

Accept (oral)

**Comment:**

All reviewer suggested to accept the paper and the rebuttal could clarify a few remaining points, leading to reviewers increasing their score. The reviewers particularly liked the systematic empirical evaluation and hope for an easy to use code-base.